# Touchstone Benchmark: Are We on the Right Way for Evaluating AI Algorithms for Medical Segmentation?

**Pedro R. A. S. Bassi**[1,2,3]*    **Wenxuan Li**[1]*    **Yucheng Tang**[4]    **Fabian Isensee**[5,6]
**Zifu Wang**[7]    **Jieneng Chen**[1]    **Yu-Cheng Chou**[1]    **Saikat Roy**[5,8]    **Yannick Kirchhoff**[5,8,9]
**Maximilian Rokuss**[5,8]    **Ziyan Huang**[10]    **Jin Ye**[11]    **Junjun He**[11]    **Tassilo Wald**[5,6]
**Constantin Ulrich**[5]    **Michael Baumgartner**[5,6]    **Klaus H. Maier-Hein**[5,12]    **Paul Jaeger**[6,13]
**Yiwen Ye**[14]    **Yutong Xie**[15]    **Jianpeng Zhang**[16]    **Ziyang Chen**[14]    **Yong Xia**[14]
**Zhaohu Xing**[17]    **Lei Zhu**[17, 18]    **Yousef Sadegheih**[19]    **Afshin Bozorgpour**[19]
**Pratibha Kumari**[19]    **Reza Azad**[20]    **Dorit Merhof**[19,21]    **Pengcheng Shi**[22]
**Ting Ma**[22]    **Yuxin Du**[10,23]    **Fan Bai**[10,23,24]    **Tiejun Huang**[23,25]    **Bo Zhao**[10,23]
**Haonan Wang**[18]    **Xiaomeng Li**[18]    **Hanxue Gu**[26]    **Haoyu Dong**[26]
**Jichen Yang**[26]    **Maciej A. Mazurowski**[26]    **Saumya Gupta**[27]    **Linshan Wu**[18]
**Jiaxin Zhuang**[18]    **Hao Chen**[28]    **Holger Roth**[4]    **Daguang Xu**[4]
**Matthew B. Blaschko**[7]    **Sergio Decherchi**[29]    **Andrea Cavalli**[2,29,30]
**Alan L. Yuille**[1]†    **Zongwei Zhou**[1]†

[1]Department of Computer Science, Johns Hopkins University
[2]Department of Pharmacy and Biotechnology, University of Bologna
[3]Center for Biomolecular Nanotechnologies, Istituto Italiano di Tecnologia
[4]NVIDIA
[5]Division of Medical Image Computing, German Cancer Research Center (DKFZ)
[6]Helmholtz Imaging, German Cancer Research Center (DKFZ)
Full affiliations are given in Appendix F.

Code, Models & Data: https://github.com/MrGiovanni/Touchstone
Leaderboard: https://mrgiovanni.github.io/Leaderboard

## Abstract

*How can we test AI performance?* This question seems trivial, but it isn't. Standard benchmarks often have problems such as in-distribution and small-size test sets, oversimplified metrics, unfair comparisons, and short-term outcome pressure. As a consequence, good performance on standard benchmarks does not guarantee success in real-world scenarios. To address these problems, we present Touchstone, a large-scale collaborative segmentation benchmark of 9 types of abdominal organs. This benchmark is based on 5,195 training CT scans from 76 hospitals around the world and 5,903 testing CT scans from 11 additional hospitals. This diverse test set enhances the statistical significance of benchmark results and rigorously evaluates AI algorithms across out-of-distribution scenarios. We invited 14 inventors of 19 AI algorithms to train their algorithms, while our team, as a third party, independently evaluated these algorithms. In addition, we also evaluated pre-existing AI frameworks—which, differing from algorithms, are more flexible and can support different algorithms—including MONAI from NVIDIA, nnU-Net from DKFZ, and numerous other open-source frameworks. We are committed to expanding this benchmark to encourage more innovation of AI algorithms for the medical domain.

---

*Equal contribution. Authors are permitted to list their name first in their CVs.
†Correspondence to: Alan L. Yuille (AYUILLE1@JHU.EDU) and Zongwei Zhou (ZZHOU82@JH.EDU)

38th Conference on Neural Information Processing Systems (NeurIPS 2024) Track on Datasets and Benchmarks.

# 1 Introduction

The development of AI algorithms has led to enormous progress in medical segmentation, but few algorithms are reliable enough for clinical use [3, 35, 10]. Most AI algorithms fall short of expert radiologists, who are much more reliable and consistent when dealing with medical images from multiple hospitals, varied in different scanners, clinical protocols, patient demographics, or disease prevalences [68, 46, 33, 89]. Therefore, the question remains: *How can we test medical AI in the diverse scenarios that are encountered by radiologists?* Establishing a trustworthy AI benchmark is important but exceptionally challenging, and seldom achieved in the medical domain. Tougher tests, like out-of-distribution evaluation on large, varied datasets, are needed.

Standard benchmarks have underlying problems that cause confusion in algorithm comparisons and delay progress. ***First****, in-distribution test sets.* In the medical domain, CT scans in the test set often share sources, scanners, and populations with the training set. As a result, AI algorithms may perform well on the test set but generalize poorly to out-of-distribution (OOD) scenarios [21, 7, 8, 46, 33]. For example, Xia et al. [80] found that AI algorithms trained on data from Johns Hopkins Hospital (Baltimore, USA) lose accuracy in pancreatic tumor detection when evaluated on CT scans from Heidelberg Medical School (Heidelberg, Germany). ***Second****, small-size test sets.* Annotating medical data is expensive and time-consuming, but training AI requires substantial annotated data [59, 60]. Therefore, most annotated data is used for training, leaving very little assigned for testing. Recent CT datasets such as TotalSegmentator [77], WORD [52], and MSD [2], offered fewer than 100 CT scans for testing. Even a single success or failure can skew results, reducing the statistical power and potentially misleading conclusions. ***Third****, over-simplified metrics.* Most standard benchmarks only compare average performance, failing to identify each AI algorithm's strengths and weaknesses in different scenarios. For instance, one algorithm might excel at segmenting small, circular structures (like the gall bladder) while another performs better on long, tubular ones (such as the aorta). Average performance across many classes can hide these nuances. ***Fourth****, unfair comparisons.* Almost every paper reports that the newly 'proposed AI' outperforms existing 'alternative AIs.' The improvement becomes more significant if alternative AIs are reproduced and evaluated on an unknown training/test split. There are biases in comparison due to asymmetric efforts made in optimizing the proposed and alternative AIs. Many independent studies have reported these comparison biases over the years [35, 37] but remain unresolved. There is a need to have more widely adopted benchmarks (e.g., challenges) where all AI algorithms are trained by their inventors and evaluated by third parties. ***Fifth****, short-term outcome pressure.* Standard benchmarks are often in short-term and non-recurring, requiring a final solution within several months. For example, RSNA 2024 Abdominal Trauma Detection [15] only opened for three months for data access and AI development & evaluation. The short-term outcome pressure can discourage new classes of AI algorithms that need considerable time and computational resources for a thorough investigation, as their vanilla versions (e.g., Mamba [22, 85] in early 2024 and Transformers [16] in early 2021) might not outperform all the alternatives judged. The benchmark must have long-term commitment and allowance.

To address this AI mismeasurement issue, we present the Touchstone benchmark, an effort towards the objective of creating a fair, large-scale, and widely-adopted medical AI benchmark. Its scale is large, featuring a training set of 5,195 publicly available CT scans from 76 hospitals and a test set of 5,903 CT scans from additional 11 hospitals. Test sets were unknown to the participants of the benchmark. All 11,098 scans are annotated per voxel for 9 anatomical structures. The training set annotations were created by collaboration between AI specialists and radiologists followed by manual revision [60], 5,160 out of 5,903 test scans are proprietary and manually annotated, and the remaining test datasets are publicly available, annotated by AI-radiologist collaboration. As of May 2024, 14 global teams from eight countries have contributed to our benchmark. These teams are known for inventing novel AI algorithms for medical segmentation. In summary, the Touchstone benchmark explores an evaluation philosophy defined by the following **five contributions**:

1. *Evaluating on out-of-distribution data:* The JHH test set (Sec. 2.1) presents 5,160 CT scans from an hospital never seen during training, introducing a new scale of external validation for abdominal CT benchmarks. The test data distribution varies in contrast enhancement (pre, venous, arterial, post-phases), disease condition (30% containing abdominal tumors at varied stages), demographics (age, gender, race), image quality (e.g., slice thickness of 0.5–1.5 mm), and scanner types. We have collected metadata information for 72% of the test set ($N$=5,160) and reported AI performance in each sub-group.

2. *Providing a large test set:* Our test set ($N$=5,903) is much larger than the test sets of all current public CT benchmarks combined. It can enhance the statistical significance of the benchmark results: a 1% average accuracy increment across 5,000 CT scans is more indicative of a genuine algorithmic improvement than a 1% variation across 50 CT scans.

3. *Analyzing pros/cons from multiple perspectives:* We evaluated segmentation performance of 9 anatomical structures, comparing the average results and analyzing them by metadata groups. We also reported per-class algorithm rankings and visualized worst-case performance. Moreover, we assessed inference time and computational cost, key factors for the clinical deployment of AI algorithms.

4. *Inviting inventors to train their own algorithms:* Each AI algorithm is configured by its own inventors, who know it best and have the most interest in its success. In our benchmark, each inventor trained their AI algorithm on 5,195 annotated CT scans in AbdomenAtlas [60], and we, as a third party, independently evaluated these algorithms on 5,903 CT scans that are unknown and inaccessible to the AI inventors. This setting protects the integrity of our results (i.e., precluding the use of test data for hyperparameter tuning).

5. *Evaluating new algorithms with long-term commitment:* Our Touchstone benchmark not only invited established AI algorithms that are already published in major conferences/journals, but also invited newly developed algorithms appearing in recent pre-prints. We have a long-term commitment to this benchmark by organizing recurring challenges for at least five years, curating larger datasets, and improving label quality and task diversity. The first edition was featured as an invitation-only challenge at ISBI-2024.

**Related benchmarks/challenges & our innovations.** In a general sense, we define a *benchmark* as an algorithmic comparison. Accordingly, the most common type of benchmark are the standard comparisons found in thousands of research papers [58, 90, 91, 12, 27, 26, 48, 79] where authors present new algorithms and compare baselines. As previously explained, this type of benchmark incurs the risk of unfairness, due to possible asymmetric efforts made in optimizing the proposed and alternative algorithms. However, open *challenges* are a different type of benchmark, where developers train their own algorithms and submit them for third-party evaluation, mitigating the risk of unfair comparisons. For this reason, Table 1 contrasts our Touchstone benchmark to a non-exhaustive list of the most influential abdominal CT segmentation challenges. Notably, our training dataset is considerably larger and comes from more hospitals than any CT dataset ever used in a challenge. Furthermore, the only challenge training datasets on a scale similar to AbdomenAtlas 1.0 have partial labels and/or unlabeled portions [2, 53]. Our dataset is 17.3× larger than the second-largest fully-annotated CT dataset [29] in Table 1. Boosting our results' statistical significance, our evaluation dataset is 8.6× larger than any CT segmentation challenge test dataset. Moreover, Touchstone is the only benchmark in Table 1 to, simultaneously, explicitly analyze the performance of AI algorithms controlled by age, sex, race, and other metadata information. Lastly, this work is the starting point of a long-term benchmark, which we commit to maintain and improve over the years. Considering the importance of long-term commitment, we must acclaim KiTS, an abdominal segmentation challenge that had 3 editions since 2019 [31, 30, 28, 29] and FLARE, a challenge being consistently held yearly since 2021 [57, 53, 55, 56].

## 2 Touchstone Benchmark

### 2.1 Datasets – Annotations, Statistics, Distribution, & Characteristics

We used one training dataset and two test datasets to perform a comprehensive out-of-distribution benchmark. The training and test datasets were collected from many hospitals worldwide. Figure 1 shows the demographics of the two test datasets, JHH and TotalSegmentator; Appendix Figures 3–4 provide examples of CT scans and per-voxel annotations for various demographic groups across all datasets. The JHH dataset is proprietary and used for third-party evaluation; participants do not have access to the CT scans or their annotations. TotalSegmentator is a publicly available dataset; we did not inform the inventors beforehand of its use in our evaluation and confirmed that their AI algorithms had not been trained on this dataset. We included this public dataset to enable future participants to easily compare their algorithms with our benchmark.

**AbdomenAtlas 1.0**—$N$=5,195; *publicly available for training purposes*—is the largest multi-organ fully-annotated CT dataset to date, encompassing 76 hospitals in 8 countries [60]. It leveraged a

Table 1: **Related benchmarks & our innovations.** We compare Touchstone with influential CT segmentation benchmarks in light of the five contributions presented in the introduction.

| contribution | promoting superior OOD performance with a large and diverse training dataset (#1) | | | boosting results' significance & large-scale OOD test (#1, #2) | multi-faceted evaluation (#3) | encouraging innovative AI (#4, #5) |
|---|---|---|---|---|---|---|
| benchmark | # CT scans train | # hospitals train | # countries train | # CT scans test | AI consistency analysis | targeted invitation |
| MSD-CT [2] | 947† | 1 | 1 | 465 IID | none | no |
| FLARE'22 [54] | 2,050† | 22 | 5+ | 200 IID, 600 OOD | sex, age | no |
| FLARE'23 [56] | 4,000† | 30 | n/a | n/a | n/a | no |
| KiTS21 [29] | 300 | 50+ | 1 | 100 OOD | sex, race | no |
| AMOS22-CT [38] | 200 | 3 | 1 | 78 IID, 122 OOD | none | no |
| LiTS [9] | 130 | 7 | 5 | 70 IID | none | no |
| BTCV [41] | 30 | 1 | 1 | 20 IID | none | no |
| CHAOS-CT [72] | 20 | 1 | 1 | 20 IID | none | no |
| **Touchstone (ours)** | **5,195** | **76** | **8** | **5,903 OOD** | **sex, age, race** | **yes** |

† Partially labeled: annotations for each organ do not cover the entire dataset, and/or may contain unlabeled samples.

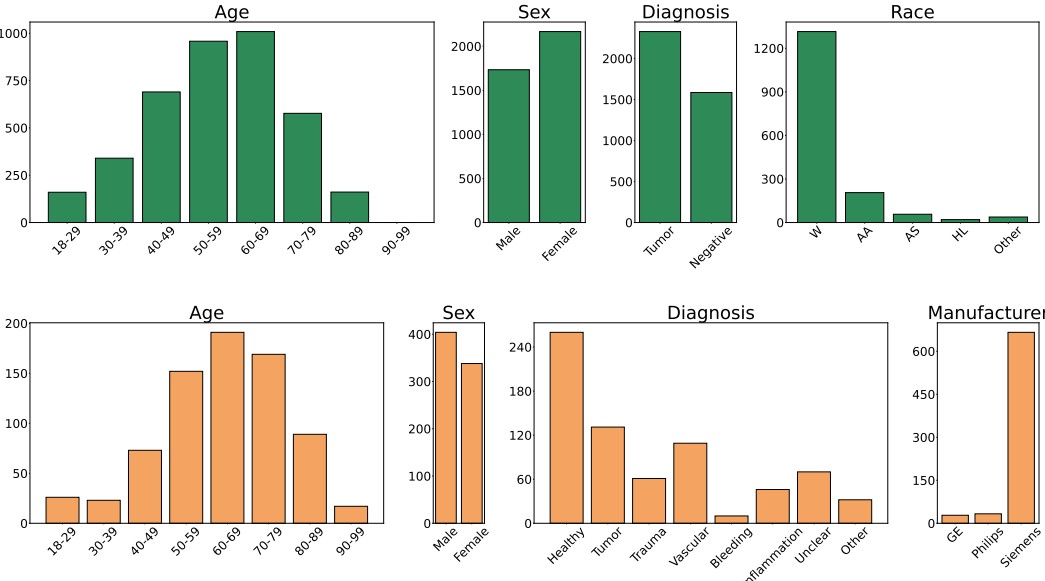

Figure 1: Summary of JHH and TotalSegmentator metadata. The diversity of data distribution includes more than just the number of centers; it also includes age, sex, manufacturer, diagnosis, and many other factors. JHH is the only dataset that provides race information, allowing us to compare the results; the race information is unknown in TotalSegmentator and most publicly available datasets. Therefore, the inclusion of JHH is value-added because it enabled the analysis on race. Races HL, W, AS, AA, O, and U indicate Hispanic & Latino, White, Asian, African American, other and unknown, respectively.

human-in-the-loop active learning strategy to empower radiologists to feasibly annotate 5,195 CT scans from 16 public datasets (listed in Appendix Table 4) and is fully annotated for 9 anatomical structures, i.e., spleen, liver, L&R kidneys, stomach, gallbladder, pancreas, aorta, and postcava. AbdomenAtlas 1.0, under CC BY-NC 4.0 License, is derived from publicly available datasets, so detailed metadata information is unfortunately not available.

**JHH**—*N =5,160; reserved for out-of-distribution test purposes*[1]—provides contrast-enhanced CT scans in venous and arterial phases. Collected from Johns Hopkins Hospital using two Siemens scanners, this dataset includes metadata on age, race, gender, and diagnosis. Notably, all per-voxel annotations in JHH were manually created by radiologists [59, 80]. Annotation time for a single

---

[1]Out-of-distribution (OOD) test data (both images and annotations) must remain private, as public release can lead to overfitting and compromise OOD evaluation integrity [21, 61]. If any OOD data is released, a new, privately preserved test set will be required to ensure reliable evaluation.

structure ranges from minutes to hours, depending on the size and complexity of the regions of interest to annotate and the local surrounding anatomical structures. Each CT scan was annotated by a team of radiologists, and confirmed by one of three additional experienced radiologists to ensure the quality of the annotation. All personally identifiable information was removed and the use of this dataset has received IRB approval from Johns Hopkins Medicine under IRB00403268. JHH is considered here an OOD test set because no CT scan from the Johns Hopkins hospital is present in the training dataset.

**TotalSegmentatorV2**—*N=743; publicly available for out-of-distribution test purposes*—is from 10 institutes within the University Hospital Basel (Switzerland) picture archiving and communication system (PACS) [77]. Being one of the largest public CT datasets, TotalSegmentator, under Apache License 2.0, was annotated by AI-assisted radiologists. It comprises both contrast-enhanced and non-contrast images, with per-sample metadata including age, sex, scanner details, diagnosis, and institution. We report AI performance on a subset of TotalSegmentator dataset[2] in Table 3 and its official test set in Appendix Tables 11–12.

## 2.2 Evaluation Protocols – Architectures, Frameworks, Metrics, & Statistical Analysis

In this study, we define an *architecture* as the overall design and structure of the entire neural network model; and define a *framework* as a set of tools or protocols that can accommodate multiple AI architectures. We evaluated 19 architectures and 3 frameworks trained by their inventors on our AbdomenAtlas 1.0[3]. We used Dice Similarity Coefficient (DSC) and Normalized Surface Distance (NSD) to evaluate segmentation performance. We enforced that the inference speed must be faster than $1e^6$ mm$^3$ per second. The inference speed for each algorithm is summarized in Appendix Table 6. We employed the same computer to evaluate all submitted algorithms. Its specifications are CPU: AMD EPYC 7713 @ 2,0Ghz×64; GPU: NVIDIA Ampere A100 (80GB); RAM: 2TB. We applied statistical hypothesis testing to each possible pair of algorithms to ensure their performance differences are significant. Following Wiesenfarth et al. [78], we used the one-sided Wilcoxon signed rank test with Holm's adjustment for multiplicity at 5% significance level and summarized results in significance maps. Per-group metadata analysis in Appendix D.5 considers Kruskal–Wallis tests, followed by post-hoc Mann-Whitney U Tests with Bonferroni correction. More statistical analyses, such as ranking stability [78], are presented in Appendix D.2.

## 3 Benchmark Results

### 3.1 Performances According to Out-of-distribution Evaluation on Large Datasets

We started by comparing the average DSC score over the 9 classes. MedNeXt and MedFormer are the winners of the JHH dataset; STU-Net and ResEncL are the winners of the TotalSegmentator dataset. Among these winners, three are CNNs (STU-Net, ResEncL and MedNeXt) and one is a CNN Transformer hybrid (MedFormer). There is no significant difference among these winners at $p = 0.05$ level, evidenced by the statistical analysis in Tables 2–3. Regarding frameworks, nnU-Net [35] is the winner since 3 out of 4 of the aforementioned winners were developed on the self-configuring nnU-Net framework.

In addition to reporting the average performance ranking, we examined the per-class performance and made the following findings. ***First****, diversified OOD evaluation is necessary.* For multiple algorithms, the DSC score for a given organ varied 15% or more across diverse test sets. E.g., the SAM-Adapter, a transformer-based 2D model, generalizes much better to JHH than to TotalSegmentator: in kidney segmentation, its DSC score differs by more than 80% across the datasets (see Appendix D.3.5 for explanations). Such stark performance variations reveal the importance of evaluating models on diverse OOD test sets. ***Second****, test dataset size matters.* More test samples increase statistical power, enabling benchmarks to more reliably detect differences between algorithms and produce stable, trustworthy rankings. Higher statistical power allows us to better differentiate the best performing

---

[2]TotalSegmentator offers 1,228 CT scans, but 485 scans were included into FLARE and subsequently inherited by AbdomenAtlas 1.0. As a result, we used only the remaining 743 scans for evaluation. Unlike JHH, this evaluation set does not come from completely unseen hospitals. However, there is a significant distribution shift between the TotalSegmentator data within AbdomenAtlas and the data in our test set (see Appendix A.2).

[3]Appendix B.1–B.3 describe in-depth the description and configuration of each architecture/framework.

Table 2: **External validation on proprietary JHH dataset ($N$=5,160).** Performance is given as DSC score (mean±s.d.). For each class, we bold the best-performing results and highlight the runners-up, which show no significant difference from the best results at $p = 0.05$ level, in red. Architectures are grouped by their frameworks and sorted in ascending order based on the number of parameters. CNNs based on the nnU-Net framework have the best performance on most classes, but other models excel at specific structures (e.g., the graph neural network-based NeXToU for aorta, and the diffusion-based Diff-UNet for kidneys). The NSD results are reported in Appendix Table 9. We measured inference speed in cm³/s (see Table 6 for details).

| framework | architecture | param | spleen | kidneyR | kidneyL | gallbladder | liver | speed | stomach | aorta | postcava | pancreas | average |
|---|---|---|---|---|---|---|---|---|---|---|---|---|---|
| nnU-Net | UniSeg† [84] | 31.0M | 94.9±6.0 | 92.2±7.2 | 91.5±7.0 | 84.7±12.6 | 96.1±4.4 | 198 | 93.3±6.0 | 82.3±10.3 | 81.2±8.1 | 82.7±10.4 | 88.8±8.0 |
| | MedNeXt [65] | 61.8M | 95.2±6.3 | 92.6±7.4 | 91.8±7.3 | 85.3±12.9 | 96.3±4.5 | 308 | 93.5±6.0 | 83.1±10.2 | 81.3±8.3 | 83.3±11.0 | **89.2±8.2** |
| | NexToU [67] | 81.9M | 94.7±8.1 | 90.1±9.5 | 89.6±9.3 | 82.3±17.0 | 95.7±5.5 | 654 | 92.7±7.5 | **86.4±8.7** | 78.1±9.1 | 80.2±13.5 | 87.8±9.8 |
| | STU-Net-B [34] | 58.3M | 95.1±6.4 | 92.5±7.3 | 91.9±7.2 | 85.5±12.3 | 96.2±4.8 | 418 | 93.5±6.0 | 82.1±10.5 | **81.3±8.2** | 83.2±10.7 | 89.0±8.1 |
| | STU-Net-L [34] | 440.3M | 95.2±6.1 | 92.5±7.1 | 91.8±7.1 | 85.7±11.8 | 96.3±4.4 | 179 | 93.7±5.6 | 81.0±10.9 | 81.3±8.2 | 83.4±10.7 | 89.0±8.0 |
| | STU-Net-H [34] | 1457.3M | 95.2±5.9 | 92.6±6.9 | 91.9±7.1 | **86.0±11.6** | 96.3±4.4 | 73 | **93.7±5.7** | 81.1±10.9 | 81.1±8.2 | **83.4±10.7** | 89.0±7.9 |
| | U-Net [63] | 31.1M | 95.1±6.3 | 92.7±6.9 | 91.9±7.2 | 84.7±13.1 | 96.2±4.5 | 1064 | 93.3±6.0 | 82.8±10.2 | 81.0±8.2 | 82.3±11.4 | 88.9±8.2 |
| | ResEncL [35, 37] | 102.0M | 95.2±6.3 | 92.6±7.0 | 91.9±6.9 | 84.9±13.0 | 96.3±4.5 | 794 | 93.4±6.0 | 81.4±11.1 | 80.5±8.8 | 82.9±10.8 | 88.8±8.3 |
| | ResEncL★ | 102.0M | 95.1±6.2 | 92.7±6.9 | 91.9±7.1 | 84.9±12.8 | 96.3±4.5 | 794 | 93.5±5.9 | 88.0±7.3 | 80.5±8.7 | 82.8±11.1 | 89.5±7.8 |
| Vision-Language | U-Net & CLIP [47] | 19.1M | 94.3±6.9 | 91.9±7.8 | 91.1±8.8 | 82.1±15.4 | 96.0±4.3 | 543 | 92.4±6.8 | 77.1±12.7 | 78.5±9.6 | 80.8±11.5 | 87.1±9.3 |
| | Swin UNETR & CLIP [47] | 62.2M | 94.1±7.7 | 91.7±9.1 | 91.0±9.1 | 80.2±18.3 | 95.8±5.6 | 606 | 92.2±8.3 | 78.1±12.6 | 76.8±11.0 | 80.2±12.5 | 86.7±10.5 |
| MONAI | LHU-Net [66] | 8.6M | 94.9±6.3 | 92.5±7.0 | 91.8±7.4 | 83.9±14.5 | 96.2±4.3 | 2273 | 93.0±6.1 | 79.5±11.2 | 79.4±9.3 | 81.0±11.3 | 88.0±8.6 |
| | UCTransNet [73] | 68.0M | 90.2±11.9 | 86.5±14.6 | 86.9±12.8 | 77.8±19.5 | 93.6±6.4 | 1163 | 81.9±12.9 | **86.5±8.0** | 68.1±15.8 | 59.0±21.6 | 81.1±13.7 |
| | Swin UNETR [69] | 72.8M | 92.7±8.8 | 89.8±11.1 | 89.7±10.2 | 76.9±20.7 | 95.2±5.3 | 2222 | 90.5±8.6 | 77.2±15.1 | 75.4±11.8 | 75.6±14.5 | 84.8±11.8 |
| | UNesT [86] | 87.2M | 93.2±7.1 | 90.9±8.1 | 90.1±8.2 | 75.1±21.2 | 95.3±5.0 | 2703 | 90.9±7.3 | 77.7±16.1 | 74.4±11.8 | 76.2±12.1 | 84.9±10.8 |
| | UNETR [25] | 101.8M | 91.7±10.1 | 90.1±9.4 | 89.2±9.6 | 74.7±20.4 | 95.0±5.3 | 2703 | 88.8±8.4 | 76.5±16.4 | 71.5±12.8 | 72.3±14.5 | 83.3±11.9 |
| | SegVol† [18] | 181.0M | 94.5±6.9 | 92.5±7.1 | 91.8±7.3 | 79.3±18.8 | 96.0±4.7 | 1923 | 92.5±7.0 | 80.2±11.3 | 77.8±9.7 | 79.1±12.4 | 87.1±9.5 |
| n/a | SAM-Adapter† [23] | 11.6M | 90.5±8.8 | 90.4±7.9 | 87.3±9.6 | 49.4±22.9 | 94.1±5.3 | 1639 | 88.0±9.3 | 62.8±12.2 | 48.0±14.2 | 50.2±12.6 | 73.4±11.4 |
| | MedFormer [19] | 38.5M | **95.5±6.1** | 92.8±7.3 | 91.9±7.4 | 85.3±13.6 | **96.4±4.4** | 535 | 93.4±6.4 | 82.1±11.7 | **80.7±10.1** | 83.1±11.2 | **89.0±8.7** |
| | Diff-UNet [82] | 434.0M | 95.0±6.9 | **92.8±7.4** | **91.9±7.5** | 83.8±14.8 | 96.2±4.7 | 442 | 93.1±6.5 | 81.2±11.3 | 80.8±8.9 | 81.9±11.4 | 88.5±8.8 |

† These architectures were pre-trained (Appendix B.3).
★ These architectures were trained on AbdomenAtlas 1.0 with enhanced label quality for the aorta and kidney classes (discussed in §4).

model from the others: for JHH ($N$=5,160), there is at most four winners for any class, but for TotalSegmentator, there is up to eight (Tables 2–3). Appendix D.4 uses box-plots and significance heatmaps [78] to confirm these findings, and Appendix D.2 shows ranking order is much more stable for JHH than for smaller test sets. This finding emphasizes the importance of test dataset size for accurate and reliable algorithm comparisons. ***Third**, average-based rankings are not enough.* Tables 2–3 show that, for the same AI algorithm, DSC scores on difficult-to-segment structures, like the gallbladder and the pancreas, are usually 10–20% lower than performance on easily identifiable structures, like the liver and the spleen. Usually, the best models for average DSC are also the best at individual structures, but per-class results reveal notable exceptions. E.g., in JHH, NexToU, a graph neural network-based hybrid architecture, excels at aorta segmentation, and Diff-UNet, a diffusion-based model, excels at kidney segmentation. Accordingly, per-class results reveal hidden strengths of

Table 3: **Validation on TotalSegmentator ($N$=743).** Performances given as DSC score (mean±s.d.). For each class, we bold the best-performing results and highlight the runners-up, which show no significant difference from the best results at $p = 0.05$ level, in red. To ease the direct comparison with other literature, we also reported the *official* test set performance in Appendix Tables 11–12. We measured inference speed in cm$^3$/s (see Table 6 for details).

| framework | architecture | param | spleen | kidneyR | kidneyL | gallbladder | liver |
|---|---|---|---|---|---|---|---|
| nnU-Net | UniSeg† [84] | 31.0M | 89.4±19.4 | 84.5±23.8 | 81.9±27.9 | 74.6±27.4 | 91.7±16.5 |
| | MedNeXt [65] | 61.8M | 91.6±18.3 | 85.5±24.8 | 86.0±23.8 | 75.8±28.5 | 93.0±15.8 |
| | NexToU [67] | 81.9M | 83.0±29.5 | 78.2±32.7 | 78.7±30.8 | 72.0±31.2 | 87.6±23.0 |
| | STU-Net-B [34] | 58.3M | 92.3±15.4 | 87.1±20.3 | 86.8±22.1 | **78.5±25.0** | 93.0±13.9 |
| | STU-Net-L [34] | 440.3M | 91.6±17.8 | 88.2±18.6 | 86.3±22.9 | 78.1±24.7 | **94.2±11.2** |
| | STU-Net-H [34] | 1457.3M | **92.4±14.6** | 88.9±16.3 | 86.5±23.4 | 77.7±25.4 | 94.0±11.4 |
| | U-Net [63] | 31.1M | 91.2±17.8 | 88.4±18.3 | 87.7±20.8 | 78.3±25.5 | 93.4±13.8 |
| | ResEncL [35, 37] | 102.0M | 91.8±17.5 | **88.9±18.0** | **88.2±20.5** | 78.0±25.2 | 91.7±18.4 |
| | ResEncL★ | 102.0M | 92.0±16.7 | 89.9±15.3 | 89.5±18.3 | 78.0±24.7 | 92.4±17.4 |
| Vision-Language | U-Net & CLIP [47] | 19.1M | 87.4±23.8 | 83.6±25.6 | 82.7±26.6 | 73.1±29.1 | 91.6±14.8 |
| | Swin UNETR & CLIP [47] | 62.2M | 87.1±22.4 | 81.1±29.0 | 77.0±32.3 | 70.3±31.0 | 91.6±16.0 |
| MONAI | LHU-Net [66] | 8.6M | 86.0±25.7 | 81.8±29.3 | 82.4±27.0 | 71.3±32.2 | 87.7±22.9 |
| | UCTransNet [73] | 68.0M | 76.4±34.5 | 74.3±35.2 | 62.0±41.5 | 69.6±31.9 | 82.6±28.2 |
| | Swin UNETR [69] | 72.8M | 66.3±36.4 | 59.7±39.4 | 58.5±40.2 | 50.6±40.6 | 80.2±28.7 |
| | UNesT [86] | 87.2M | 79.5±26.7 | 73.8±32.4 | 72.0±33.8 | 50.3±40.0 | 87.6±20.9 |
| | UNETR [25] | 101.8M | 60.4±37.9 | 47.9±39.6 | 41.9±39.8 | 40.0±36.9 | 78.1±29.9 |
| | SegVol† [18] | 181.0M | 87.1±23.0 | 82.8±23.5 | 82.6±24.8 | 68.1±29.3 | 89.4±20.5 |
| n/a | SAM-Adapter† [23] | 11.6M | 53.5±33.4 | 8.5±11.1 | 19.9±22.1 | 11.5±17.6 | 66.4±35.5 |
| | MedFormer [19] | 38.5M | 90.7±15.0 | 85.5±18.5 | 84.0±21.5 | 74.1±26.8 | 92.8±12.4 |
| | Diff-UNet [82] | 434.0M | 88.3±23.6 | 81.3±27.9 | 81.0±28.4 | 71.8±30.0 | 92.4±14.9 |

| framework | architecture | speed | stomach | aorta | IVC‡ | pancreas | average |
|---|---|---|---|---|---|---|---|
| nnU-Net | UniSeg† [84] | 198 | 74.0±29.5 | 69.2±31.5 | 72.8±25.9 | 70.3±30.9 | 78.7±25.9 |
| | MedNeXt [65] | 308 | 77.2±28.7 | 71.9±30.1 | 75.2±23.5 | 71.6±31.4 | 80.9±25.0 |
| | NexToU [67] | 654 | 69.0±34.7 | 61.5±33.0 | 59.4±32.7 | 66.8±32.0 | 72.9±31.1 |
| | STU-Net-B [34] | 418 | 78.6±26.5 | 74.2±28.9 | 77.3±19.6 | 74.9±27.5 | 82.5±22.1 |
| | STU-Net-L [34] | 179 | 79.7±24.6 | **75.7±27.0** | **77.6±18.7** | 75.2±27.0 | **83.0±21.4** |
| | STU-Net-H [34] | 73 | 78.5±25.5 | 74.7±28.1 | 76.9±19.0 | 74.5±27.6 | 82.7±21.3 |
| | U-Net [63] | 1064 | 78.9±26.3 | 71.0±28.4 | 76.4±21.8 | 75.2±27.0 | 82.3±22.2 |
| | ResEncL [35, 37] | 794 | 78.9±25.3 | 73.8±25.9 | 76.4±20.2 | **76.3±25.9** | 82.7±21.9 |
| | ResEncL★ | 794 | 80.9±23.0 | 84.2±20.5 | 76.3±20.0 | 77.3±24.9 | 84.5±20.1 |
| Vision-Language | U-Net & CLIP [47] | 543 | 77.7±26.8 | 59.0±32.8 | 65.8±27.2 | 74.6±25.7 | 77.3±25.8 |
| | Swin UNETR & CLIP [47] | 606 | 71.2±30.7 | 58.6±34.5 | 63.6±27.4 | 70.3±28.9 | 74.5±28.0 |
| MONAI | LHU-Net [66] | 2273 | 71.3±31.8 | 63.0±34.1 | 67.5±28.5 | 68.6±32.6 | 75.5±29.3 |
| | UCTransNet [73] | 1163 | 61.6±36.1 | 49.7±34.8 | 49.3±36.4 | 59.0±35.1 | 64.9±34.9 |
| | Swin UNETR [69] | 2222 | 52.2±35.2 | 54.5±37.0 | 38.1±34.7 | 42.3±34.5 | 55.8±36.3 |
| | UNesT [86] | 2703 | 63.9±31.5 | 54.7±37.0 | 38.9±36.2 | 50.0±33.0 | 63.4±32.4 |
| | UNETR [25] | 2703 | 42.1±32.1 | 41.0±31.4 | 41.3±32.3 | 28.2±29.2 | 46.8±34.3 |
| | SegVol† [18] | 1923 | 71.6±29.9 | 60.8±29.8 | 63.0±24.3 | 66.3±28.1 | 74.6±25.9 |
| n/a | SAM-Adapter† [23] | 1639 | 48.4±30.9 | 15.2±18.6 | 4.8±8.1 | 30.9±21.7 | 28.8±22.1 |
| | MedFormer [19] | 535 | 80.4±23.6 | 70.3±28.0 | 70.0±24.5 | 72.5±27.9 | 80.0±22.0 |
| | Diff-UNet [82] | 442 | 73.4±29.8 | 61.0±34.5 | 60.7±33.3 | 69.7±29.8 | 75.5±28.0 |

† These architectures were pre-trained (Appendix B.3).
‡ The class IVC (inferior vena cava) shares the same meaning as the class postcava in other datasets (e.g., AbdomenAtlas 1.0 and JHH).
★ These architectures were trained on AbdomenAtlas 1.0 with enhanced label quality for the aorta and kidney classes (discussed in §4).

AI algorithms. For a more comprehensive evaluation, Appendix C analyzes performance measured by NSD scores. *Fourth, inviting innovation is important.* As in past 3D medical segmentation challenges [2], CNNs with the nnU-Net framework [35] showed strong performance in our benchmark. However, by searching for innovative algorithms, sending target invitations to their inventors, and performing comprehensive evaluations, we could reveal strengths of new and less well known models, such as vision-language algorithms and Diff-UNet, the first 3D medical image segmentation method based on diffusion models, and MedFormer, a hybrid architecture that combines convolutional inductive bias with efficient, scalable bidirectional multi-head attention. Meanwhile, the LHU-Net, a hybrid architecture combining CNN and transformer attention mechanisms, excels in computational efficiency: it is 2 to 4 times faster than models with similar accuracy.

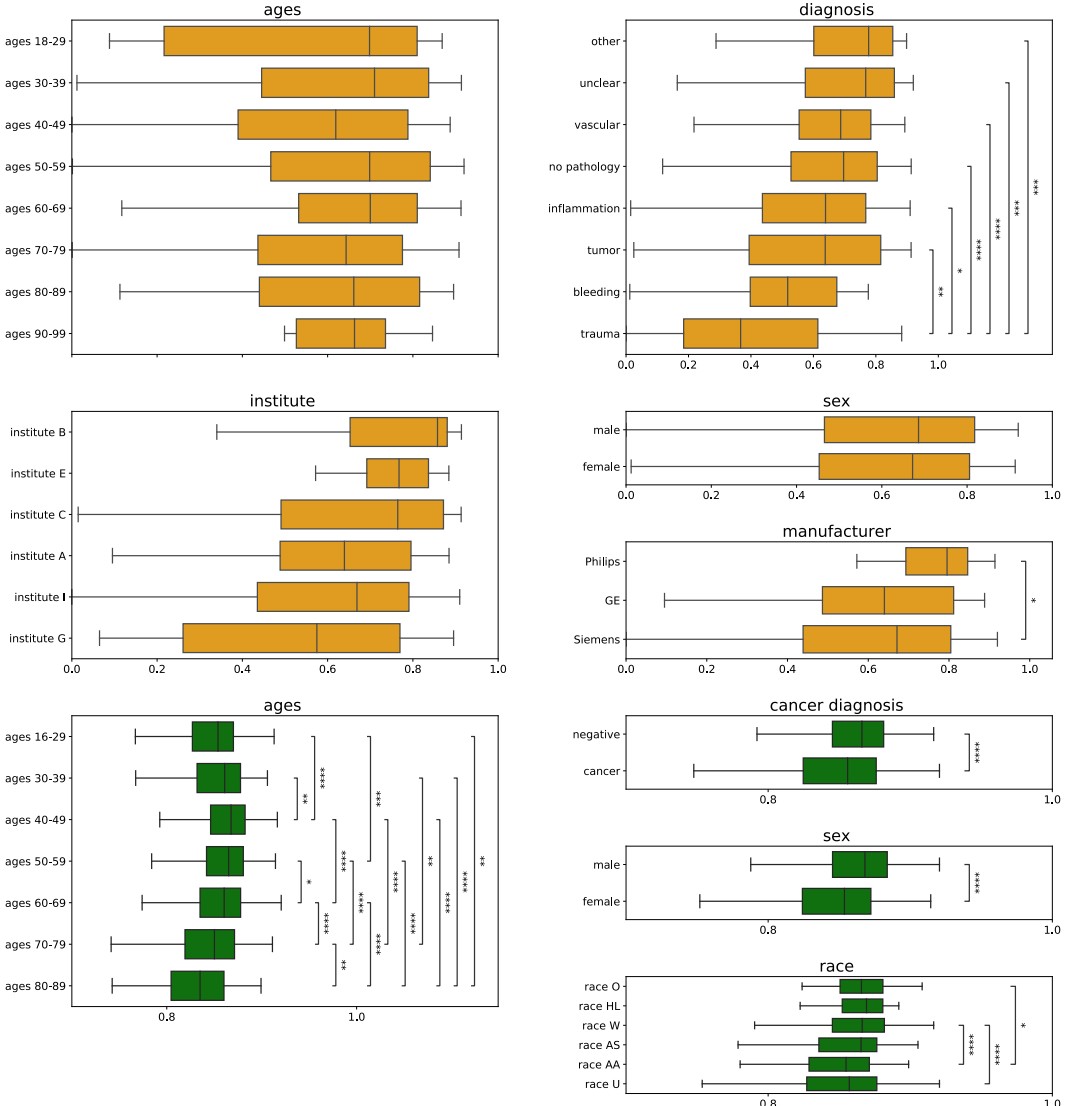

Figure 2: **Potential confounders significantly impact AI performance.** Boxplots showing the average DSC score of nine classes and 19 algorithms for diverse demographic groups in two OOD test sets: TotalSegmentator and JHH. Whiskers indicate $1.5\times$IQR (interquartile range). Statistical significance is indicated by stars: $* p < 0.05, ** p < 0.01, *** p < 0.001, **** p < 0.0001$. We perform Kruskal–Wallis tests followed by post-hoc Mann-Whitney U Tests with Bonferroni correction. Greater performance differences are observed in the JHH dataset compared to TotalSegmentator, likely due to the larger number of CT scans. Differences are apparent across demographic groups such as age, diagnoses, scanner manufacturer, sex, and medical institutions. Races HL, W, AS, AA, O, and U indicate Hispanic&Latino, White, Asian, African American, other and unknown, respectively.

## 3.2 Potential Confounders Significantly Impact AI Performance

We leveraged the metadata available in test datasets to assess AI' performance consistency across diverse demographic groups. We studied correlation between AI performance and the five types of metadata: age, sex, and diagnosis are analyzed on all two datasets, race is only analyzed on one dataset, JHH, since most public test sets lack this information, and manufacturer is only analyzed in one dataset.

Figure 2 displays per-group DSC for an average AI model, i.e., the average performance across our 19 evaluated algorithms. The statistical analysis further highlights the need for large test datasets: JHH's

large sample size ($N$=5,160) allows detection of statistically significant DSC differences across all metadata, but some of these differences (for age and sex) are noticeable but not significant in the smaller TotalSegmentator dataset. *Notably, AI performance reduces for advanced **age**.* Median DSC starts dropping around the fifties. JHH shows multiple statistically significant performance drops after this age. The creators of the TotalSegmentator observed that aging caused attenuation in CT scans [77], which may explain the common descending DSC trend after age 50, despite the fact that the 60-69 age group is the most populous in most datasets (Figure 1). This trend exists for all tested AI algorithms (Appendix D.5 displays per-group performances for each algorithm and organ). *Sex only significantly confounds some AI algorithms.* The median DSC is significantly smaller for women in JHH. However, multiple top-performing models show no significant performance difference across sexes in any dataset (e.g., nnU-Net, STU-Net, and Diff-UNet), showcasing current AI can be robust to this confounder. *We found significant performance differences for diverse **races**.* AI performance for white patients was significantly superior to the performance for African Americans, showing the need to increase the presence of this demographic group in public CT scan datasets. Again, many of the best performing algorithms did not present statistically significant differences for these two races (Appendix D.5). *In all datasets, diagnosis significantly impacted AI performance.* Cancer patients have significantly smaller DSC scores in JHH ($p < 0.0001$), and trauma patients have median DSC scores below other groups in TotalSegmentator. ***Scanner manufacturer** changes cause significant DSC differences ($p < 0.05$) in TotalSegmentator.*

## 4 Conclusion & Discussion

**Conclusion.** *Are we on the right way for evaluating AI algorithms for medical segmentation?* This paper outlines five properties of an ideal benchmark: (I) diverse data distribution in both training and test datasets, (II) a large number of test samples, (III) varied evaluation perspectives, (IV) equitably optimized AI algorithms, and (V) a long-term commitment. Touchstone sets itself apart from previous benchmarks in these criteria, enabling us to share unique insights that often missing in standard benchmarks. Our findings indicate: (1) AI performance can vary significantly across different datasets, with per-class differences of 10–20% common, and up to 80% observed (SAM-Adapter in kidney); thus, out-of-distribution evaluation across multiple datasets is crucial for ensuring AI's reliability and clinical adoption. (2) Larger test datasets reveal more significant differences between AI algorithms, allowing for meaningful rankings and nuanced analyses. (3) Average rankings can obscure AI's specific strengths; per-organ and metadata analysis is crucial in highlighting the benefits of innovative vision-language algorithms and the first diffusion-based 3D medical segmentation model. (4) By evaluating diverse AI architectures trained by their inventors, we establish a fair reference point for future development, which Touchstone will continually support with a long-term commitment.

**Label Noise in Training Set.** There is no perfect ground truth in segmentation datasets (except for synthetic data [32, 42, 13, 17, 14, 40, 45]), especially in the abdominal region where anatomical boundaries can be blurry due to disease or age (examples in Appendix A.3). Identifying these boundaries is challenging for both human annotators and AI algorithms. Many recent datasets, including TotalSegmentator [77] and AbdomenAtlas 1.0 [60], use human-in-the-loop strategies, combining AI-predicted annotations and manual annotations by radiologists, which inevitably contain label errors. The errors in AbdomenAtlas 1.0 arise from poor CT image quality (e.g., BDMAP_00000339, BDMAP_00001044, BDMAP_00003725), mistakes in AI predictions but not revised by humans, and inconsistency in label standards across the public datasets incorporated into AbdomenAtlas 1.0 [43]. With the feedback from our benchmark participants, we can *partially* detect these label errors, primarily in the aorta (32.4%), a structure with high annotation standard inconsistency in public data (e.g., in BTCV and FLARE) [47, 48], and in the L&R kidneys (2.6%). We revised AbdomenAtlas 1.0 by reducing label errors in the aorta to 5.4% and in the L&R kidneys to 0.6%. A ResEncL trained on the revised AbdomenAtlas 1.0 showed statistically significant performance gains in the aorta, but gains for kidneys were small and not always statistically significant (see Tables 2–3). These results highlight that current AI may be resistant to moderate levels of label noise (2.6%), but not to high levels (32.4%), as we detail in Appendix E. As future work, an improved label error detector will be a valuable tool for automatically assessing the quality of publicly available datasets and quickly improving quality through human annotation based on detected errors.

**High-Quality, Proprietary Test Set.** Having JHH ($N$=5,160) available for third-party evaluation is a big plus for OOD benchmarks. It was completely annotated by radiologists, manually and following

a well-defined annotation standard, for several years [59]. Thus, it can serve as a gold standard for our benchmark. The fact that JHH is a private dataset has both problems and benefits. It can significantly increases feedback time for AI performance evaluation, as it requires additional procedures to submit the AI to a third party, set it up, and run it on over 5,000 CT scans. If a benchmark takes too much work to run, it will not gain wide traction. But making test set (either images or annotations) publicly available can cause more problems—including completely destroying the OOD benchmark. For example, Medical Segmentation Decathlon (MSD) [2] was a benchmark with publicly accessible test images and its test annotations were private. Similarly, BTCV [41] released both testing images and annotations. However, due to the growing need for more annotated data in the medical domain, even MSD/BTCV test sets have been annotated and integrated into recent public datasets, like FLARE [53, 54, 56] and AbdomenAtlas [60, 44, 43]. Therefore, any AI models trained or pre-trained on these public datasets are problematic in the MSD/BTCV leaderboard. With widespread access to test data, it becomes challenging to fairly compare models, as some may be overly optimized for the benchmark rather than for real-world performance. As a result, researchers must continue to seek or develop new datasets—preferably with images and annotations that have never been disclosed. This is critical in many fields as well. Yann Lecun—*beware of testing on the training set*—in response to the incredible results achieved by GPT. Therefore, our proprietary JHH dataset is a valuable resource that other researchers can exploit to reduce data leakage risks and improve the reliability of OOD benchmark results. Our Touchstone Benchmark is still in the initial stage, so we are very careful with the decision of releasing JHH images/annotations. It must be managed carefully to ensure its benefits outweigh the risks.

**Per-Group Metadata Analysis.** Our study underscores the need for detailed metadata for algorithmic benchmark, which is currently a big limitation in the medical domain. Evidenced by Table 1, only KiTS & FLARE provided metadata analysis on sex, age, and/or race. Our Touchstone not only provides more extensive metadata analyses, including diagnosis, but also offers an order of magnitude more test data ($N$=5,903) for benchmarking. We have analyzed AI performance by metadata such as sex, age, and race but realized that a more rigorous analysis could be based on combined criteria (e.g., white females aged 30–40). Therefore, in the next round of benchmarking, instead of only providing average performance per class, we will also offer participants per-case performance along with each case's metadata information. This approach will provide a richer understanding of the pros/cons of AI algorithms and potentially stimulate AI innovation.

**Architectural Insights.** In Appendix D.3, we have provided architectural comparison of both the top-ranking and bottom-ranking algorithms. But we find it difficult to extract trustworthy architectural insights directly from our current benchmark results. For example, Tables 2–3 show that top performing models in our benchmark are usually CNNs within the nnU-Net framework. However, it is unclear if this is due to an intrinsic advantage of CNNs over Transformers or just an indication of nnU-Net's superior pipeline configuration. Given that Transformers are newer, future frameworks, designed for them, could potentially enhance their performance. I.e., mature frameworks that extract the best from both CNNs and transformers should allow fairer architectural comparisons in the future. Beyond medical imaging, the architectural debate between CNNs and Transformers in computer vision has been ongoing and remains unresolved [5, 74]. Our benchmark provides 'predictions-only' results, which can be heavily influenced by many factors such as preprocessing, data augmentation, post-processing, and training hyper-parameters [35]. To draw convincing architectural insights, extensive ablation studies under controlled settings are required. However, conducting ablation studies for all 19 AI algorithms would be extremely costly for us. We anticipate further insights and details from the AI inventors' upcoming technical reports, including extensive ablation studies. We are also happy to assist the inventors in their ablation studies by providing feedback on the OOD evaluation results of their algorithm variants.

With the success of the first edition of Touchstone Benchmark, we are actively pursuing multi-center, OOD datasets, to further enhance the benchmark. This is difficult for many well-known reasons—patient privacy, ethical compliance, data annotation, intellectual property, etc. *Rome wasn't built in a day*. A multi-center, OOD dataset can never be made without accumulating the contribution of every single-center dataset. We hope this benchmark initiative at Johns Hopkins University, a highly regarded institution, could also inspire more institutes to contribute their private datasets for third-party OOD evaluation.

## Acknowledgements and Disclosure of Funding

This work was supported by the Lustgarten Foundation for Pancreatic Cancer Research and the Patrick J. McGovern Foundation Award. We gratefully acknowledge the Data Science and Computation Facility and its HPC Support Team at Fondazione Istituto Italiano di Tecnologia. P.R.A.S.B. thanks the funding from the Center for Biomolecular Nanotechnologies, Istituto Italiano di Tecnologia (73010, Arnesano, LE, Italy). A.C. and S.D. thank the funding from the Istituto Italiano di Tecnologia (16163, Genova, GE, Italy). Z.W. and M.B.B. acknowledge support from the Research Foundation - Flanders (FWO) through project numbers G0A1319N and S001421N, and funding from the Flemish Government under the Onderzoeksprogramma Artificiële Intelligentie (AI) Vlaanderen programme. Z.W. and M.B.B. acknowledge LUMI-BE for awarding this project access to the LUMI supercomputer, owned by the EuroHPC JU, hosted by CSC (Finland) and the LUMI consortium, and EuroHPC JU for awarding this project access to the Leonardo supercomputer, hosted by CINECA. Y.S., A.B., P.K., R.A. and D.M. acknowledge the scientific support and HPC resources provided by the Erlangen National High-Performance Computing Center (NHR@FAU) of the Friedrich-Alexander-Universität Erlangen-Nürnberg (FAU) under the NHR project "DeepNeuro - Exploring novel deep learning approaches for the analysis of diffusion imaging data." NHR funding is provided by federal and Bavarian state authorities. NHR@FAU hardware is partially funded by the German Research Foundation (DFG) – 440719683. Part of this work was funded by Helmholtz Imaging (HI), a platform of the Helmholtz Incubator on Information and Data Science. This work is partially funded by NSFC-62306046. We thank Thomas Brox for supporting the benchmark of the U-Net architecture.

We thank Di Liang for providing consultant of the statistical analysis in this benchmark; thank Zeyu Zhao for creating an live leaderboard of Touchstone; thank Xiaoxi Chen for analyzing AI predictions; thank Seth Zonies and Andrew Wichmann for providing legal advice on the release of AbdomenAtlas 1.0. The content of this paper covered by patents pending.

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
