# Appendix

## Table of Contents

# A  Extensive Datasets in Touchstone

## A.1  Construction of AbdomenAtlas 1.0

Table 4: **Public datasets composing AbdomenAtlas 1.0 and their details [60].** The naive aggregation of these public datasets results in a database with partial and incomplete labels, e.g., LiTS only had labels for the liver and its tumors, and KiTS only had labels for the kidneys and its tumors. Conversely, our AbdomenAtlas 1.0 is fully-annotated, offering detailed per-voxel labels for all 9 organs within each CT scan. We detected and removed duplicated CT scans across public datasets like LiTS and FLARE'23. Duplicate scans were identified by generating a 3D perceptual hash [83] for each image in the dataset. By comparing the similarity of these hashes, duplicates were reliably detected, a finding that was further confirmed through manual inspection of CT scans with high perceptual hash similarities. After aggregating all datasets and removing duplicates, we obtained a total of 5,195 fully-annotated CT scans.

| Dataset | # of organs | # of scans | # of centers | source countries | license | # of *unique* scans in AbdomenAtlas 1.0 |
|---|---|---|---|---|---|---|
| 1. Pancreas-CT [64] | 1 | 82 | 1 | US | CC BY 3.0 | 42 |
| 2. LiTS [9] | 1 | 201 | 7 | DE, NL, CA, FR, IL | CC BY-SA 4.0 | 131 |
| 3. KiTS [30] | 1 | 300 | 50+ | US | CC BY-NC-SA 4.0 | 300 |
| 4. AbdomenCT-1K [57] | 4 | 1,112 | 12 | DE, NL, CA, FR, IL, US, CN | CC BY-NC-SA | 1000 |
| 5. CT-ORG [62] | 5 | 140 | 8 | DE, NL, CA, FR, IL, US | CC BY 3.0 | 140 |
| 6. CHAOS [72] | 4 | 40 | 1 | TR | CC BY-SA 4.0 | 20 |
| 7-11. MSD CT Tasks [2] | 9 | 947 | 1 | US | CC BY-SA 4.0 | 945 |
| 12. BTCV [41] | 12 | 50 | 1 | US | CC BY 4.0 | 47 |
| 13. AMOS22 [38] | 15 | 500 | 2 | CN | CC BY-NC-SA | 200 |
| 14. WORD [52] | 16 | 150 | 1 | CN | GNU GPL 3.0 | 120 |
| 15. FLARE'23 | 13 | 4,000 | 30 | - | CC BY-NC-ND 4.0 | 2200 |
| 16. AbdomenCT-12organ [54] | 12 | 50 | - | - | CC BY-NC-SA | 50 |

US: United States    DE: Germany    NL: Netherlands    CA: Canada    FR: France    IL: Israel
CN: China    TR: Turkey    CH: Switzerland

## A.2  Domain Shift in TotalSegmentator

Table 5: **Percentage of Missing Classes in the two Partitions of TotalSegmentator.** Part of TotalSegmentator is included in the AbdomenAtlas dataset ($N$=485), because it is contained in FLARE, one of the AbdomenAtlas constituents. We leveraged the remaining sample of TotalSegmentator ($N$=743) for testing, providing a public test set anyone can easily use to compare segmentation models to Touchstone results. Unlike for the JHH test set, the hospitals in TotalSegmentator are present in AbdomenAtlas. However, the part of TotalSementator inside AbdomenAtlas ($N$=485) and the 743 test samples are not identically distributed. Table 5 analyzes these two subsets, and shows that the one inside AbdomenAtlas was carefully selected to focus on the abdominal region, with a regular Region of Interest: almost all of these 485 images contain the 9 abdominal organs considered in this Touchstone. Conversely, the 743 TotalSegmentator images in our test set are more challenging, presenting varying regions of interest, which can extend outside of the abdomen and usually crop out some of the 9 classes in this benchmark. Therefore, Table 5 demonstrates a substantial distribution shift between the two TotalSegmentator partitions, making our TotalSegmentator test images ($N$=743) out-of-distribution and a challenging test scenario. Interestingly, our results show this scenario was even more challenging to the AI algorithms than the JHH test set, which contains only images from an unseen hospital (see Sec. 3).

| dataset | aorta | gallbladder | kidneyL | kidneyR | liver | spleen | stomach | pancreas | postcava |
|---|---|---|---|---|---|---|---|---|---|
| AbdomenAtlas1.0 | 0% | 3.9% | 0.4% | 0.4% | 0% | 0% | 0% | 0% | 0% |
| TotalSegmentator | 17.4% | 81.8% | 60.3% | 63.0% | 40.4% | 60.3% | 35.3% | 47.2% | 45.1% |

### A.3 Dataset Visualization by Metadata Information

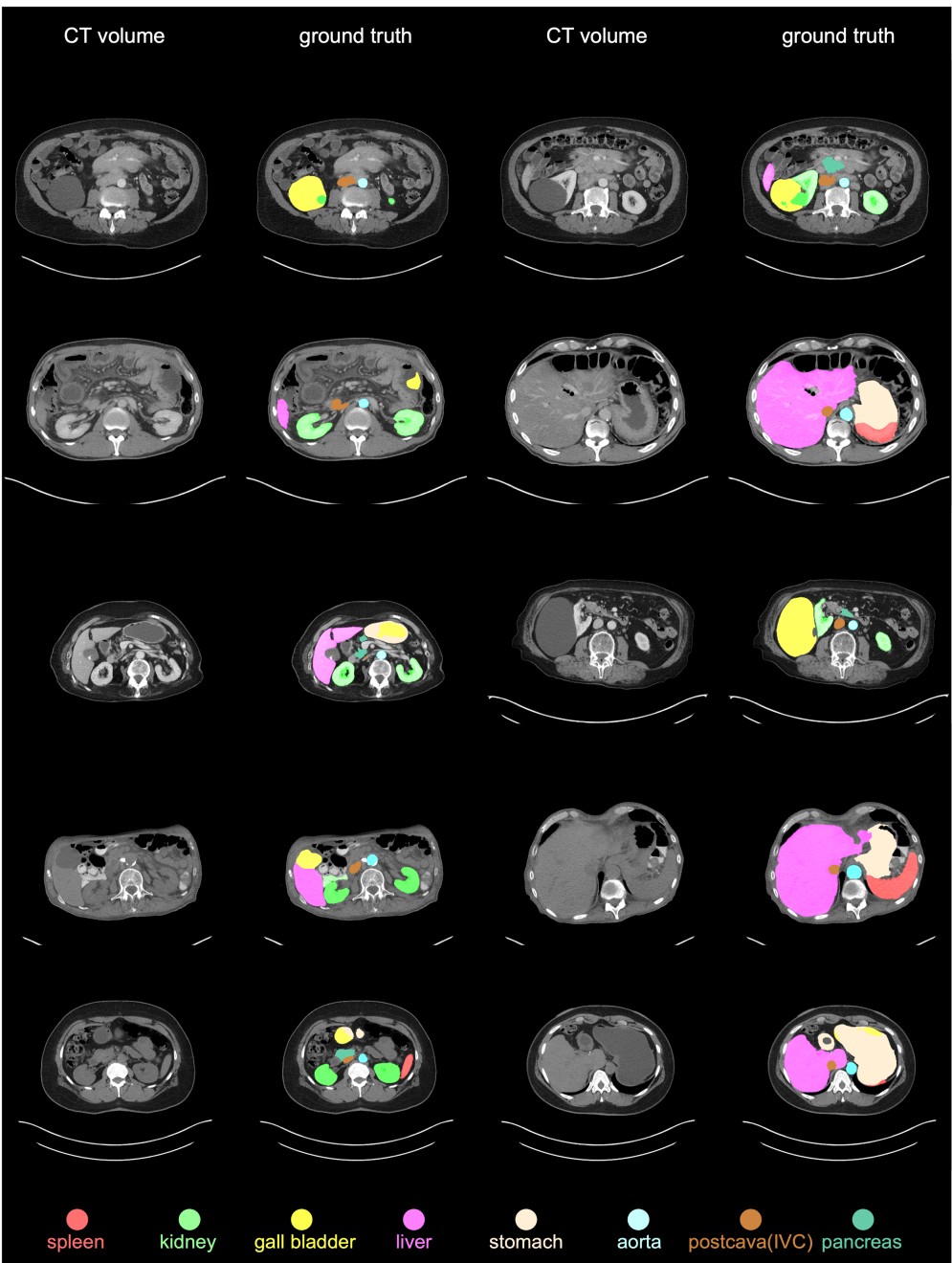

Figure 3: Anatomical boundaries and structures can be indistinct due to disease, as seen in the JHH dataset. We display CT volumes with patients depicted under unhealthy conditions that are challenging for most AI algorithms to identify. The CT volumes are from patients in unhealthy conditions. As shown in the first row on the left side, a kidney cyst is mistakenly annotated as the gall bladder. This example highlights that in the abdominal region, diseases can obscure anatomical boundaries and even lead to misidentification of structures.

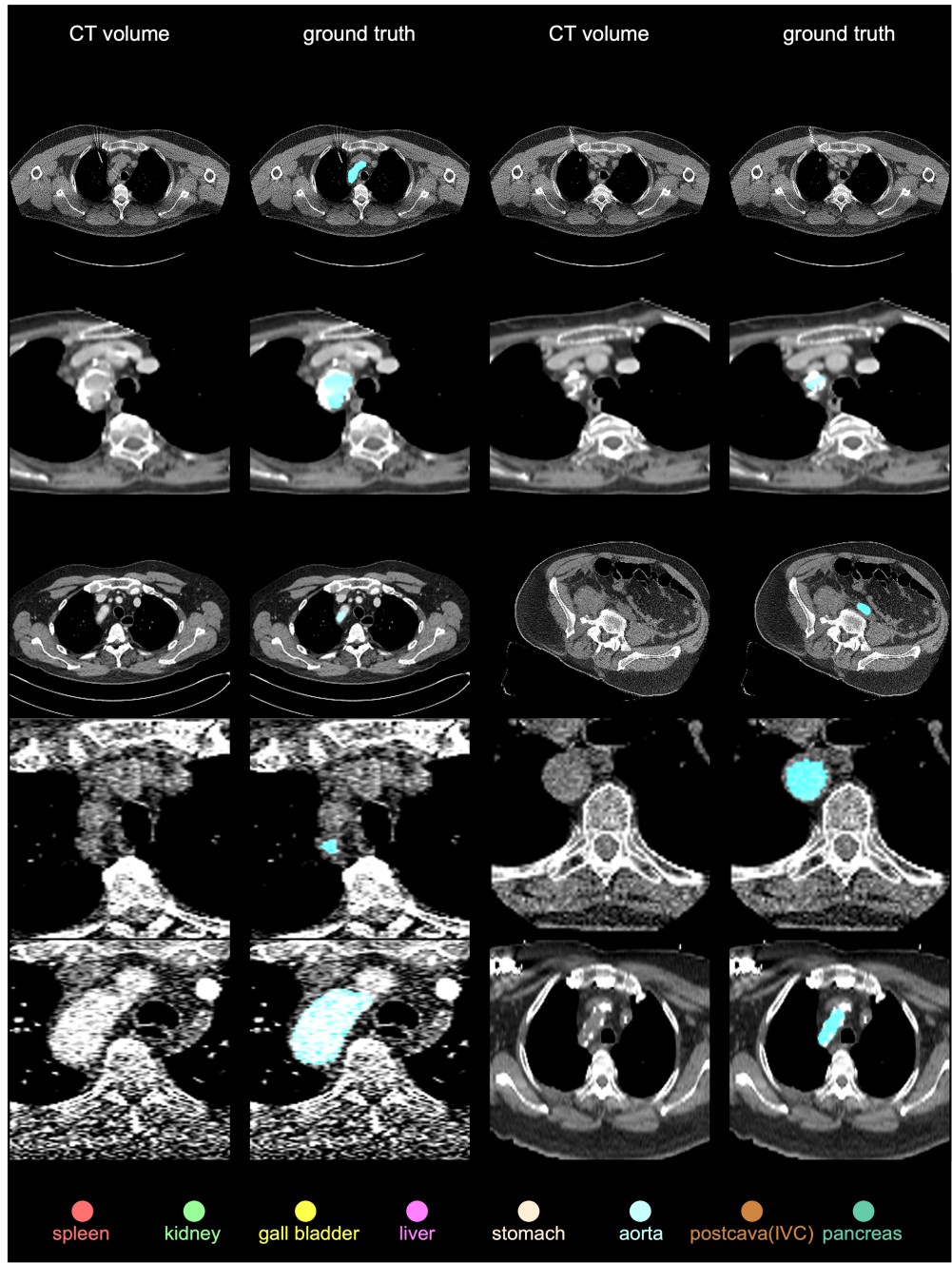

Figure 4: **Anatomical boundaries can be blurry due to factors such as patient disease, age, and CT scan quality in TotalSegmentator.** We display CT scans that are challenging for most AI algorithms to identify. The CT scans in the top three rows are from patients diagnosed with the tumors specified in the pathology metadata. The remaining images feature patients in their 70s and 80s. As shown in the fourth row on the right side, the boundary of the aorta in a 78-year-old patient is challenging to identify, not only for AI algorithms but also for human annotators in determining the ground truth.

# B    Extensive Number of AI Algorithms in Touchstone

## B.1    Description of AI Architectures

### B.1.1    Category CNN

**U-Net.** The U-Net [63] is a fully-convolutional neural network, based on an encoder-decoder structure joint by multiple skip-connections. The encoder performs down-sampling operations, and it is designed to capture high-level semantics and context information. The decoder conducts up-sampling, and the long-range skip connections allow it to fuse the high-level semantics available at deep encoder layers, with the precise spatial information extracted from earlier encoder layers. The U-Net is the most influential architecture in biomedical segmentation; almost one decade after its release, the model is still the base of multiple novel architectures in this Benchmark.

**ResEncL.** nnU-Net [35, 37] is a self-configuring segmentation framework. It automatically configures pre-processing, network architecture, training and post-processing. Auto-configuration is guided by fixed parameters, interdependent rules that consider dataset properties and computational limitations, and empirical parameters. nnU-Net's default model has recently been updated to ResEncL default, which is based on a U-Net architecture with residual connections in its encoder [37]. The encoder is computationally expensive while the decoder is as lightweight as possible. For hyper-parameter configuration the nnU-Net default values are used except for the modality which was declared as "nonCT", resulting in z-score intensity normalization. ResEncL serves as a modernized nnU-Net baseline to compare new methodological innovations against.

**MedNeXt.** MedNeXt [65] is a fully ConvNeXt-based 3D Encoder-Decoder Network designed to benefit from the scalability of Transformer-based networks while leveraging the inductive bias inherent to convolutions. This enables effective training on large datasets while still being beneficial on small data-scarce settings common to 3D medical image segmentation in the last decade. In the 3-layer residual structure of a MedNeXt block, the first layer computes features using a depthwise convolutional kernel, and it is followed by an expansion and compression layer, akin to a Swin Transformer. The architecture primarily benefits from using its MedNeXt blocks in all layers of the architecture, including up and downsampling blocks. The MedNeXt block enables effective representation learning in standard layers while allowing the network to maintain semantic richness in all resampling operations.

**STU-Net.** STU-Net [34] is a family of scalable and transferable medical image segmentation models based on the nnU-Net framework and the U-Net architecture. The STU-Net models introduce innovations such as refined convolutional blocks with residual connections for better scalability and weight-free interpolation for enhanced transferability. The models are available in different sizes: STU-Net-S with 14 million parameters, STU-Net-B (with 58.3M), STU-Net-L (440.3M), and STU-Net-H with 1.4 billion parameters. Improvements in segmentation accuracy stem from the empirical scaling of network depth and width. The primary goal of STU-Net is to enhance the scalability and transferability of medical image segmentation algorithms, facilitating their application across a variety of downstream tasks in transfer learning.

**UniSeg.** UniSeg [84] is a prompt-driven universal segmentation framework designed for multi-task medical image segmentation, offering transfer capabilities across various modalities and domains. Based on the nnU-Net framework, UniSeg has a vision encoder and a fusion module, which together enable a prompt-driven decoder. A key innovation of UniSeg is its universal learnable prompt that models complex inter-task relationships. UniSeg integrates task-specific prompts early in the training process, enhancing the training effectiveness of the entire decoder. The primary goal of UniSeg is not only to excel in multi-task learning but also to serve as a pre-trained model that improves the accuracy of downstream segmentation tasks. UniSeg was pre-trained (supervised) on 5 datasets before fine-tuning on AbdomenAtlas 1.0: MOTS [88, 81], VerSe20 [51], Prostate[49], BraTS21[6], and AutoPET2022 [20].

### B.1.2    Category Transformer

**UNETR.** UNETR [25] was proposed as a 3D transformer-based segmentation backbone network. The method leverages the Transformer model and CNN as a hybrid architecture, to capture long-range dependencies within volumetric medical data. The architecture integrates a Vision Transformer (ViT) as the encoder to handle the 3D input patches and extract rich feature representations. These

features are then progressively merged with a convolutional neural network (CNN)-based decoder in a UNet-like structure.

**Swin UNETR.** SwinUNETR [69] adapted Swin Transformers to enhance volumetric medical image segmentation by capturing both local and global features through a hierarchical, window-based self-attention mechanism, outperforming the original UNETR, and using Swin-transformers for global context. Additionally, self-supervised pre-training of Swin Transformers on large-scale unlabeled 3D medical images datasets, using techniques like masked autoencoding, can significantly boost the model's robustness and performance on downstream tasks. These features enabled leading results in various 3D medical image analysis applications, especially in CT segmentation tasks.

**UNEST.** UNEST [86] is an advanced 3D segmentation model designed to leverage the strengths of the hierarchical vision transformer architecture for handling 3D medical image data. UNEST also employed a U-shape encoder-decoder structure, where the encoder is based on the 2-stage nested ViT. This transformer-based encoder extracts hierarchical features from the input CT scan using self-attention mechanisms, which capture long-range dependencies and spatial relationships efficiently. The decoder consists of 4-levels of CNN-based blocks that reconstruct the segmentation map by upsampling the features and incorporating skip connections from the encoder to retain spatial information. The model's architecture and training protocol are optimized to provide a robust and efficient solution for 3D segmentation tasks such as whole body, regional, and whole brain segmentation.

**SegVol.** SegVol [18] is based on the SAM architecture [39] and 3D transformers, enabling universal and interactive volumetric medical image segmentation on over 200 anatomical categories. SegVol supports spatial-prompt, semantic-prompt, and combined-prompt segmentation, aiming for high-precision segmentation and semantic disambiguation. SegVol introduced a zoom-out-zoom-in mechanism to provide users with an easy SAM-like interface on volumetric images, while significantly reducing computational cost and preserving the segmentation precision. Pseudo labels are used to relieve the problem of spurious correlation between predictions and data distributions. Prior to training on AbdomenAtlas, SegVol was pre-trained on 90K unlabeled CT scans from M3D-Cap, and 5,772 labeled CT scans from M3D-Seg [4].

**SAM-Adapter.** The SAM-Adapter [23] is a 2D segmentation model, unlike the other networks in this study. Thus, it individually analyzes the 2D slices that compose a CT scan. The model is based on fine-tuning the MobileSAM [87] encoder and decoder using Adapter layers. The SAM-Adapter follows the philosophy that model size has limited effect over the accuracy of medical segmentation algorithms [23].

### B.1.3 Hybrid Architectures

**LHU-Net.** LHU-Net [66] is a compact and efficient U-Net-based architecture created for 3D medical image segmentation. It utilizes a hierarchical encoder-decoder structure with convolutional layers followed by hybrid attention mechanisms to capture both local and global features. Key innovations include the integration of CNN-based spatial attention and Vision Transformer (ViT) attention mechanisms, such as the OmniFocus attention and self-adaptive contextual fusion modules, which enhance discriminative feature extraction while keeping the model lightweight. These attention mechanisms' objective is to ensure high precision and detail in the segmentation results. The main aim of LHU-Net is to achieve high segmentation accuracy with minimal computational resources and parameters, making it a practical and accessible tool for medical imaging tasks.

**UCTransNet.** UCTransNet [73] is a hybrid architecture, based on U-Net with transformer blocks as skip connections. It introduces the Channel-wise Cross Fusion Transformer (CCT) to fuse multi-scale context with cross attention from a channel-wise perspective. CCT captures local cross-channel interaction for adaptive fusing of multi-scale features with possible scale semantic gap. Additionally, a channel-wise cross attention (CCA) module is proposed for fusing features from decoder stages and fused multi-scale features to solve inconsistent semantic levels. Both cross attention modules are called CTrans and replace the original skip connections in the U-Net. Here, the UCTransNet 2D components were substituted by their 3D versions, including convolution layers, patch embedding layers, and patch merging layers. The main goal is to discover an efficient approach for integrating CNNs and Transformers for medical image segmentation.

**Diff-UNet.** Diff-UNet [82] is the first generic 3D medical image segmentation model based on a denoising diffusion model. It mainly consists of two branches: the boundary prediction branch and the diffusion denoising branch. The boundary prediction branch is based on the U-Net structure, while the diffusion denoising branch is based on a denoising U-Net structure with noise input. To aggregate the low-level and high-level features from both branches for better boundary perception, Diff-UNet also includes a Multi-granularity Boundary Aggregation (MBA) module. Next, Diff-U-Net proposes a Monte Carlo Diffusion (MC Diffusion) module to obtain uncertainty maps and guide segmentation loss to focus on hard-to-segment regions during training. Finally, Diff-UNet devises a Progressive Uncertainty-driven REfinement (PURE) strategy to obtain a more robust prediction result during inference, based on the inference steps and uncertainty maps estimated by the MC Diffusion module.

**NexToU.** NexToU [67] is a hybrid architecture that follows a hierarchical 3D U-shaped encoder-decoder structure, based on CNNs and graph neural networks (GNNs). It incorporates a hierarchical, topology-aware strategy inspired by human cognitive processes, progressively decomposing anatomical semantics from simpler to more complex structures. Concurrently, it also learns containment, connection, and exclusion relationships among various anatomical classes. To facilitate learning and speed up training, NexToU employs a semantic tree and a novel hierarchical topological interaction (HTI) module. Additionally, it enhances spatial topology perception by incorporating Vision GNN [24] and Swin GNN modules, which adeptly represent topology on both global and local scales. The primary goal of NexToU's innovations is to improve segmentation accuracy for homogeneous multi-class anatomical structures, such as vasculature and skeletons. The HTI module is designed to be more effective when dealing with a large number of classes.

**MedFormer.** MedFormer [19] is a hybrid architecture that combines the inductive bias of convolution with the global modeling capabilities of Transformers. A key innovation in the design is the bidirectional multi-head attention (B-MHA) mechanism, which addresses the quadratic complexity typically associated with self-attention on long sequences. B-MHA employs a low-rank projection mechanism to achieve linear complexity attention, making it computationally efficient for both low- and high-resolution feature maps. Furthermore, B-MHA's architecture captures the most salient features in its hidden state, enhancing model robustness by reducing focus on irrelevant details. Through this design, MedFormer demonstrates good scalability, efficiency, and generalizability, performing effectively on both small and large datasets without requiring pre-trained weights.

## B.2 Description of AI Frameworks

### B.2.1 nnU-Net

nnU-Net [35] is a framework for automatically configuring AI-based semantic segmentation pipelines. Given a new segmentation dataset, it will extract relevant metadata from the training cases to automatically determine its hyperparameters. Despite its first release dating back to 2019 and despite its use of a standard U-Net [63], it stood the test of time and continues to produce state-of-the-art results. nnU-Net powerfully demonstrates that carefully configuring and validating segmentation pipelines across a wide range of segmentation tasks yields a surprisingly potent algorithm. As a framework for method development, it is widely used and extended by the community to push the boundaries of semantic segmentation [84, 34, 65, 67, 71, 36]. A recent update to the nnU-Net presets [37] includes reference implementations for a U-Net with residual connections in the encoder, optimized for different VRAM budgets.

### B.2.2 MONAI

MONAI (Medical Open Network for AI) [11] is an open-source framework designed to support artificial intelligence in healthcare data. Built on top of PyTorch, MONAI facilitates a comprehensive suite of tools for configuring, training, inference, and deploying AI models tailored to medical applications. It includes components for data loading, preprocessing, and augmentation, as well as prebuilt architectures for common tasks such as segmentation, registration, detection, and classification. MONAI is designed to be flexible, extensible, and performance-optimized, enabling researchers and practitioners to accelerate their AI development cycle in the medical domain.

### B.2.3 Vision-Language Models

**CLIP-Driven Universal Model.** The CLIP-Driven Universal Model [47] framework, which is designed for organ and tumor segmentation, integrates a label taxonomy from various public datasets. The architecture consists of a text branch and a vision branch. In the text branch, the model generates CLIP embeddings for each organ and tumor using label prompts, enhancing the anatomical structure of the feature embedding. These embeddings are concatenated with global image features, termed the text-based controller, to produce prompt features for segmentation. The vision branch pre-calculates CT scans to mitigate domain gaps across different datasets. These extracted features are processed by three sequential convolutional layers, referred to as the text-driven segmentor, which utilize the parameters generated by the text branch to predict segmentation masks for each class. The decoder also includes a "one vs. all" approach, using Sigmoid activation for each class to generate individual predictions, ensuring robust and dynamic segmentation across diverse medical imaging datasets.

## B.3 Implementation and Configuration Details

Tables 6-8 present details on the algorithms we benchmarked, and on their training configurations, respectively.

Table 6: **Details on the AI algorithms and speed.**

| framework | architecture | parameters | category | inference time $(\mu s/mm^3)^\dagger$ | inference memory (average)$^\dagger$ |
|---|---|---|---|---|---|
| nnU-Net | UniSeg | 31.0M | CNN | 5.04 | 3.9 GB |
| | MedNeXt | 61.8M | CNN | 3.25 | 4.3 GB |
| | NexToU | 81.9M | Hybrid | 1.53 | 1.9 GB |
| | STU-Net-B | 58.3M | CNN | 2.39 | 2.0 GB |
| | STU-Net-L | 440.3M | CNN | 5.6 | 5.4 GB |
| | STU-Net-H | 1457.3M | CNN | 13.66 | 12.5 GB |
| | U-Net | 31.1M | CNN | 0.94 | 1.9 GB |
| | ResEncL | 102.0M | CNN | 1.26 | 3.7 GB |
| Vision-Language | U-Net & CLIP | 19.1M | Hybrid | 1.84 | 8.0 GB |
| | SwinUNETR & CLIP | 62.2M | Hybrid | 1.65 | 7.5 GB |
| MONAI | LHU-Net | 8.6M | Hybrid | 0.44 | 0.6 GB |
| | UCTransNet | 68.0M | Hybrid | 0.86 | 2.8 GB |
| | SwinUNETR | 72.8M | Hybrid | 0.45 | 4.2 GB |
| | UNesT | 87.2M | Hybrid | 0.37 | 2.4 GB |
| | UNETR | 101.8M | Hybrid | 0.37 | 2.4 GB |
| | SegVol | 181.0M | Transformer | 0.52 | 0.8 GB |
| n/a | SAM-Adapter | 11.6M | Transformer | 0.61 | 0.5 GB |
| | MedFormer | 38.5M | Hybrid | 1.87 | 2.8 GB |
| | Diff-UNet | 434.0M | Hybrid | 2.26 | 3.9 GB |

$^\dagger$ The time and average GPU memory for inference were measured with an NVIDIA V100 GPU and an Intel Xeon Silver 4210 CPU, evaluating a CT scan with 259×259×283 voxels and spacing of 1.5 mm/voxel. Measurements consider the entire segmentation pipeline, from loading the CT scan and the AI algorithm, to saving the inference. We observed that the way each AI algorithm deals with spacing and re-shapes its input scan plays a major role in their inference speed.

Table 7: **Training configuration on AbdomenAtlas 1.0.**

| architecture | pre-trained | iterations$^\dagger$ | hours | GPU$^\ddagger$ | GPU memory | hyper-parameter |
|---|---|---|---|---|---|---|
| UniSeg | Yes | 2M | 186 | 1×RTX 3090 | 8.2 GB | Self-configuring |
| MedNeXt | No | 250K | 67 | 4×A100 | 17.6 GB | Manual trial-and-error |
| NexToU | No | 500K | 186 | 1×RTX 3090 | 17.2 GB | Self-configuring |
| STU-Net-B | No | 500K | 30 | 1×A100 | 8.8 GB | Self-configuring |
| U-Net | No | 250k | 7.5 | 1×A100 | 7 GB | Self-configuring |
| ResEncL | No | 250K | 28 | 1×A100 | 24 GB | Self-configuring |
| U-Net & CLIP | No | 200K | 120 | 8×RTX 8000 | 12GB | Self-configuring |
| SwinUNETR & CLIP | No | 200K | 120 | 4×A100 | 48 GB | Self-configuring |
| LHU-Net | No | 250K | 40 | 1×A100 | 8 GB | Pre-defined, from [66, 44] |
| UCTransNet | No | 200K | 20 | 2×A100 | 16 GB | Self-configuring |
| SwinUNETR | Yes | 250k | 24 | 8×V100 | 32 GB | Self-configuring |
| UNesT | No | 250k | 24 | 8×V100 | 16 GB | Self-configuring |
| UNETR | No | 250k | 24 | 8×V100 | 12 GB | Self-configuring |
| SegVol | Yes | 18.75K | 60 | 8×A800 | 50 GB | Manual trial-and-error |
| SAM-Adapter | Yes | 32.5K | 170 | 1×RTX A6000 | 37 GB | Pre-defined, from [23] |
| MedFormer | No | 300K | 72 | 16×V100 | 27.5 GB | Pre-defined, Manual trial-and-error |
| Diff-UNet | No | 500K | 48 | 1×RTX 4090 | 16 GB | Self-configuring |

$^\dagger$ 1 iteration is 1 batch, not a full iteration over all dataset.
$^\ddagger$ GPU: number of GPUs used for training × specific (NVIDIA) GPU.

Table 8: **Additional Training Hyper-parameters.**

| architecture | patch size | batch size | optimizer | learning rate | loss function | WD |
|---|---|---|---|---|---|---|
| UniSeg | [48, 160, 224] | 2 | SGD | 0.01, PolyLRScheduler | Dice, CE | 3.00E-05 |
| MedNeXt | [128, 128, 128] | 8 | AdamW | 1.00E-03 | Dice, CE | 3.00E-05 |
| NexToU | [96, 160, 160] | 2 | SGD | 0.01, PolyLRScheduler | Dice, CE, HTI | 3.00E-05 |
| STU-Net-B | [80, 128, 192] | 2 | SGD | 0.01, PolyLRScheduler | Dice, CE | 3.00E-05 |
| U-Net | [64, 160, 192] | 2 | SGD | 0.01, PolyLRScheduler | Dice, CE | 3.00E-05 |
| ResEncL | [96, 192, 288] | 2 | SGD | 0.01, PolyLRScheduler | Dice, CE | 3.00E-05 |
| U-Net & CLIP | [96,96,96] | 2 | AdamW | 1,00E-4, cosineScheduler | Dice, BCE | 1.00E-05 |
| SwinUNETR & CLIP | [96,96,96] | 2 | AdamW | 1,00E-4, cosineScheduler | Dice, BCE | 1.00E-05 |
| LHU-Net | [96,96,96] | 2 | SGD | 0.01 | Dice, CE | 1,00E-05 |
| UCTransNet | [128,128,128] | 4 | AdamW | 1.00E-04 | Dice, CE | 1.00E-04 |
| SwinUNETR | [96,96,96] | 2 | AdamW | 1.00E-3, cosineScheduler | Dice, CE | 1.00E-05 |
| UNesT | [96,96,96] | 2 | AdamW | 1,00E-3, cosineScheduler | Dice, CE | 1.00E-05 |
| UNETR | [96,96,96] | 2 | AdamW | 1,00E-3, cosineScheduler | Dice, CE | 1.00E-05 |
| SegVol | [4, 16, 16] | 64 | AdamW | 1,00E-04 | Dice, BCE | 1.00E-05 |
| SAM-Adapter | [1, 1024, 1024] | 32 | AdamW | 1.00E-03, warmup | Dice, CE | 0.1 |
| MedFormer | [128,128,128] | 32 | AdamW | 6.00E-4 | Dice, CE | 5.00E-2 |
| Diff-UNet | [128,128,128] | 2 | SGD | 0.01, PolyLRScheduler | CE | 1.00E-03 |

# C Extensive Results on Four Test Datasets

## C.1 NSD scores on the entire JHH dataset

Table 9: **External validation on proprietary JHH dataset ($N$=5,160) - NSD.** For each class, we bold the best-performing results and highlight the runners-up, which show no significant difference ($P > 0.05$) from the best results, in red. Architectures are grouped by their frameworks and sorted in ascending order based on the number of parameters. NSD considers a tolerance of 1.5mm.

| framework | architecture | param | spleen | kidneyR | kidneyL | gallbladder | liver |
|---|---|---|---|---|---|---|---|
| | UniSeg[†] [84] | 31.0M | 88.8±9.7 | 79.8±10.5 | 78.7±9.8 | 75.6±16.8 | 79.5±8.9 |
| | MedNeXt [65] | 61.8M | 88.9±10.3 | 80.0±11.2 | 78.8±10.3 | 75.2±17.5 | 79.0±9.3 |
| | NexToU [67] | 81.9M | 88.2±11.6 | 75.7±13.0 | 75.1±11.7 | 72.2±20.6 | 76.2±10.3 |
| | STU-Net-B [34] | 58.3M | 88.7±10.4 | 80.2±11.1 | 79.3±10.2 | 75.6±16.8 | 78.6±9.4 |
| nnU-Net | STU-Net-L [34] | 440.3M | 89.1±10.1 | 79.7±11.2 | 79.0±10.2 | 76.1±16.9 | 79.0±9.3 |
| | STU-Net-H [34] | 1457.3M | 89.1±10.0 | 80.1±10.9 | 79.2±10.2 | 76.8±16.6 | 79.4±9.3 |
| | U-Net [63] | 31.1M | 88.6±10.5 | 79.9±11.1 | 79.1±10.4 | 73.6±17.9 | 78.1±9.5 |
| | ResEncL [35, 37] | 102.0M | 89.0±10.3 | 80.3±11.0 | 79.1±10.2 | 74.1±18.1 | 78.9±9.5 |
| | ResEncL[★] | 102.0M | 88.6±10.4 | 80.0±11.1 | 78.8±10.3 | 74.0±17.9 | 78.9±9.5 |
| Vision-Language | U-Net & CLIP [47] | 19.1M | 86.5±10.8 | 78.7±10.2 | 78.7±10.4 | 71.4±18.5 | 77.8±8.9 |
| | Swin UNETR & CLIP [47] | 62.2M | 86.0±11.4 | 79.0±11.1 | 78.1±10.7 | 70.2±20.4 | 78.1±9.8 |
| | LHU-Net [66] | 8.6M | 87.1±10.9 | 79.1±10.8 | 78.7±10.1 | 73.0±18.1 | 77.8±9.1 |
| | UCTransNet [73] | 68.0M | 78.7±16.0 | 73.3±15.5 | 73.3±13.5 | 66.0±21.8 | 71.4±11.6 |
| MONAI | Swin UNETR [69] | 72.8M | 80.5±13.4 | 73.7±13.1 | 74.6±12.2 | 62.5±20.6 | 73.7±9.6 |
| | UNesT [86] | 87.2M | 80.7±12.4 | 72.6±12.2 | 72.2±12.1 | 57.8±20.1 | 73.3±9.1 |
| | UNETR [25] | 101.8M | 78.4±15.0 | 73.2±12.3 | 72.8±12.5 | 59.2±21.4 | 73.1±9.6 |
| | SegVol[†] [18] | 181.0M | 86.7±11.1 | 80.2±10.5 | 79.2±9.9 | 68.5±20.7 | 77.9±9.7 |
| | SAM-Adapter[†] [23] | 11.6M | 70.9±15.2 | 70.0±11.6 | 66.2±11.8 | 19.8±11.7 | 62.3±9.7 |
| n/a | MedFormer [19] | 38.5M | **91.3±9.5** | **83.0±10.3** | **80.7±9.7** | **77.3±17.0** | **81.2±9.1** |
| | Diff-UNet [82] | 434.0M | 88.7±10.7 | 81.0±11.0 | 79.5±10.4 | 72.1±18.9 | 78.2±9.5 |

| framework | architecture | param | stomach | aorta | postcava | pancreas | average |
|---|---|---|---|---|---|---|---|
| | UniSeg[†] [84] | 31.0M | 72.4±11.2 | 78.3±13.2 | 70.2±10.8 | 69.9±11.1 | 77.0±11.4 |
| | MedNeXt [65] | 61.8M | 71.5±11.8 | 80.2±12.9 | 70.8±11.0 | 69.3±11.7 | 77.1±11.8 |
| | NexToU [67] | 81.9M | 70.0±12.7 | **83.8±11.9** | 66.2±11.2 | 68.6±14.1 | 75.1±13.0 |
| | STU-Net-B [34] | 58.3M | 70.5±12.1 | 78.3±13.4 | 70.5±10.9 | 69.0±11.5 | 76.8±11.8 |
| nnU-Net | STU-Net-L [34] | 440.3M | 71.7±12.0 | 77.4±13.8 | 70.7±10.9 | 69.7±11.5 | 76.9±11.8 |
| | STU-Net-H [34] | 1457.3M | 72.4±11.9 | 78.0±13.6 | 70.7±10.9 | 69.7±11.5 | 77.3±11.7 |
| | U-Net [63] | 31.1M | 70.1±11.8 | 79.4±13.4 | 70.2±11.0 | 67.4±11.9 | 76.3±11.9 |
| | ResEncL [35, 37] | 102.0M | 70.7±11.8 | 78.2±14.1 | 69.8±11.2 | 68.2±11.5 | 76.5±12.0 |
| | ResEncL[★] | 102.0M | 71.0±11.8 | 81.1±12.5 | 69.7±11.1 | 68.3±11.8 | 76.7±11.8 |
| Vision-Language | U-Net & CLIP [47] | 19.1M | 69.9±11.5 | 74.4±13.9 | 68.0±11.0 | 67.4±12.0 | 74.7±11.9 |
| | Swin UNETR & CLIP [47] | 62.2M | 70.1±12.0 | 75.0±13.6 | 66.2±11.7 | 66.9±12.7 | 74.4±12.6 |
| | LHU-Net [66] | 8.6M | 69.3±11.9 | 75.5±13.3 | 68.1±11.3 | 65.1±11.9 | 74.9±11.9 |
| | UCTransNet [73] | 68.0M | 51.4±13.4 | 82.0±11.5 | 56.3±16.1 | 44.9±18.1 | 66.4±15.3 |
| MONAI | Swin UNETR [69] | 72.8M | 61.6±11.8 | 72.1±15.6 | 60.8±12.6 | 59.2±13.5 | 68.7±13.6 |
| | UNesT [86] | 87.2M | 61.6±11.2 | 71.3±16.0 | 60.4±12.1 | 58.0±11.3 | 67.6±12.9 |
| | UNETR [25] | 101.8M | 53.8±11.8 | 69.2±15.3 | 54.7±12.4 | 54.5±12.8 | 65.4±13.7 |
| | SegVol[†] [18] | 181.0M | 68.2±11.9 | 78.0±13.9 | 66.7±11.4 | 65.9±12.3 | 74.6±12.4 |
| | SAM-Adapter[†] [23] | 11.6M | 48.0±10.5 | 48.8±8.1 | 38.2±9.7 | 22.4±6.2 | 49.6±10.5 |
| n/a | MedFormer [19] | 38.5M | **72.9±12.2** | 82.8±13.4 | **71.8±11.8** | **71.4±12.2** | **79.2±11.7** |
| | Diff-UNet [82] | 434.0M | 68.9±12.0 | 79.3±13.4 | 70.2±11.5 | 66.9±12.3 | 76.1±12.2 |

[†] These architectures were pre-trained (Appendix B.3).
[★] These architectures were trained on AbdomenAtlas 1.0 with enhanced label quality for the aorta class (discussed in §4).

Table 10: **Validation on the TotalSegmentator dataset ($N$=743) - NSD.** For each class, we bold the best-performing results and highlight the runners-up, which show no significant difference ($P > 0.05$) from the best results, in red. Architectures are grouped by their frameworks and sorted in ascending order based on the number of parameters. NSD considers a tolerance of 1.5mm.

| framework | architecture | param | spleen | kidneyR | kidneyL | gallbladder | liver |
|---|---|---|---|---|---|---|---|
| | UniSeg[†] [84] | 31.0M | 87.1±21.6 | 81.1±24.7 | 78.9±27.3 | 73.2±29.4 | 83.5±19.9 |
| | MedNeXt [65] | 61.8M | 90.1±20.1 | 82.4±24.8 | 82.8±24.2 | 74.9±29.1 | 86.7±18.3 |
| | NexToU [67] | 81.9M | 79.7±30.6 | 74.1±31.5 | 74.6±29.9 | 70.4±31.5 | 78.5±24.9 |
| | STU-Net-B [34] | 58.3M | 90.6±17.8 | 83.4±21.5 | 83.3±23.0 | **77.5±25.3** | 85.4±18.8 |
| nnU-Net | STU-Net-L [34] | 440.3M | 90.0±20.0 | 84.4±20.4 | 83.0±23.6 | 76.7±25.3 | **87.9±15.3** |
| | STU-Net-H [34] | 1457.3M | **90.6±17.0** | 85.1±18.5 | 82.9±24.3 | 76.5±25.6 | 87.2±16.4 |
| | U-Net [63] | 31.1M | 89.6±19.4 | 84.4±19.3 | 83.9±21.7 | 77.5±26.0 | 86.6±15.9 |
| | ResEncL [35, 37] | 102.0M | 90.4±19.1 | **85.6±19.2** | **85.2±21.1** | 76.6±26.6 | 85.1±20.0 |
| | ResEncL[★] | 102.0M | 90.4±18.7 | 86.6±17.2 | 86.6±19.2 | 76.6±25.9 | 86.0±18.6 |
| Vision-Language | U-Net & CLIP [47] | 19.1M | 84.3±26.0 | 79.8±25.4 | 78.9±25.9 | 71.5±29.0 | 81.9±18.6 |
| | Swin UNETR & CLIP [47] | 62.2M | 83.2±25.6 | 78.0±28.3 | 74.2±31.3 | 68.8±31.4 | 82.0±19.7 |
| | LHU-Net [66] | 8.6M | 82.2±28.3 | 77.4±28.9 | 78.0±27.3 | 69.8±32.2 | 77.9±27.0 |
| | UCTransNet [73] | 68.0M | 72.2±35.3 | 71.1±34.3 | 59.6±39.7 | 67.3±32.1 | 71.3±29.5 |
| MONAI | Swin UNETR [69] | 72.8M | 58.9±35.3 | 53.2±36.9 | 53.1±37.4 | 46.3±38.6 | 65.1±27.1 |
| | UNesT [86] | 87.2M | 71.7±27.5 | 69.5±30.8 | 66.7±32.6 | 45.7±38.4 | 75.8±20.8 |
| | UNETR [25] | 101.8M | 48.8±34.9 | 40.1±35.0 | 35.5±35.3 | 32.9±32.1 | 58.0±25.3 |
| | SegVol[†] [18] | 181.0M | 83.2±24.1 | 77.2±23.1 | 76.6±25.1 | 63.3±28.8 | 79.0±21.5 |
| | SAM-Adapter[†] [23] | 11.6M | 36.7±25.2 | 8.8±9.8 | 24.3±19.7 | 6.4±10.5 | 40.8±26.1 |
| n/a | MedFormer [19] | 38.5M | 86.5±19.0 | 79.7±20.6 | 79.2±23.0 | 71.0±27.4 | 83.0±17.2 |
| | Diff-UNet [82] | 434.0M | 85.4±25.9 | 76.5±27.5 | 76.2±28.3 | 68.9±31.5 | 84.7±17.5 |

| framework | architecture | param | stomach | aorta | IVC[‡] | pancreas | average |
|---|---|---|---|---|---|---|---|
| | UniSeg[†] [84] | 31.0M | 64.0±31.1 | 67.3±31.9 | 68.3±26.7 | 67.8±30.8 | 74.6±27.1 |
| | MedNeXt [65] | 61.8M | 67.1±30.9 | 69.5±31.0 | 70.0±24.8 | 68.6±31.2 | 76.9±26.1 |
| | NexToU [67] | 81.9M | 58.6±34.2 | 59.5±32.9 | 54.0±31.3 | 62.1±31.6 | 68.0±31.0 |
| | STU-Net-B [34] | 58.3M | 68.1±30.2 | 71.8±30.0 | 71.8±22.1 | 72.0±27.3 | 78.2±24.0 |
| nnU-Net | STU-Net-L [34] | 440.3M | 69.2±28.6 | **74.0±27.5** | **72.0±21.2** | 72.9±26.9 | **78.9±23.2** |
| | STU-Net-H [34] | 1457.3M | 68.4±28.8 | 72.7±28.6 | 71.5±21.2 | 71.9±27.8 | 78.5±23.2 |
| | U-Net [63] | 31.1M | 68.6±28.6 | 68.4±28.5 | 71.0±24.0 | 72.1±27.4 | 78.0±23.4 |
| | ResEncL [35, 37] | 102.0M | 68.7±28.8 | 71.3±26.6 | 70.9±22.2 | **73.5±26.6** | 78.6±23.3 |
| | ResEncL[★] | 102.0M | 70.1±27.1 | 81.9±22.0 | 71.0±21.9 | 74.5±25.6 | 80.4±21.8 |
| Vision-Language | U-Net & CLIP [47] | 19.1M | 66.7±28.2 | 57.5±32.2 | 61.6±26.8 | 70.6±26.1 | 72.5±26.5 |
| | Swin UNETR & CLIP [47] | 62.2M | 58.9±31.3 | 56.6±34.0 | 58.9±26.9 | 66.2±28.8 | 69.6±28.6 |
| | LHU-Net [66] | 8.6M | 60.5±32.5 | 59.2±33.5 | 62.6±27.9 | 65.4±32.0 | 70.3±30.0 |
| | UCTransNet [73] | 68.0M | 48.1±34.2 | 48.1±33.8 | 45.2±34.8 | 54.4±33.7 | 59.7±34.1 |
| MONAI | Swin UNETR [69] | 72.8M | 37.1±29.4 | 51.2±36.1 | 31.6±30.8 | 35.1±30.4 | 48.0±33.6 |
| | UNesT [86] | 87.2M | 48.8±28.7 | 51.6±35.5 | 34.0±32.9 | 42.8±28.8 | 56.3±30.7 |
| | UNETR [25] | 101.8M | 25.3±22.7 | 36.8±28.8 | 32.4±27.3 | 21.2±22.8 | 36.8±29.3 |
| | SegVol[†] [18] | 181.0M | 58.7±28.7 | 57.6±28.8 | 56.1±23.5 | 59.9±26.5 | 68.0±25.6 |
| | SAM-Adapter[†] [23] | 11.6M | 27.0±19.6 | 17.1±17.2 | 5.4±8.1 | 21.7±14.4 | 20.9±16.7 |
| n/a | MedFormer [19] | 38.5M | **69.3±26.7** | 67.9±29.2 | 65.5±25.5 | 69.0±27.8 | 74.6±24.0 |
| | Diff-UNet [82] | 434.0M | 59.8±31.2 | 57.7±34.6 | 55.4±33.4 | 65.5±29.4 | 70.0±28.8 |

[†] These architectures were pre-trained (Appendix B.3).

[‡] The class IVC (inferior vena cava) shares the same meaning as the class postcava in other datasets (e.g., AbdomenAtlas 1.0 and JHH).

[★] These architectures were trained on AbdomenAtlas 1.0 with enhanced label quality for the aorta class (discussed in §4).

Table 11: **Validation on the official test set of TotalSegmentator (N=59) - DSC.** TotalSegmentator provides an official split of training and testing sets. To align with other papers, we hereby also provide the benchmark results on the test set of TotalSegmentator (N=59). Notably, the average scores in the official test set are usually higher than the ones in the entire TotalSegmentator dataset.

| framework | architecture | param | spleen | kidneyR | kidneyL | gallbladder | liver |
|---|---|---|---|---|---|---|---|
| | UniSeg[†] [84] | 31.0M | 94.7±6.8 | 86.5±17.8 | 88.2±13.3 | 78.0±27.8 | 96.2±2.4 |
| | MedNeXt [65] | 61.8M | 93.5±12.0 | 83.6±24.8 | 89.7±14.8 | 73.1±34.7 | 96.8±2.3 |
| | NexToU [67] | 81.9M | 90.0±22.8 | 82.1±26.2 | 79.4±26.4 | 76.2±32.8 | 90.8±18.5 |
| | STU-Net-B [34] | 58.3M | **96.5±2.6** | 86.8±18.3 | 90.2±9.7 | 78.4±30.9 | 96.4±4.9 |
| nnU-Net | STU-Net-L [34] | 440.3M | 96.1±3.4 | 85.2±22.0 | 89.4±14.5 | 82.0±24.6 | 96.8±2.6 |
| | STU-Net-H [34] | 1457.3M | 96.3±3.2 | 85.7±19.9 | **92.5±5.6** | **84.4±22.2** | **97.2±1.6** |
| | U-Net [63] | 31.1M | 94.9±12.3 | **88.3±18.1** | 88.6±12.3 | 78.3±29.7 | 95.7±5.8 |
| | ResEncL [35, 37] | 102.0M | 94.7±12.3 | 84.9±23.5 | 90.7±11.0 | 78.4±29.7 | 95.7±8.2 |
| | ResEncL[★] | 102.0M | 95.6±8.8 | 87.0±20.7 | 91.6±10.3 | 78.0±29.0 | 96.7±2.7 |
| Vision-Language | U-Net & CLIP [47] | 19.1M | 94.6±7.0 | 85.2±22.5 | 83.1±24.0 | 70.1±33.9 | 95.3±4.6 |
| | Swin UNETR & CLIP [47] | 62.2M | 92.5±10.1 | 76.7±34.6 | 73.4±34.8 | 72.2±34.2 | 96.2±2.8 |
| | LHU-Net [66] | 8.6M | 92.3±15.5 | 84.9±21.4 | 89.5±10.6 | 74.8±33.3 | 94.2±10.0 |
| | UCTransNet [73] | 68.0M | 89.3±19.4 | 82.7±27.6 | 59.3±41.7 | 70.3±32.5 | 92.8±15.9 |
| MONAI | Swin UNETR [69] | 72.8M | 80.8±28.9 | 69.9±35.7 | 57.7±40.2 | 47.4±44.1 | 89.8±16.5 |
| | UNesT [86] | 87.2M | 90.2±11.3 | 79.0±26.7 | 70.4±34.6 | 49.7±40.2 | 95.0±3.3 |
| | UNETR [25] | 101.8M | 74.4±31.3 | 60.0±37.1 | 47.5±39.7 | 40.1±40.2 | 84.6±23.9 |
| | SegVol[†] [18] | 181.0M | 91.2±16.7 | 82.1±21.2 | 82.5±21.9 | 69.9±30.8 | 94.8±5.6 |
| | SAM-Adapter[†] [23] | 11.6M | 50.4±34.1 | 9.2±10.5 | 18.0±21.2 | 7.2±12.3 | 77.5±21.3 |
| n/a | MedFormer [19] | 38.5M | 95.4±1.7 | 84.0±22.5 | 89.2±9.3 | 76.5±28.5 | 96.2±2.7 |
| | Diff-UNet [82] | 434.0M | 95.3±6.3 | 85.0±22.9 | 86.7±16.9 | 72.3±34.5 | 93.6±15.9 |

| framework | architecture | param | stomach | aorta | IVC[‡] | pancreas | average |
|---|---|---|---|---|---|---|---|
| | UniSeg[†] [84] | 31.0M | 80.8±27.3 | 82.6±19.7 | 79.5±20.1 | 82.1±17.2 | 85.4±16.9 |
| | MedNeXt [65] | 61.8M | **87.8±13.3** | 84.9±17.2 | 82.2±16.0 | 83.9±16.8 | 86.2±16.9 |
| | NexToU [67] | 81.9M | 82.4±25.8 | 72.5±27.1 | 66.4±30.2 | 78.9±19.2 | 79.9±25.4 |
| | STU-Net-B [34] | 58.3M | 86.1±20.1 | 85.5±16.3 | 82.1±17.3 | **84.1±15.9** | 87.3±15.1 |
| nnU-Net | STU-Net-L [34] | 440.3M | 88.7±14.2 | **87.0±11.2** | **84.5±8.9** | 83.4±17.2 | 88.1±13.2 |
| | STU-Net-H [34] | 1457.3M | 88.4±14.2 | 86.7±11.1 | 84.0±9.7 | 82.9±17.5 | **88.7±11.7** |
| | U-Net [63] | 31.1M | 85.7±21.1 | 82.6±18.8 | 79.7±20.4 | 83.1±16.0 | 86.3±17.2 |
| | ResEncL [35, 37] | 102.0M | 85.4±21.1 | 83.7±17.6 | 79.0±20.4 | 83.4±16.7 | 86.2±17.8 |
| | ResEncL[★] | 102.0M | 86.9±17.8 | 91.1±8.8 | 80.6±16.2 | 83.8±16.3 | 87.9±14.5 |
| Vision-Language | U-Net & CLIP [47] | 19.1M | 84.0±19.1 | 70.7±28.7 | 77.0±20.4 | 79.8±21.7 | 82.2±20.2 |
| | Swin UNETR & CLIP [47] | 62.2M | 79.9±25.7 | 72.3±27.7 | 72.9±21.9 | 77.6±21.8 | 79.3±23.7 |
| | LHU-Net [66] | 8.6M | 80.5±26.4 | 72.2±29.9 | 73.6±24.5 | 80.0±21.9 | 82.5±21.5 |
| | UCTransNet [73] | 68.0M | 74.4±31.8 | 61.7±32.8 | 63.7±32.6 | 76.0±18.1 | 74.5±28.0 |
| MONAI | Swin UNETR [69] | 72.8M | 55.1±36.8 | 69.9±27.5 | 52.7±32.0 | 57.2±32.8 | 64.5±32.7 |
| | UNesT [86] | 87.2M | 70.4±30.0 | 65.0±33.9 | 53.2±33.8 | 65.2±25.7 | 70.9±26.6 |
| | UNETR [25] | 101.8M | 52.6±31.0 | 50.4±30.2 | 52.7±28.8 | 45.1±30.8 | 56.4±32.6 |
| | SegVol[†] [18] | 181.0M | 78.5±26.5 | 74.4±21.8 | 69.9±19.5 | 76.0±16.9 | 79.9±20.1 |
| | SAM-Adapter[†] [23] | 11.6M | 48.7±32.9 | 25.1±23.3 | 7.0±8.6 | 37.7±20.0 | 31.2±20.5 |
| n/a | MedFormer [19] | 38.5M | 87.8±13.9 | 83.9±15.8 | 79.6±10.5 | 81.2±18.5 | 86.0±13.7 |
| | Diff-UNet [82] | 434.0M | 82.0±25.0 | 74.4±26.8 | 73.6±27.4 | 79.0±21.4 | 82.4±21.9 |

[†] These architectures were pre-trained (Appendix B.3).

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

# D    Additional Analysis of Benchmark Results

## D.1    Worst-case Analysis

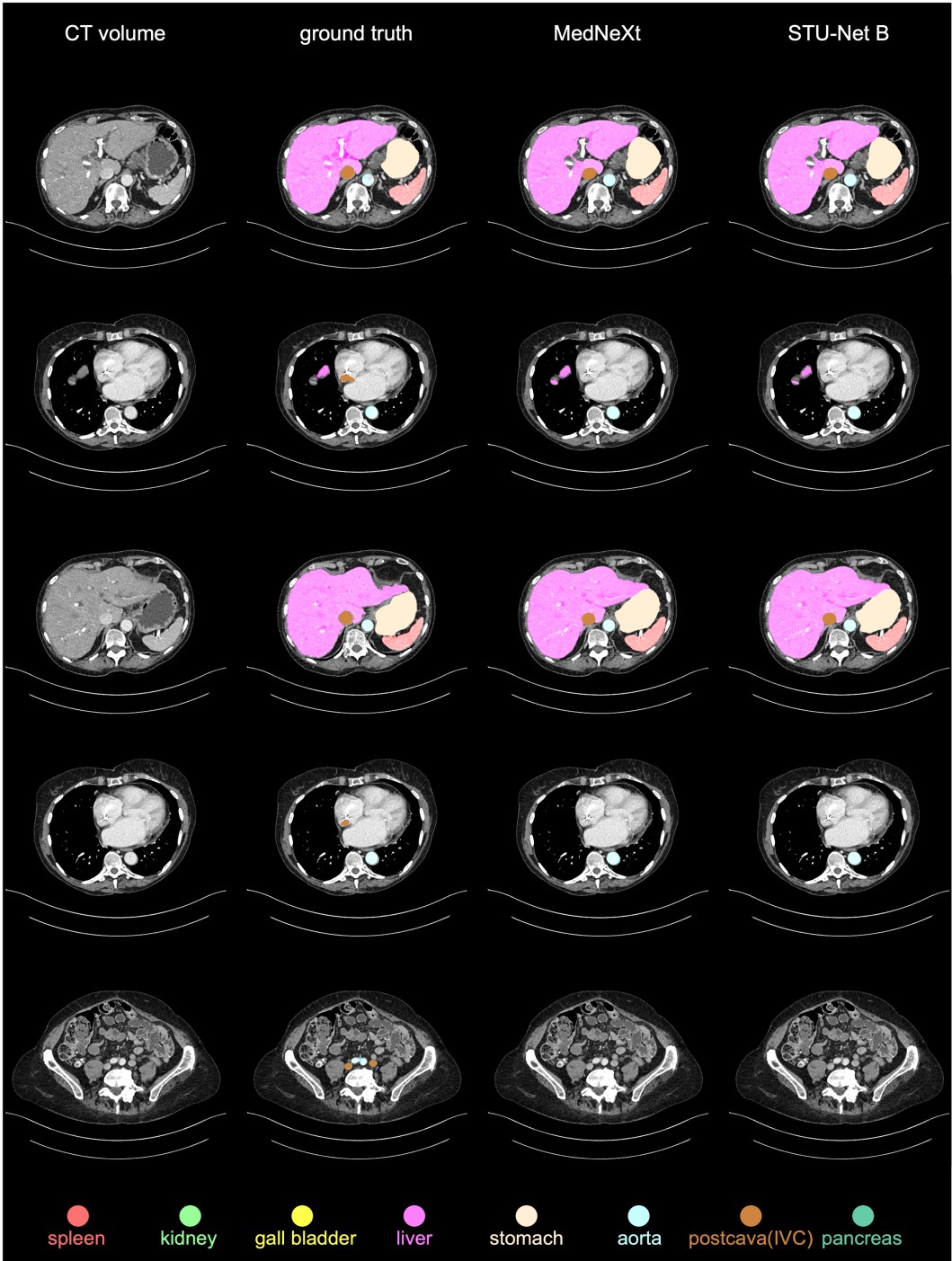

Figure 5: **Worst case analysis for JHH.** This figure displays CT scans that are particularly challenging for most AI algorithms to identify. To illustrate these difficult cases, we also include visualizations from the top-performing algorithm, MedNext, and the first runner-up, STU-Net Base.

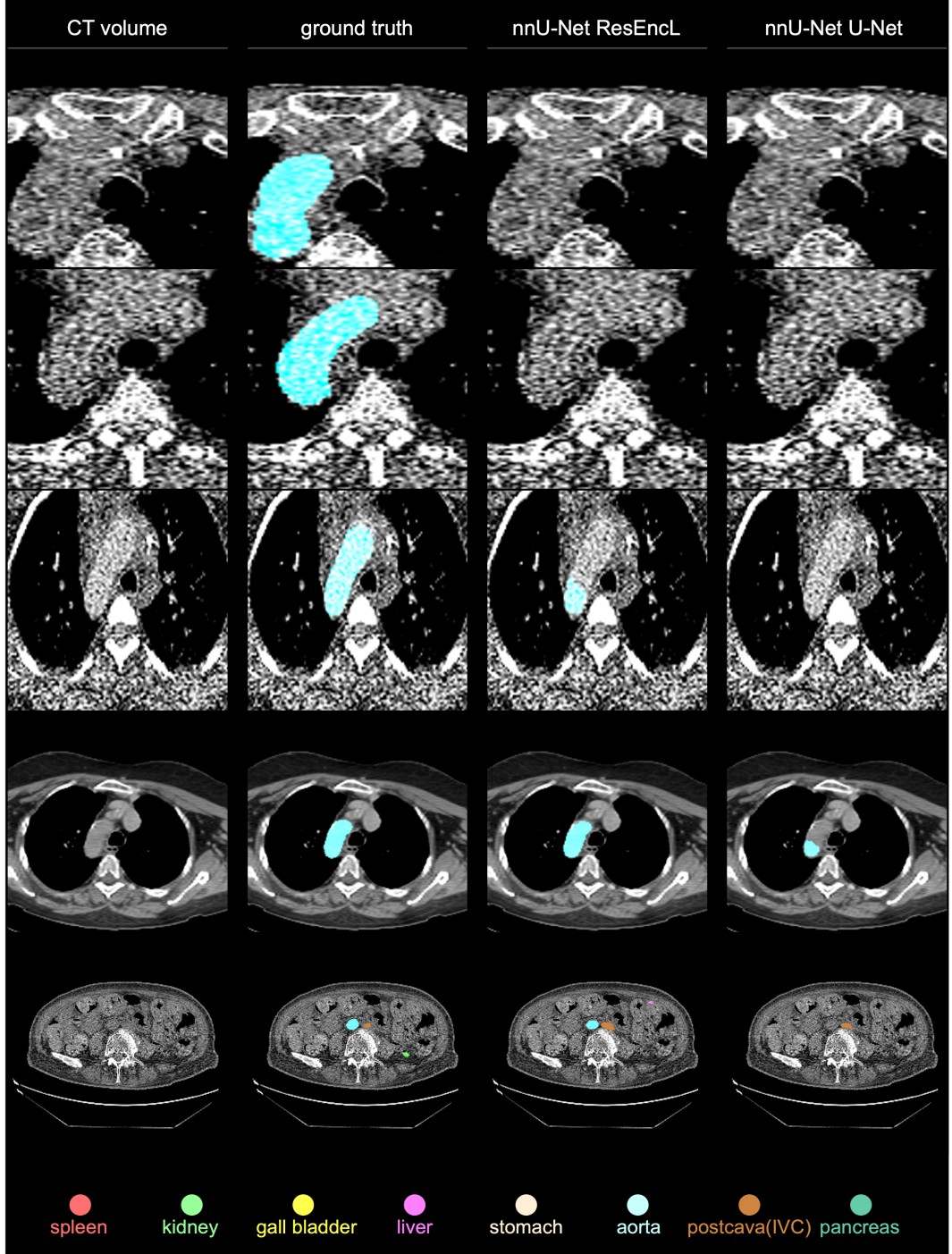

Figure 6: **Worst case analysis for TotalSegmentator.** This figure displays CT scans that are particularly challenging for most AI algorithms to identify. To illustrate these difficult cases, we also include visualizations from the top-performing algorithm, ResEncL, and the first runner-up, U-Net. The results show ResEncL does perform better than U-Net in these worst cases.

### D.2 Ranking Stability Analyses

#### D.2.1 Evaluation Metrics

Every evaluation metric reflects a certain aspect of the results and choosing the right one is important to emphasize those properties that we care about. In this section, we assess the ranking stability with respect to different evaluation metrics.

The Dice Similarity Coefficient (DSC) is a widely used metric in medical imaging to measure the overlap between the prediction and the ground truth. Additionally, Normalized Surface Distance (NSD) focuses on the segmentation quality between two boundaries.

Due to the existence of NaNs (which represent some organs that are missing in some CT scans), averaging per-case-per-class values by case first and then by class differs from averaging them by class first and then by case [75]. Let's focus on DSC (note that this also applies to other metrics such as NSD) and denote the first version as $DSC^C$ and the second as $DSC^I$. $DSC^C$ allows us to evaluate model performance on a class-wise scale, emphasizing difficult classes, and it alleviates the limitation of $DSC^I$, which can be biased towards classes with less NaNs. On the other hand, $DSC^I$ facilitates statistical tests across different cases. Due to these considerations, we use $DSC^C$ for reporting per-class performance and utilize $DSC^I$ to conduct statistical tests. In the rest of the paper, we drop the superscripts for simplicity unless stated otherwise.

Besides the standard DSC, in this section, we also consider a worst-case metric to emphasize difficult cases [75]. In particular, it only averages over cases whose scores fall below the 10% quantile.

Except accuracy metrics such DSC and NSD, we also study bias metrics. Specifically, we choose Demographic Parity Difference (DPD) [1, 70], which captures bias across diverse demographic groups. Originally proposed for classification problems, we extend it to medical segmentation and define it as the maximum differences in DSC among different sensitive demographic groups.

The results for different metrics are shown in Figures 7-9. We find that models tend to retain a similar rank across different accuracy metrics, indicating that these models do not overfit to a specific metric. However, performance on the worst-case $DSC^C$ is significantly lower than the $DSC^C$ itself, showing that a need for improvements in model performance on these hard cases, or indicating the existence of some label noise in test sets. We visualize some worse-case examples in Appendix D.1. Regarding the bias metrics, although there are some variations in rankings, we find models with high accuracy usually have low bias.

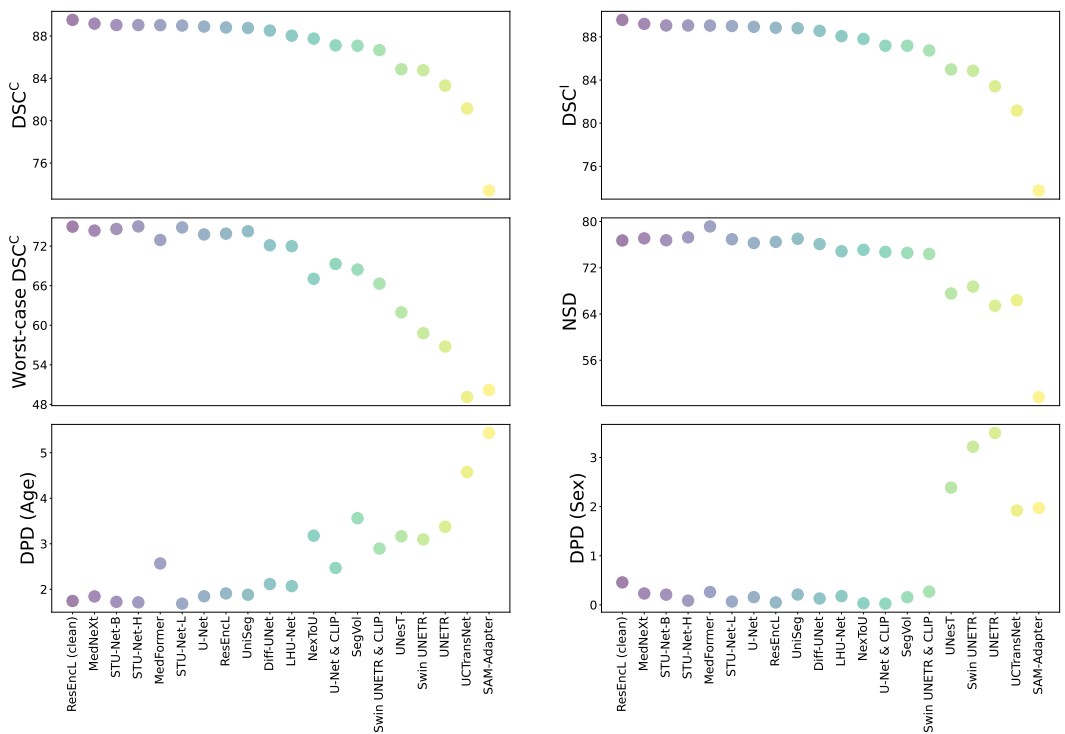

Figure 7: **Comparison of different evaluation metrics on proprietary JHH dataset.** Models tend to retain a similar rank across different metrics.

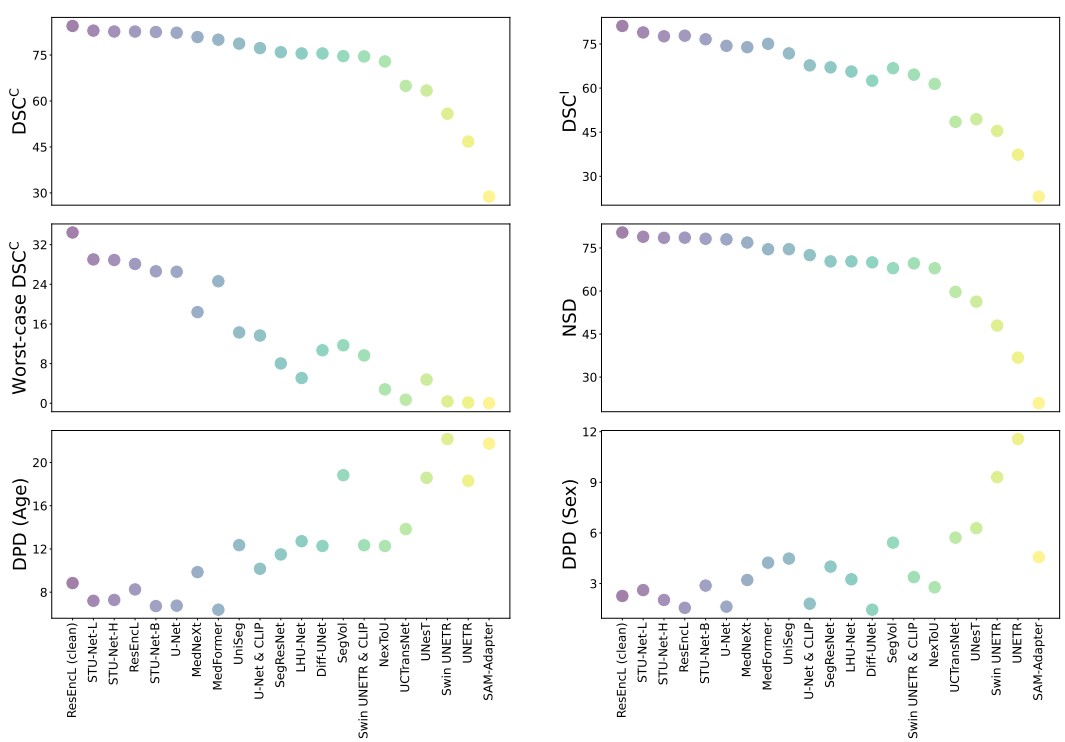

Figure 8: **Comparison of different evaluation metrics on TotalSegmentator.** Models tend to retain a similar rank across different metrics.

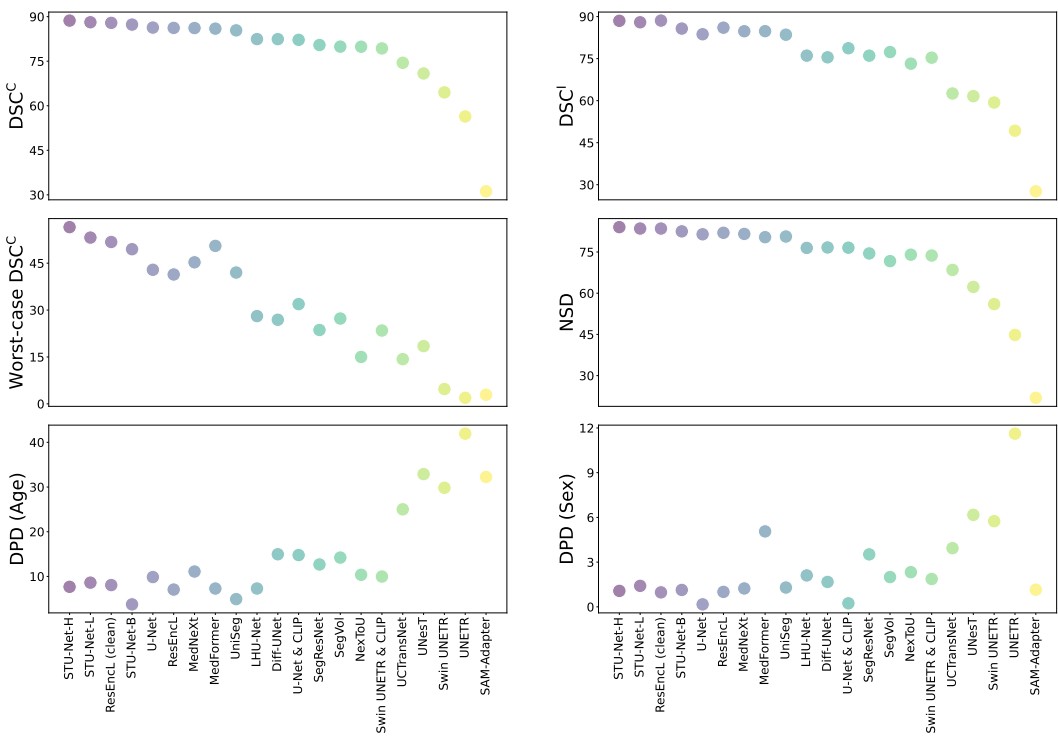

Figure 9: **Comparison of different evaluation metrics on the official test set of TotalSegmentator.** Models tend to retain a similar rank across different metrics.

### D.2.2 Bootstrap Sampling

To evaluate ranking stability, we perform bootstrap sampling as described in [78]. A bootstrap sample of a dataset with $n$ test cases consists of $n$ test cases randomly drawn from the dataset with replacement. A total of 1,000 bootstrap samples are drawn, and the results are visualized as blob plots in Figure 10.

Our findings indicate that datasets with more test cases tend to present fewer variations in ranks. For example, we find fewer variations in ranks on the entire TotalSegmentator ($N = 743$) compared with the ranks on the TotalSegmentator official test set ($N = 59$). On the proprietary JHH dataset ($N = 5,160$), we observe minimum ranking variations due to its large number of test cases. Additionally, the ranks are relatively robust for the highest- and lowest-performing models, but they can be more unstable for models in the middle range.

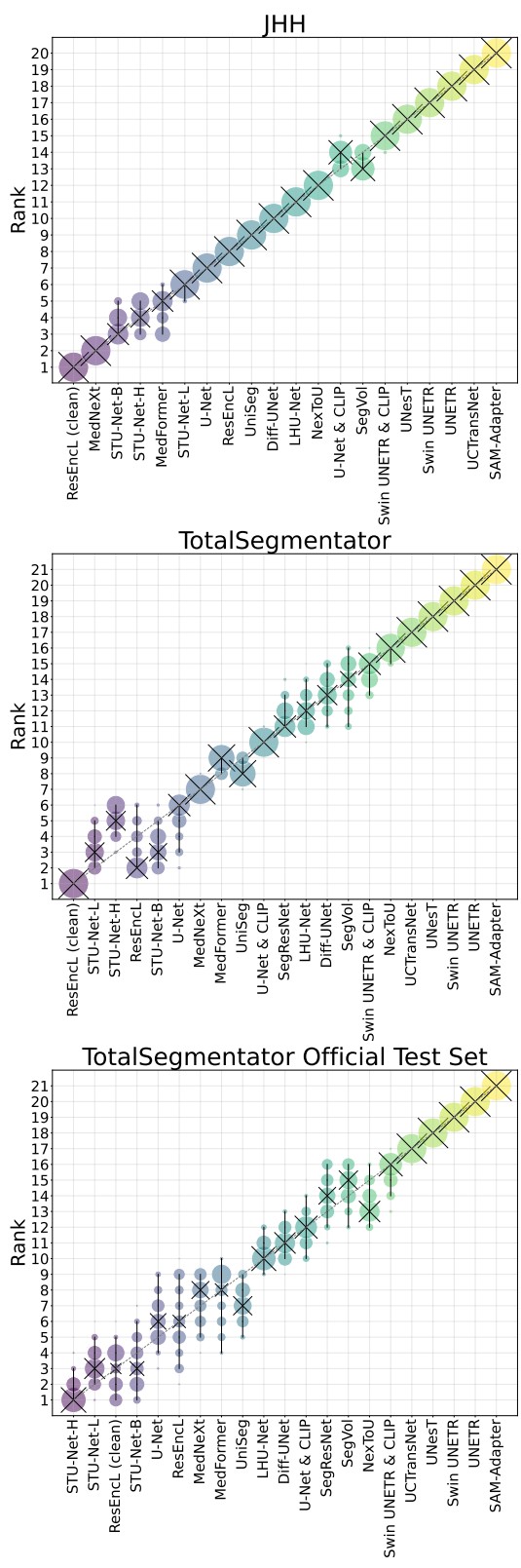

Figure 10: **Blob plots for visualizing ranking stability based on 1,000 bootstrap samples.** The area of each blob is proportional to the relative frequency. The median rank for each model is marked by a black cross. 95% bootstrap intervals (ranging from the 2.5th to the 97.5th percentile of the bootstrap distribution) are connected by black lines. We observe more stable rankings for larger tests sets.

### D.2.3   Significance Maps

To further investigate ranking stability, we performed pair-wise comparisons between each possible pair of algorithms. Comparisons use statistical tests to understand if an algorithm's scores are significantly better than the other model's results. We employed one-sided Wilcoxon signed rank tests with Holm's adjustment and 5% significance level.

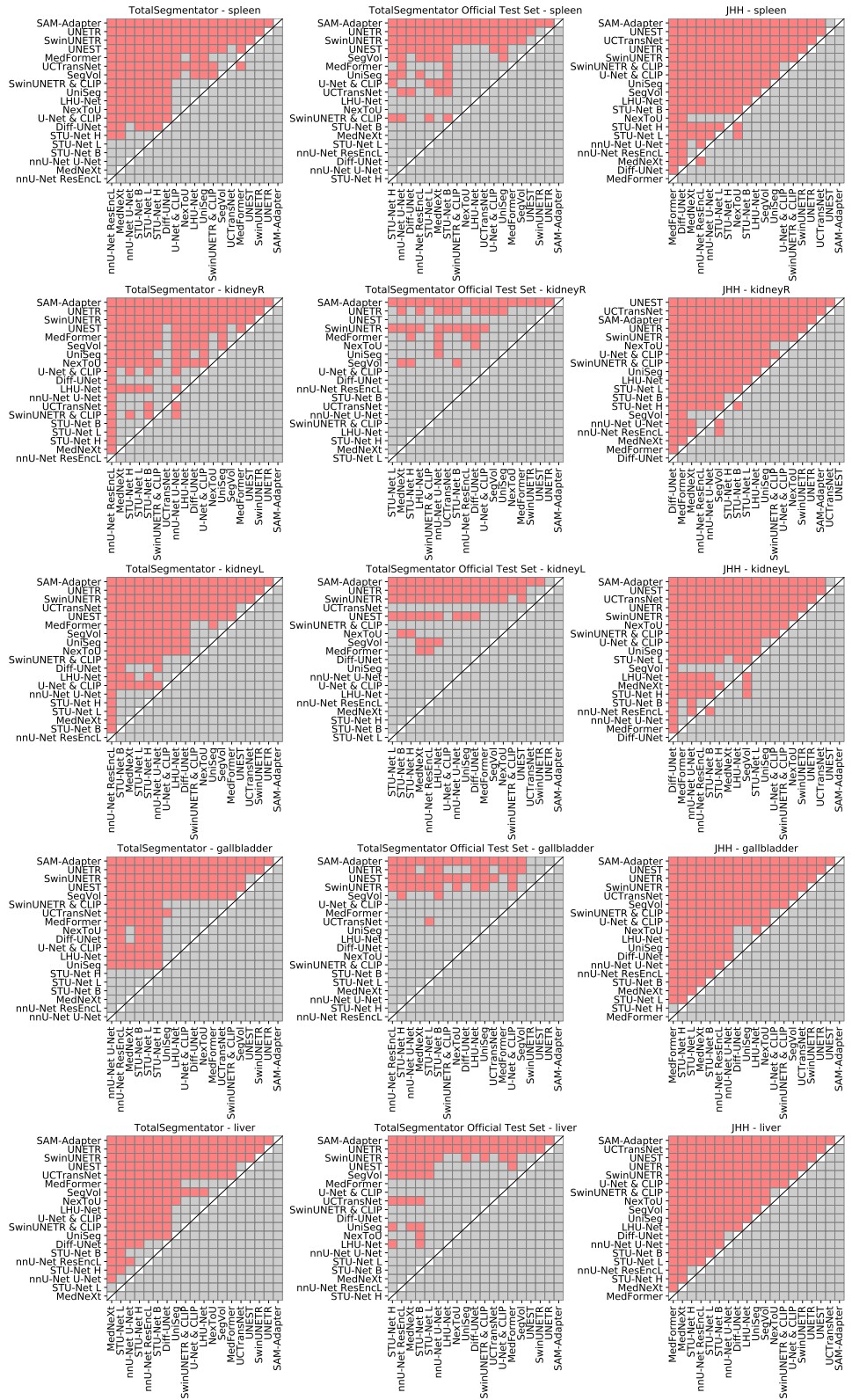

Figure 11: **DSC significance maps.** Each cell represents a pair-wise comparison between two algorithms, according to DSC score. Yellow colors indicate that the x-axis AI algorithm is significantly superior to the y-axis algorithm in terms of DSC score (considering all organs). Blue represents no significant superiority. Comparisons employed one-sided Wilcoxon signed rank tests with Holm's adjustment and 5% significance level.

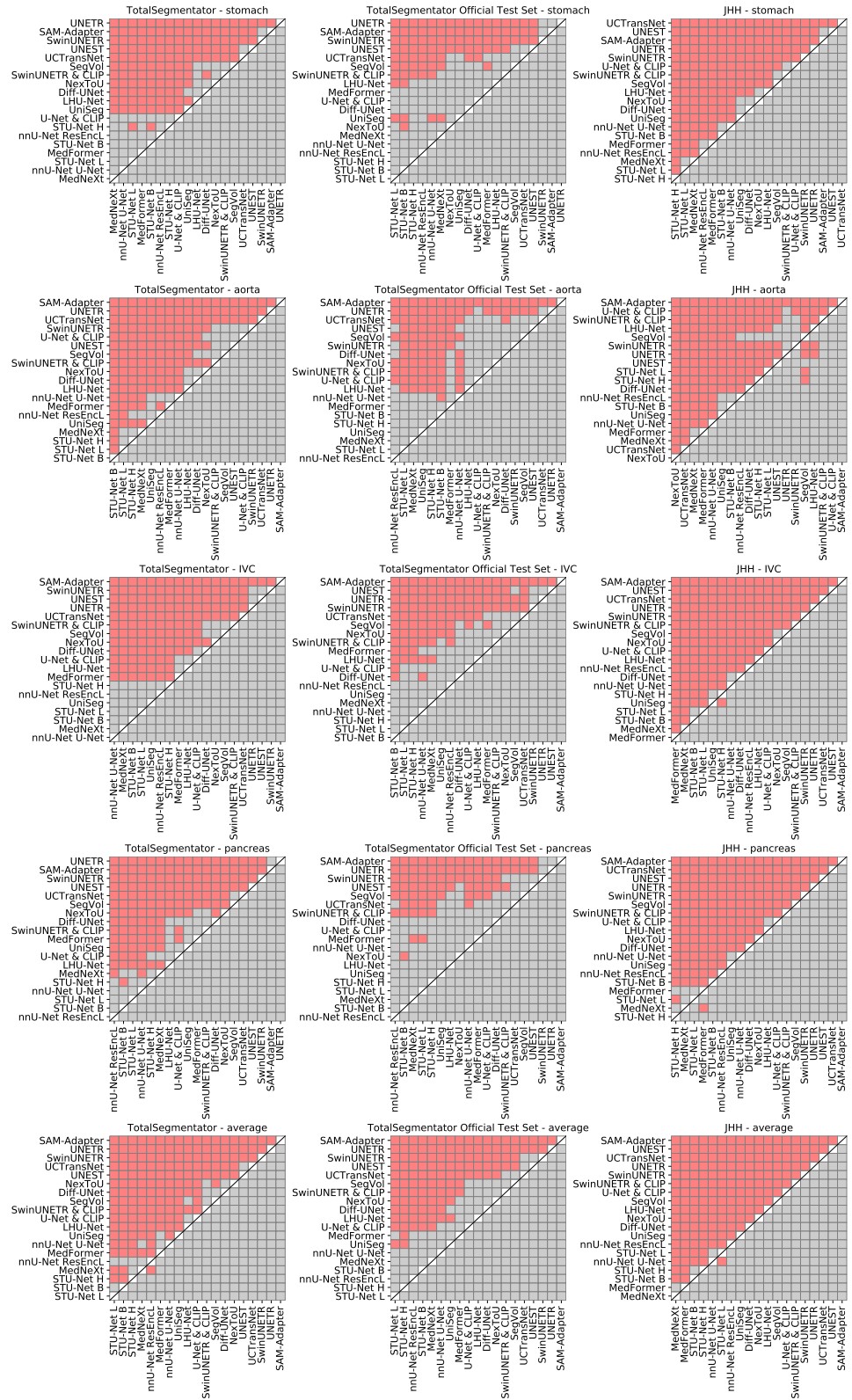

Figure 12: **Continuation of DSC significance maps.** Each cell represents a pair-wise comparison between two algorithms, according to DSC score. Yellow colors indicate that the x-axis AI algorithm is significantly superior to the y-axis algorithm in terms of DSC score (considering all organs). Blue represents no significant superiority. Comparisons employed one-sided Wilcoxon signed rank tests with Holm's adjustment and 5% significance level.

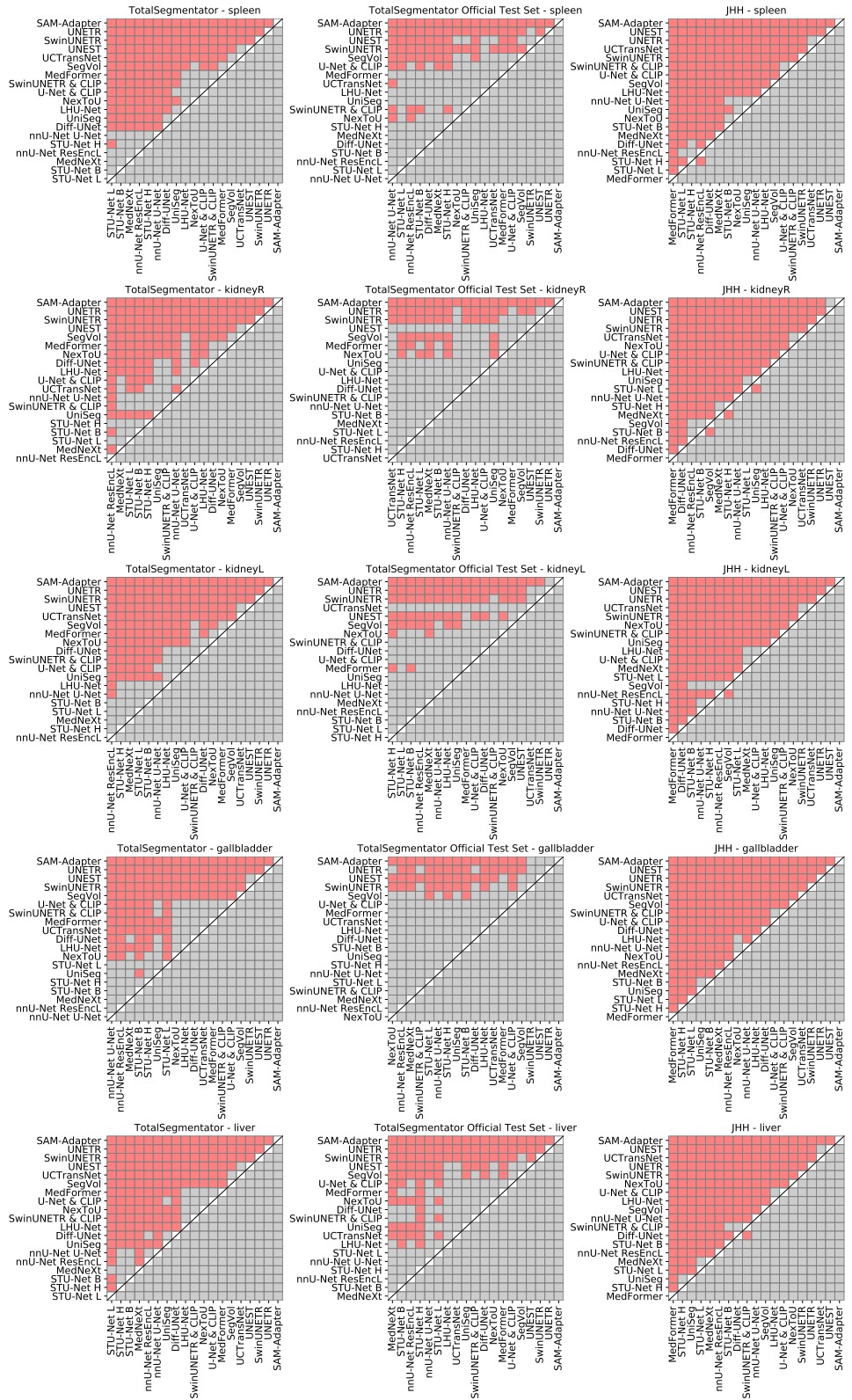

Figure 13: **NSD significance maps.** Each cell represents a pair-wise comparison between two algorithms, according to NSD. Yellow colors indicate that the x-axis AI algorithm is significantly superior to the y-axis algorithm in terms of NSD score (considering all organs). Blue represents no significant superiority. Comparisons employed one-sided Wilcoxon signed rank tests with Holm's adjustment and 5% significance level. NSD considers a threshold of 1.5mm.

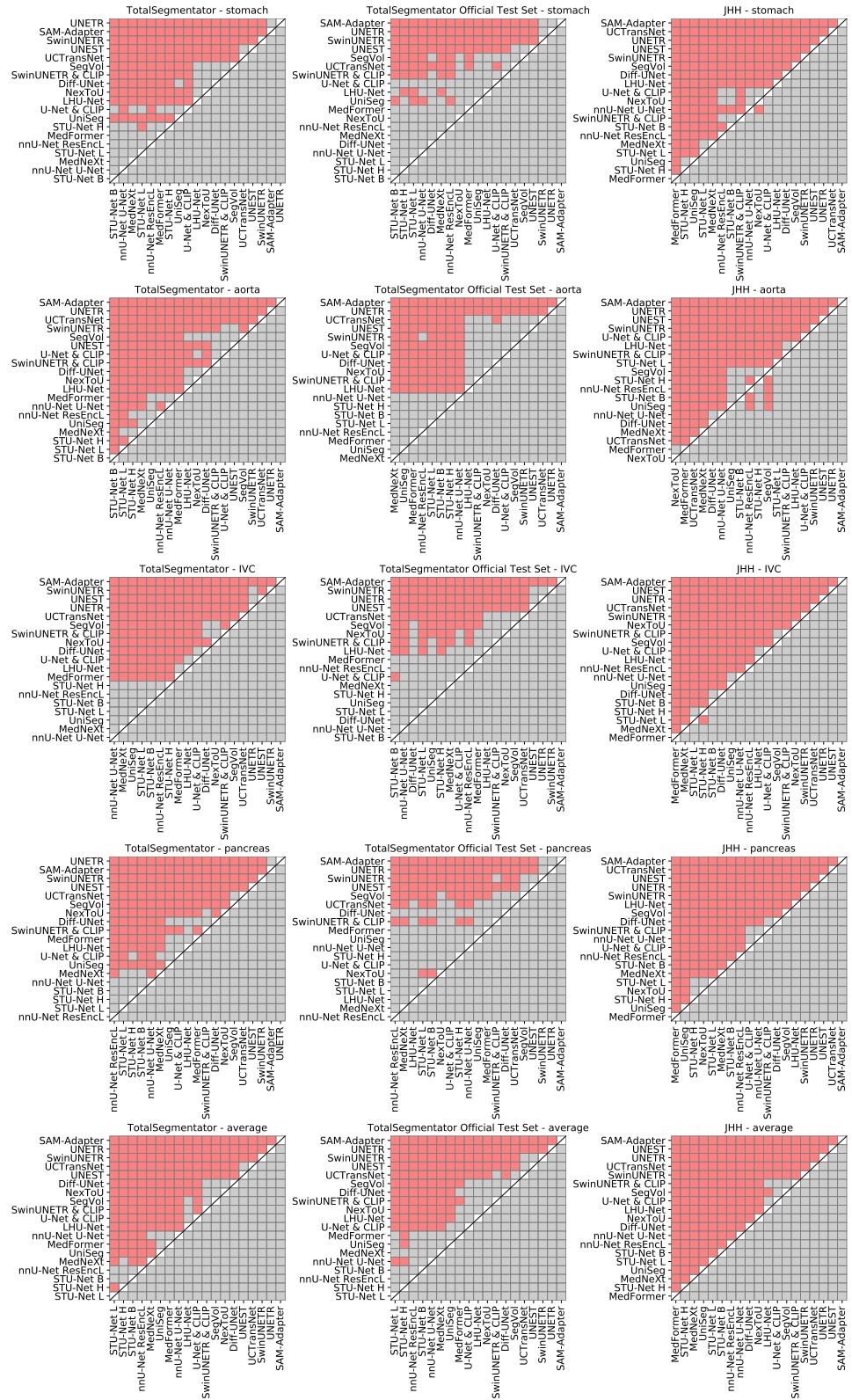

Figure 14: **Continuation of NSD significance maps.** Each cell represents a pair-wise comparison between two algorithms, according to NSD. Yellow colors indicate that the x-axis AI algorithm is significantly superior to the y-axis algorithm in terms of NSD score (considering all organs). Blue represents no significant superiority. Comparisons employed one-sided Wilcoxon signed rank tests with Holm's adjustment and 5% significance level. NSD considers a threshold of 1.5mm.

### D.3 Per-Algorithm Analysis

#### D.3.1 MedNeXt

The central theme of the ConvNeXt [50] architecture was decoupling the scalability of the Transformer architecture and using it in a convolutional fashion, without self-attention. Scalability becomes relevant for medical images when creating large 3D networks while not overfitting. MedNeXt [65] builds upon this principle by using these blocks across the network, leading to the performance seen in this work.

#### D.3.2 STU-Net

The STU-Net [34] is built upon the nnU-Net framework, which was proven effective in our experiments. Additionally, STU-Net is based on scaling the AI model size, which may be exceptionally useful for dealing with large-scale datasets like AbdomenAtlas 1.0. The combination of a high-performance framework and an appropriately scaled model may be the key for STU-Net's high segmentation accuracy in this study.

#### D.3.3 NexToU

NexToU [67] is a hybrid architecture that combines a hierarchical 3D U-shaped encoder-decoder structure with both Convolutional Neural Networks (CNNs) and Graph Neural Networks (GNNs). This innovative approach employs a hierarchical, topology-aware strategy inspired by human cognitive processes, allowing the model to progressively decompose anatomical semantics from simpler to more complex structures. On the JHH dataset, NexToU's results were relatively close to the best-performing models. However, we observed a significant performance difference on the TotalSegmentator dataset. This discrepancy is likely due to our model not utilizing a resampling step to the average spatial resolution during inference for data with fewer slices along the z-axis. While this approach saves inference time, it compromises performance on data with low z-axis resolution. Additionally, to further reduce inference time, Test Time Augmentation (TTA) was minimized, leading to a decline in performance for bilaterally symmetric classes like kidneyR and kidneyL, as well as for some small sample classes.

#### D.3.4 DiffU-Net

We hypothesize that two main factors contributed to Diff-UNet's [82] high segmentation accuracy: its nnU-Net-inspired hyper-parameter selection procedure and the use of stable diffusion. The diffusion model excels in handling details, generating high-resolution images when used as a generative model. During inference, the model predicts multiple times using the DDIM sampling strategy, further enhancing Diff-UNet's outputs. Moreover, considering that the diffusion model includes noised information, DiffU-Net has a boundary branch, which takes the 3D medical image as input. This branch supplies clear image information to complement the diffusion branch, further improving segmentation accuracy.

#### D.3.5 SAM-Adapter

We observed a lower performance for the fine-tuned Segment Anything model, which we hypothesize may be due to the following reasons:

- The SAM-based model is a 2D-based model that performs multi-class segmentation solely on 2D slices. This approach relies mainly on 2D information, such as location relations, rather than 3D organ shape information. When tested on out-of-distribution (OOD) sets, images from different hospitals may introduce spatial variations and voxel spacing, leading to varying spatial distributions of abdomen regions compared to the training images. These spatial changes can cause the 2D-based model to lose its segmentation accuracy.

- During the training of this fine-tuned model, no spatial transformations for augmentation were used, which might have been used in other comparison methods. This lack of augmentation could lead to poorer generalization on spatial changes in OOD data.

Possibly, the use of spatial transformations during training and inference could improve the SAM-Adapter results.

## D.4 Per-Class Analysis

### D.4.1 JHH

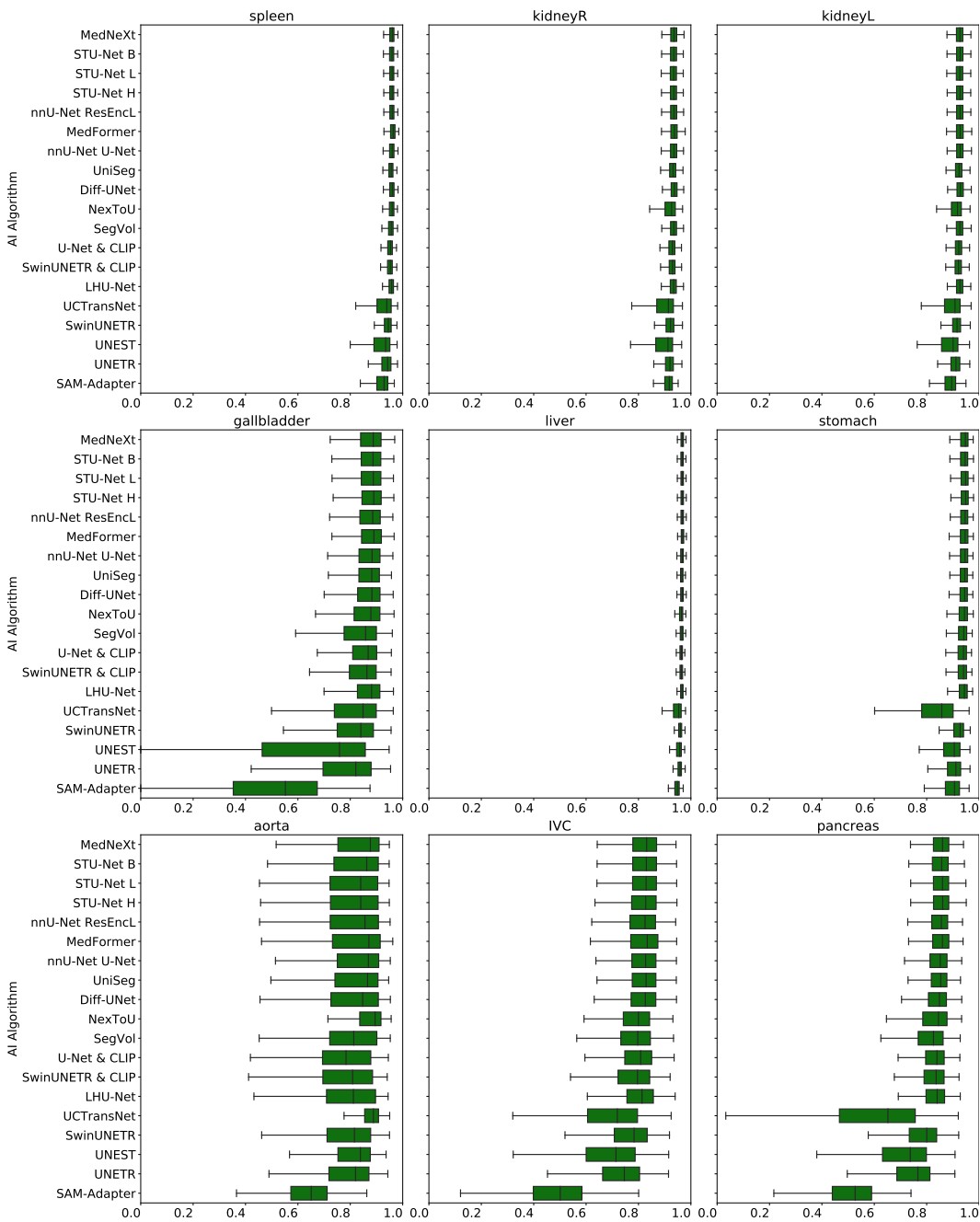

Figure 15: **Boxplots showing DSC score in JHH, per class.** Performances are not homogeneous across classes: structures like the liver, which are easier to segment, show higher median scores and smaller score variation, when compared to more difficult structures, like the gallbladder.

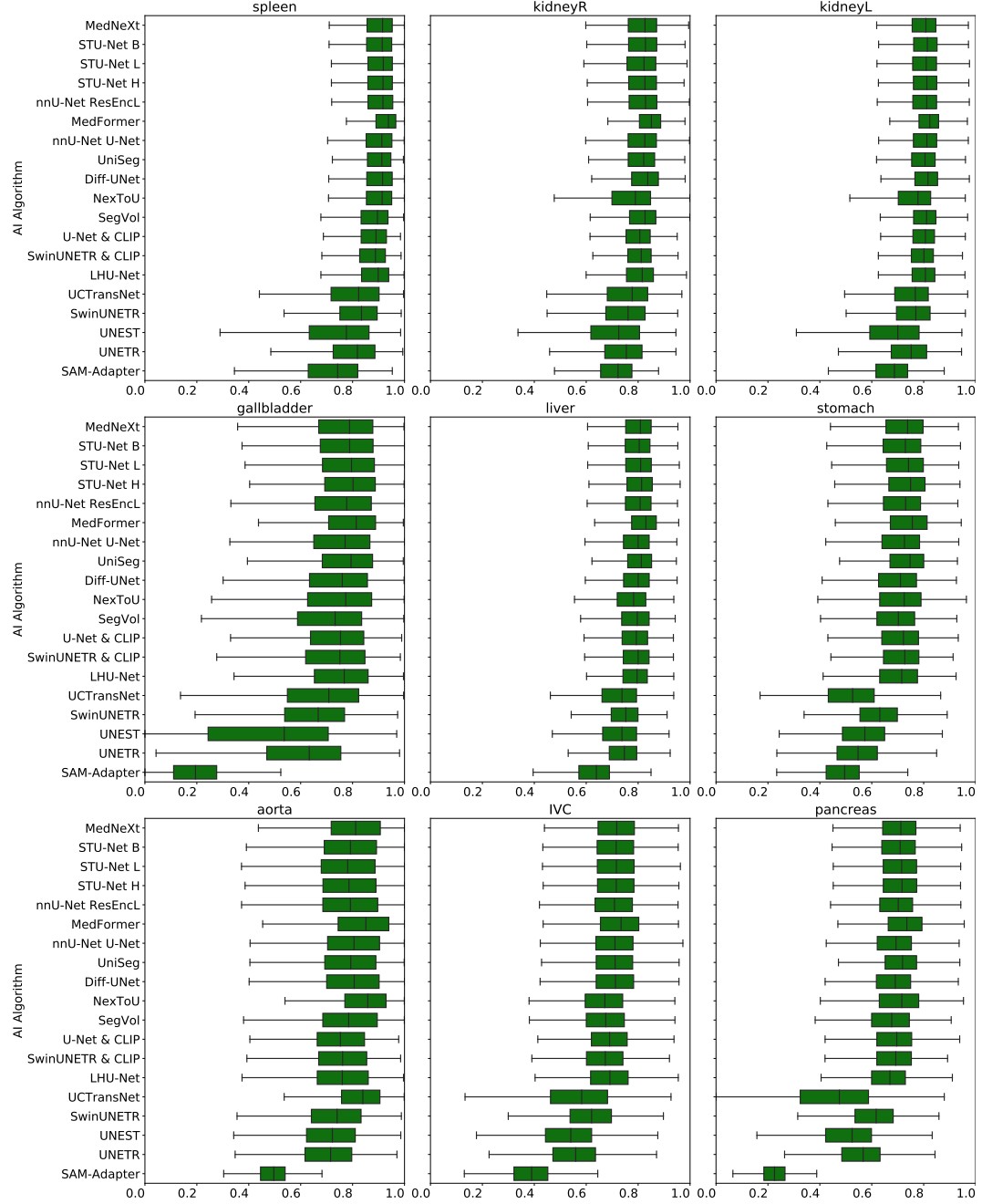

Figure 16: **Boxplots showing NSD score in JHH, per class.** Performances are not homogeneous across classes: structures like the liver, which are easier to segment, show higher median scores and smaller score variation, when compared to more difficult structures, like the gallbladder. NSD considers a threshold of 1.5mm.

## D.4.2 TotalSegmentator

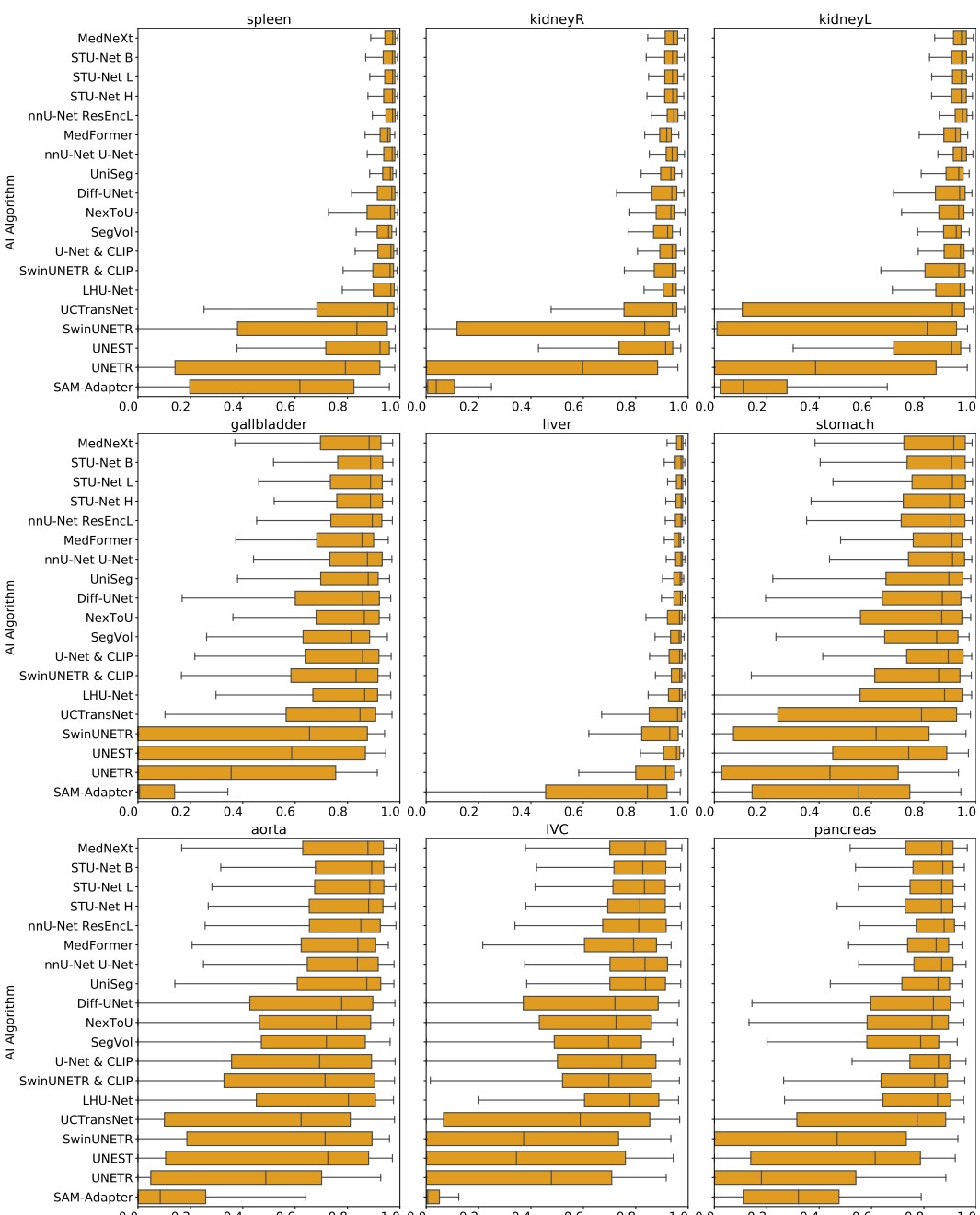

Figure 17: **Boxplots showing DSC score in the entire TotalSegmentator dataset, per class.** Performances are not homogeneous across classes: structures like the liver, which are easier to segment, show higher median scores and smaller score variation, when compared to more difficult structures, like the gallbladder.

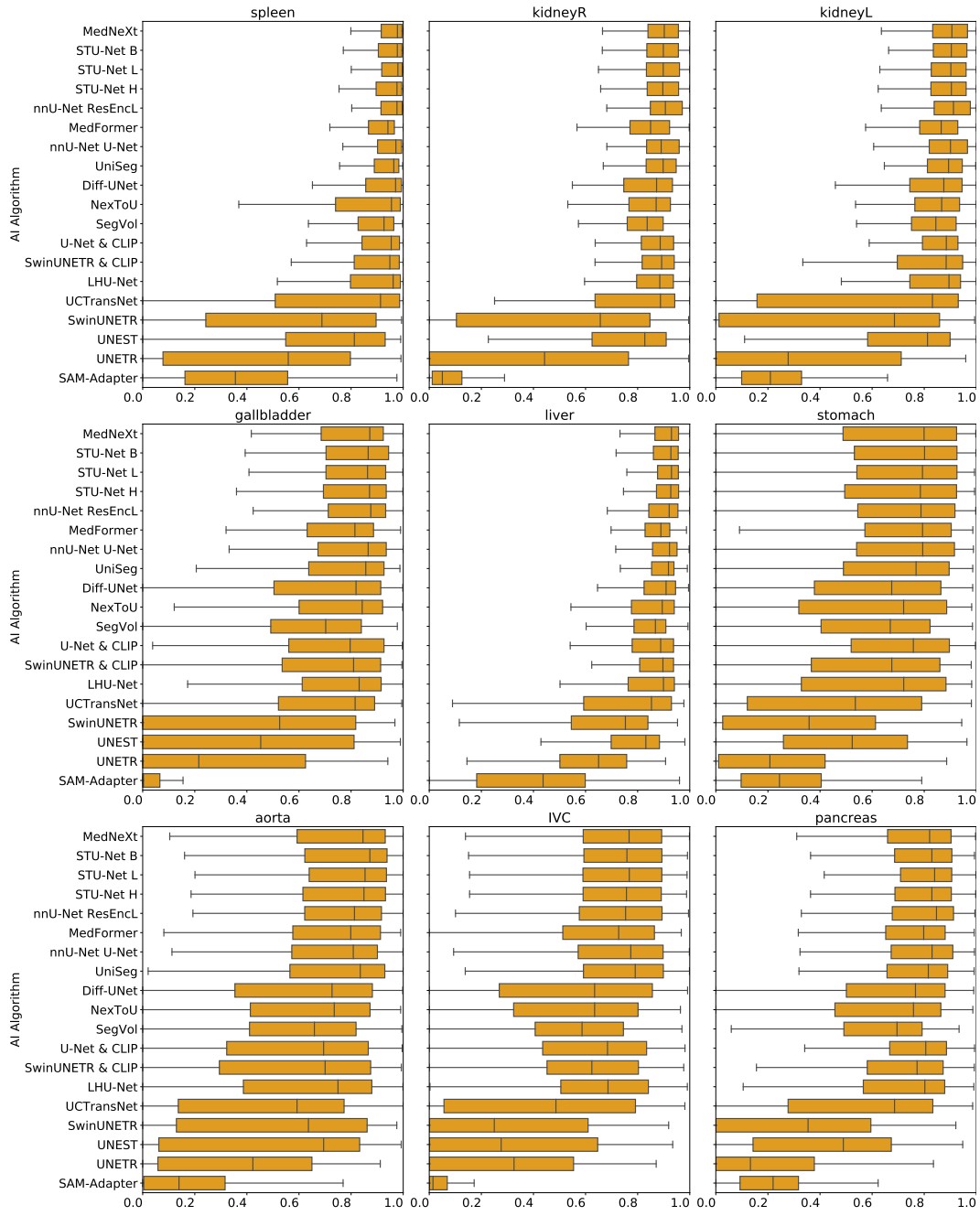

Figure 18: **Boxplots showing NSD score in the entire TotalSegmentator dataset, per class.** Performances are not homogeneous across classes: structures like the liver, which are easier to segment, show higher median scores and smaller score variation, when compared to more difficult structures, like the gallbladder. NSD considers a threshold of 1.5mm.

### D.4.3 TotalSegmentator Official Test Set

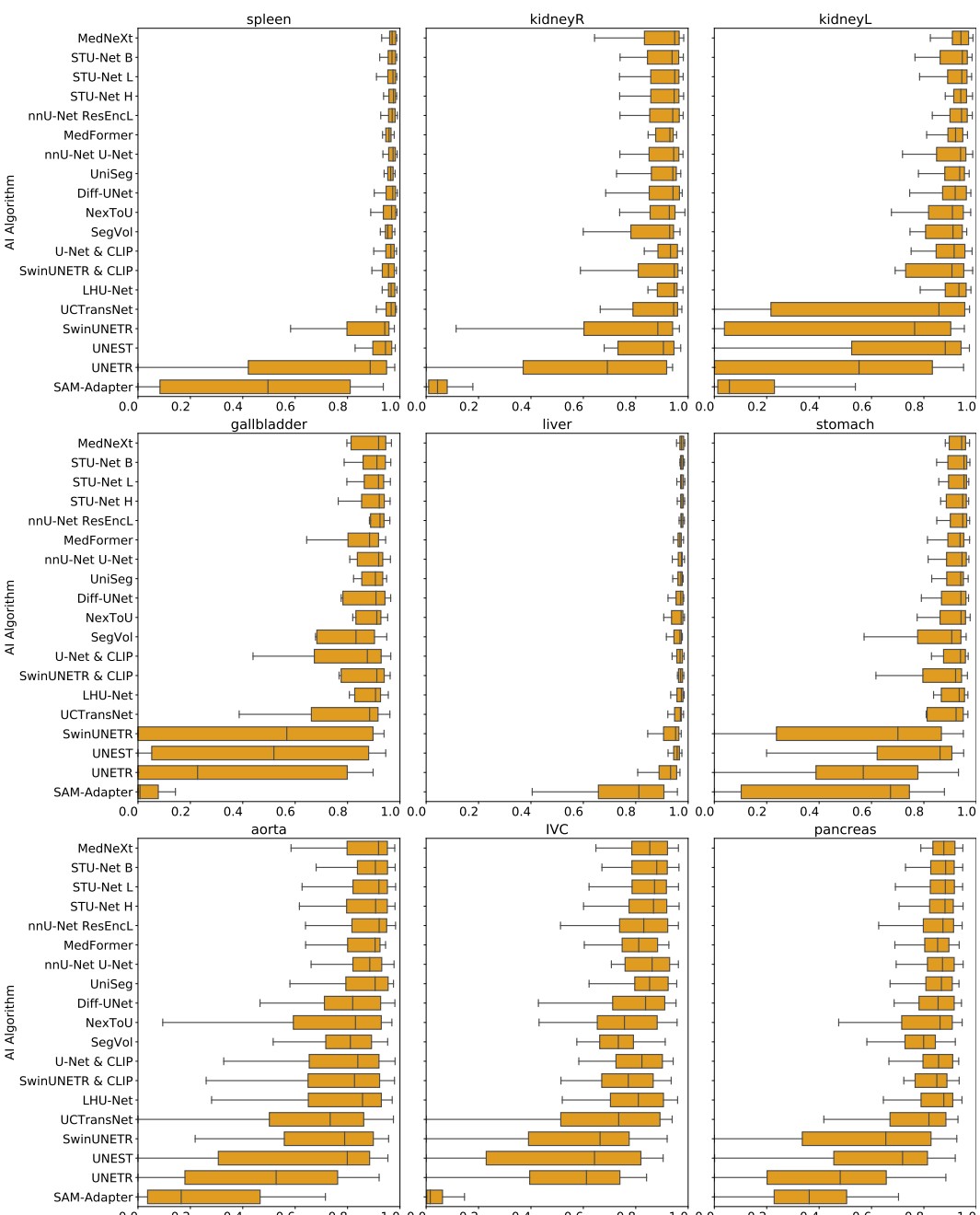

Figure 19: **Boxplots showing DSC score in the TotalSegmentator official test dataset, per class.** Performances are not homogeneous across classes: structures like the liver, which are easier to segment, show higher median scores and smaller score variation, when compared to more difficult structures, like the gallbladder.

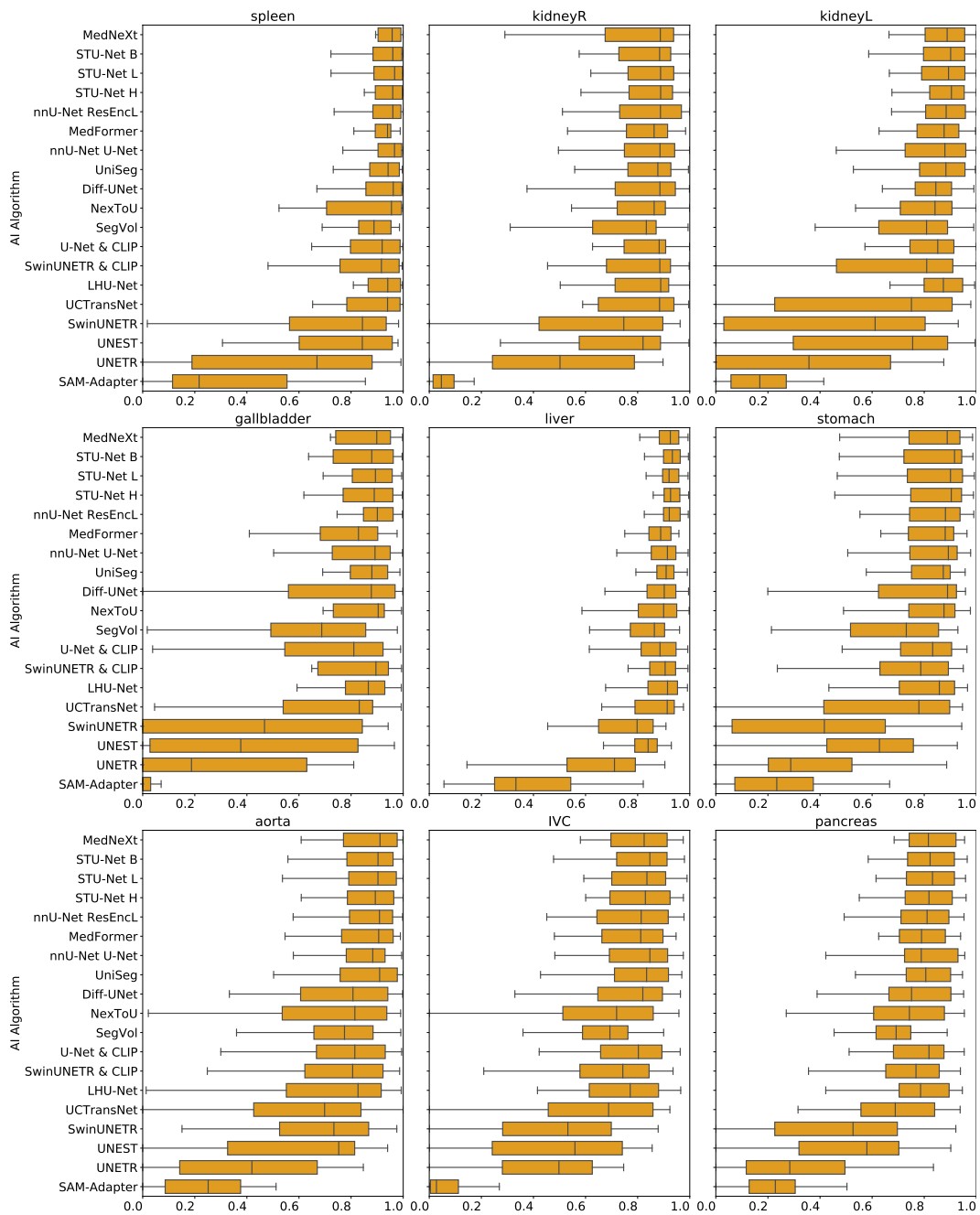

Figure 20: **Boxplots showing NSD score in the TotalSegmentator official test dataset, per class.** Performances are not homogeneous across classes: structures like the liver, which are easier to segment, show higher median scores and smaller score variation, when compared to more difficult structures, like the gallbladder. NSD considers a threshold of 1.5mm.

## D.5 Per-Group Metadata Analysis

### D.5.1 Age

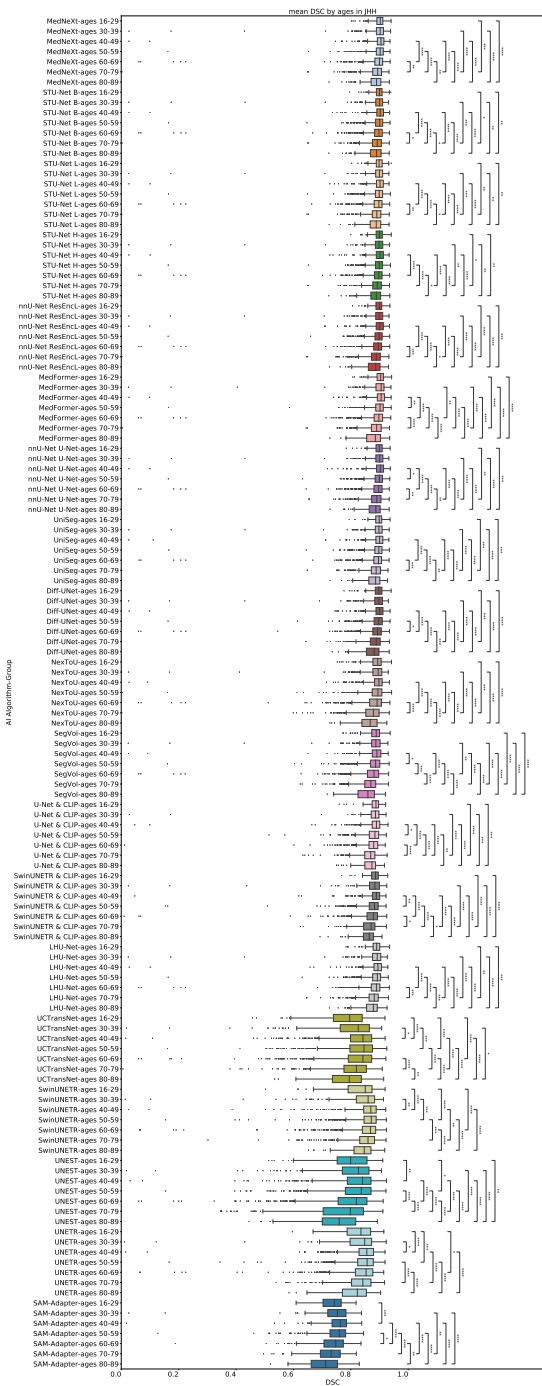

Figure 21: **Boxplot showing average DSC score by age in JHH.** Statistical significance is indicated by stars: * p < 0.05, ** p <0.01, *** p < 0.001, **** p < 0.0001. We perform Kruskal–Wallis tests followed by post-hoc Mann-Whitney U Tests with Bonferroni correction. Here, we did not perform statistical comparisons between diverse AI algorithms. Significant (at least p<0.05) reductions in DSC score for groups with advanced age are observed for all AI algorithms.

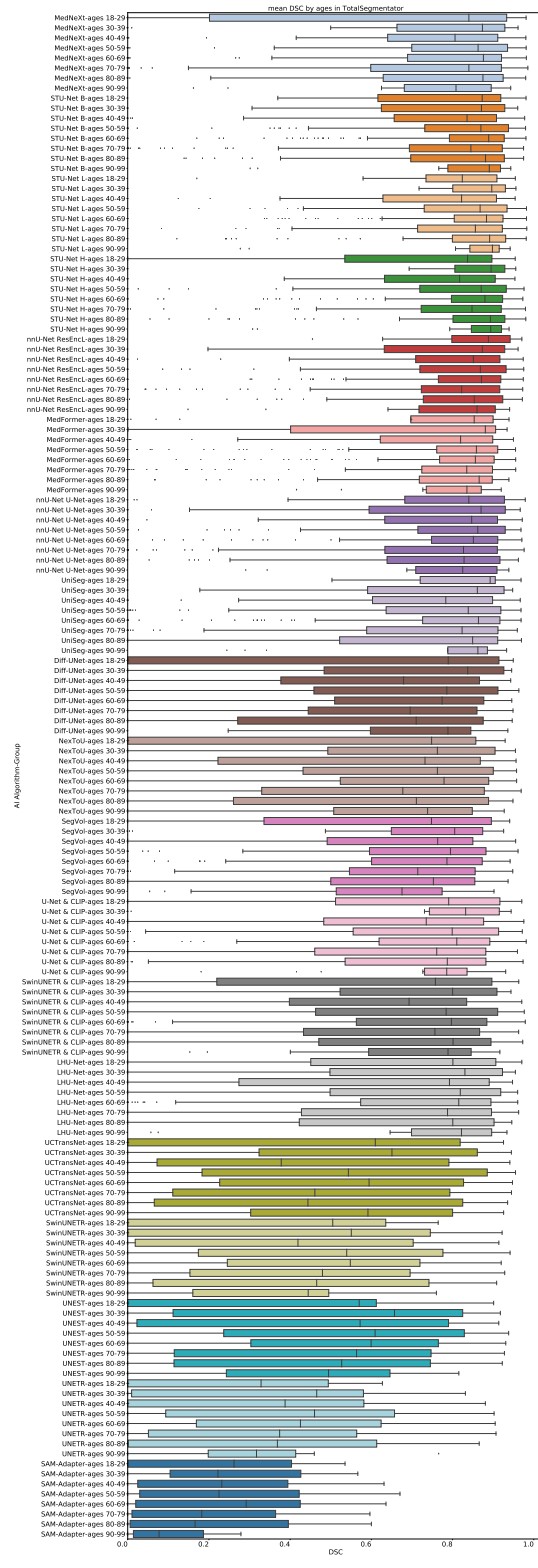

Figure 22: **Boxplot showing average DSC score by age in the whole TotalSegmentator dataset.** Statistical significance is indicated by stars: * p < 0.05, ** p <0.01, *** p < 0.001, **** p < 0.0001. We perform Kruskal–Wallis tests followed by post-hoc Mann-Whitney U Tests with Bonferroni correction. Here, we did not perform statistical comparisons between diverse AI algorithms. Significant differences are not observed, possibly due to the higher variability in the TotalSegmentator results, when compared to other datasets.

### D.5.2 Diagnosis

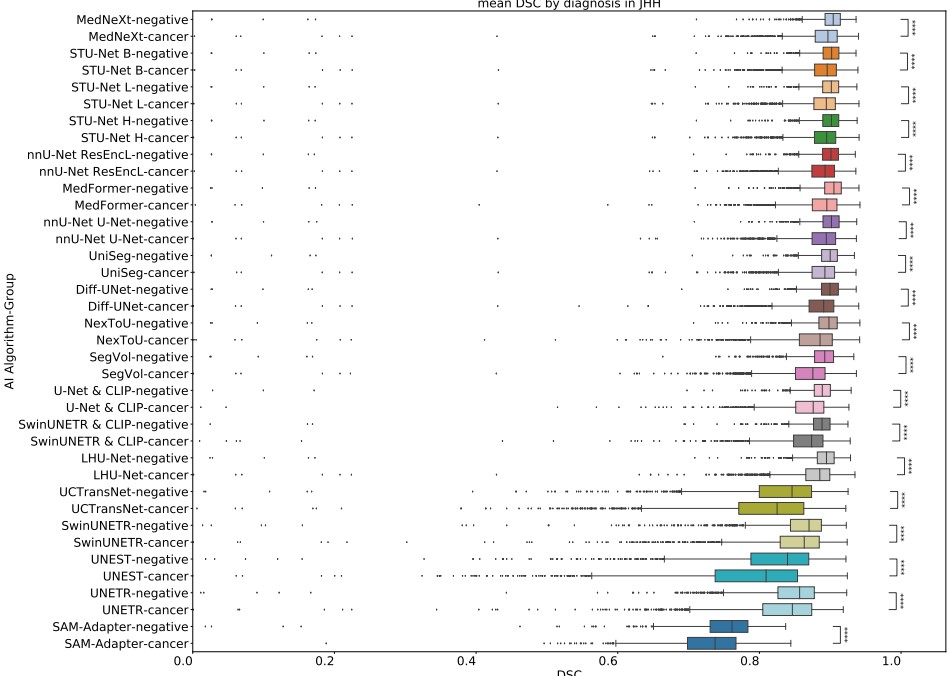

Figure 23: **Boxplot showing average DSC score by diagnosis in JHH.** Statistical significance is indicated by stars: * p < 0.05, ** p <0.01, *** p < 0.001, **** p < 0.0001. We perform Kruskal–Wallis tests followed by post-hoc Mann-Whitney U Tests with Bonferroni correction. Here, we did not perform statistical comparisons between diverse AI algorithms.

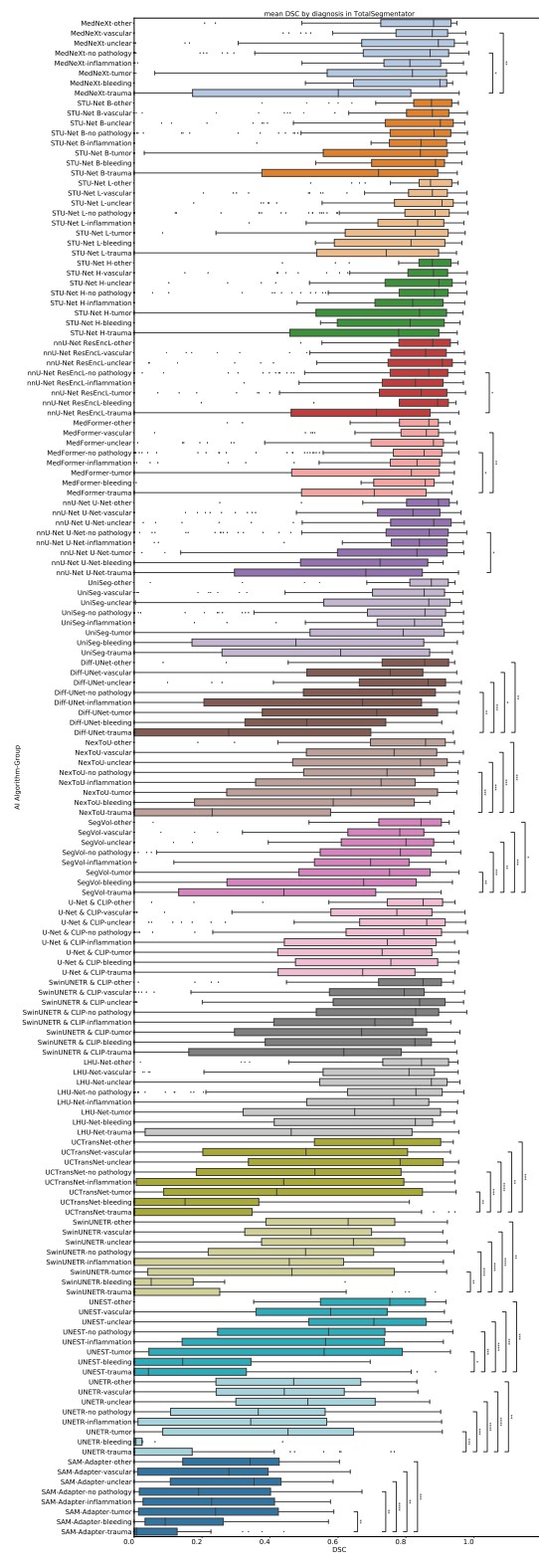

Figure 24: **Boxplot showing average DSC score by diagnosis in the whole TotalSegmentator dataset.** Statistical significance is indicated by stars: * p < 0.05, ** p <0.01, *** p < 0.001, **** p < 0.0001. We perform Kruskal–Wallis tests followed by post-hoc Mann-Whitney U Tests with Bonferroni correction. Here, we did not perform statistical comparisons between diverse AI algorithms.

### D.5.3 Sex

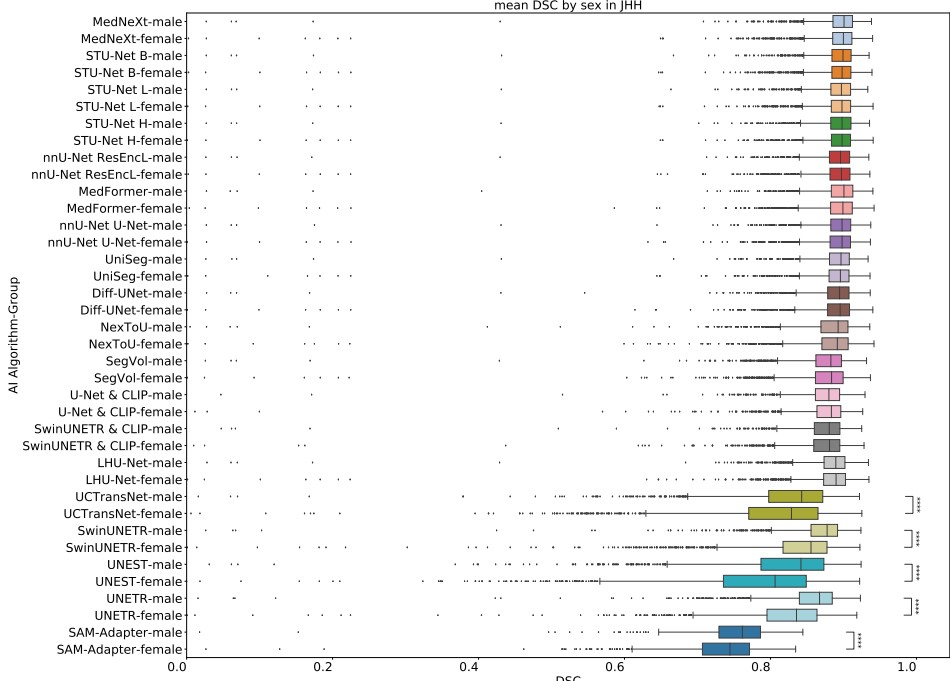

Figure 25: **Boxplot showing average DSC score by sex in JHH.** Statistical significance is indicated by stars: * p < 0.05, ** p <0.01, *** p < 0.001, **** p < 0.0001. We perform Kruskal–Wallis tests followed by post-hoc Mann-Whitney U Tests with Bonferroni correction. Here, we did not perform statistical comparisons between diverse AI algorithms. Only the worst performing algorithms show significant performance difference for the male and female groups, with better scores for male. The best performing models show no significant difference.

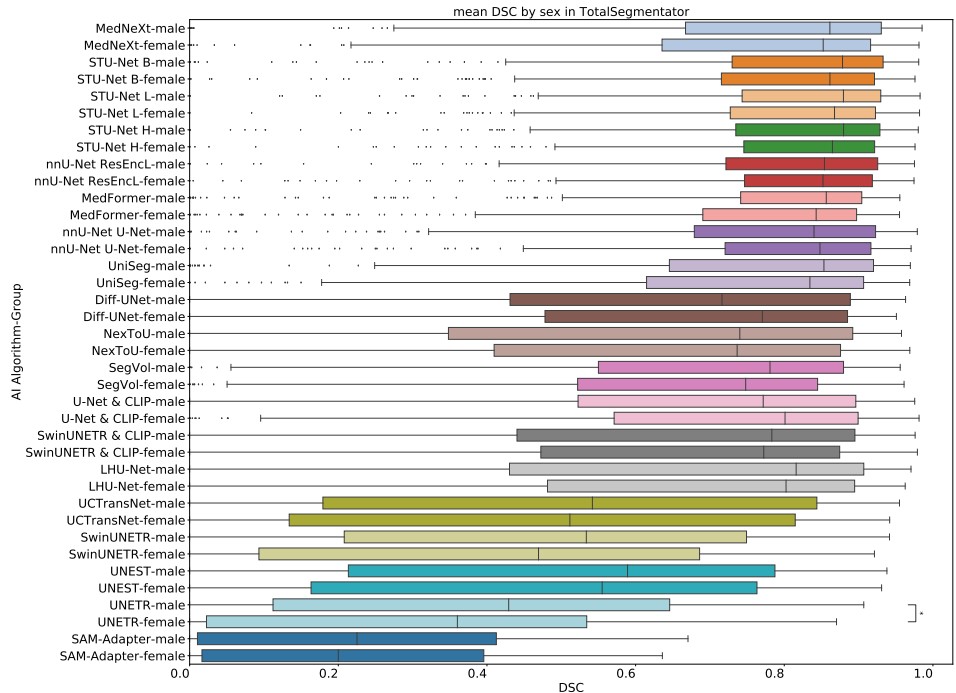

Figure 26: **Boxplot showing average DSC score by sex in the whole TotalSegmentator dataset.** Statistical significance is indicated by stars: * p < 0.05, ** p <0.01, *** p < 0.001, **** p < 0.0001. We perform Kruskal–Wallis tests followed by post-hoc Mann-Whitney U Tests with Bonferroni correction. Here, we did not perform statistical comparisons between diverse AI algorithms. Only the worst performing algorithms show significant performance difference for the male and female groups, with better scores for male. The best performing models show no significant difference.

**D.5.4  Race**

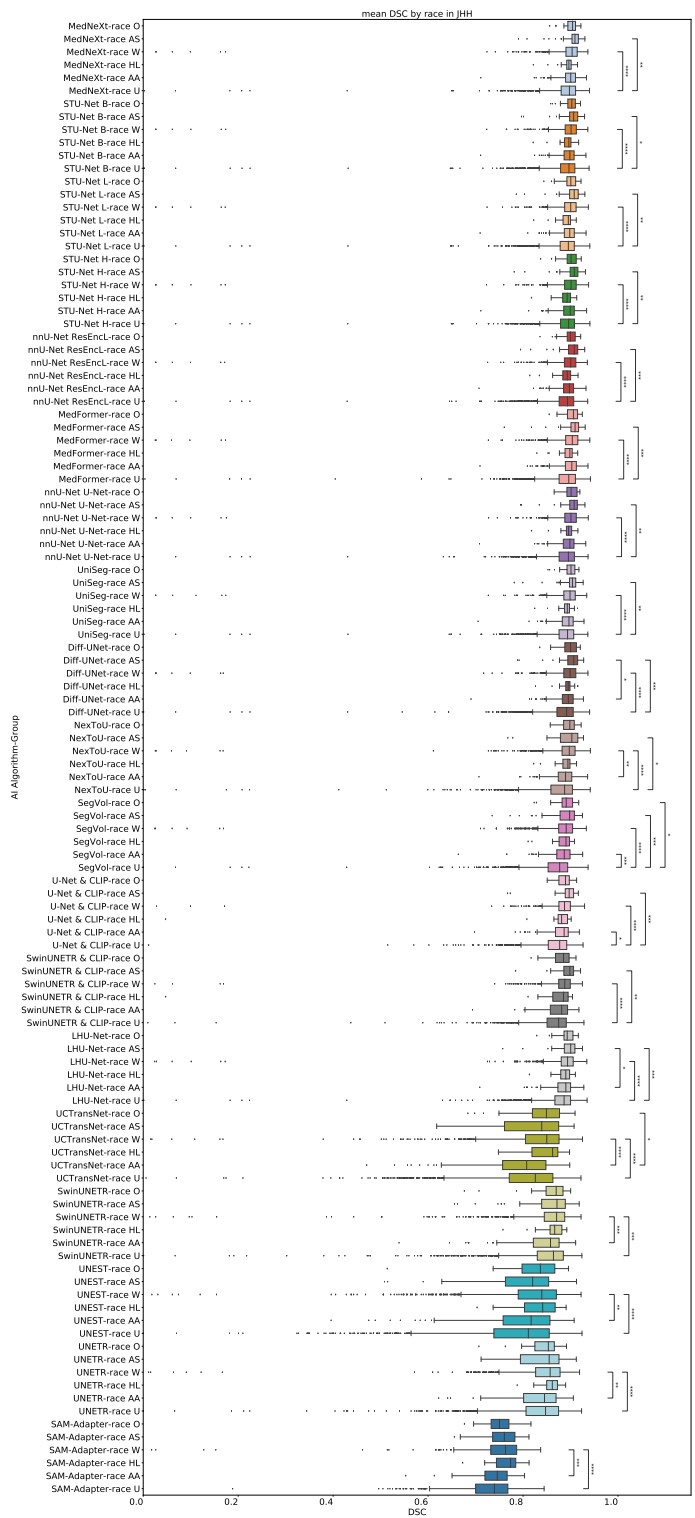

Figure 27: **Boxplot showing average DSC score by race in JHH.** Statistical significance is indicated by stars: * p < 0.05, ** p <0.01, *** p < 0.001, **** p < 0.0001. We perform Kruskal–Wallis tests followed by post-hoc Mann-Whitney U Tests with Bonferroni correction. Here, we did not perform statistical comparisons between diverse AI algorithms. Only some algorithms show significant performance differences across race groups. In these cases, the white or Asian groups have significantly better results than African American or Hispanic Latino (usually than African American). Possibly, this finding indicates a predominance of white and Asian people in the training data, and the necessity of increasing the proportion of African Americans and Hispanic Latinos in the training dataset.

### D.5.5 Manufacturer

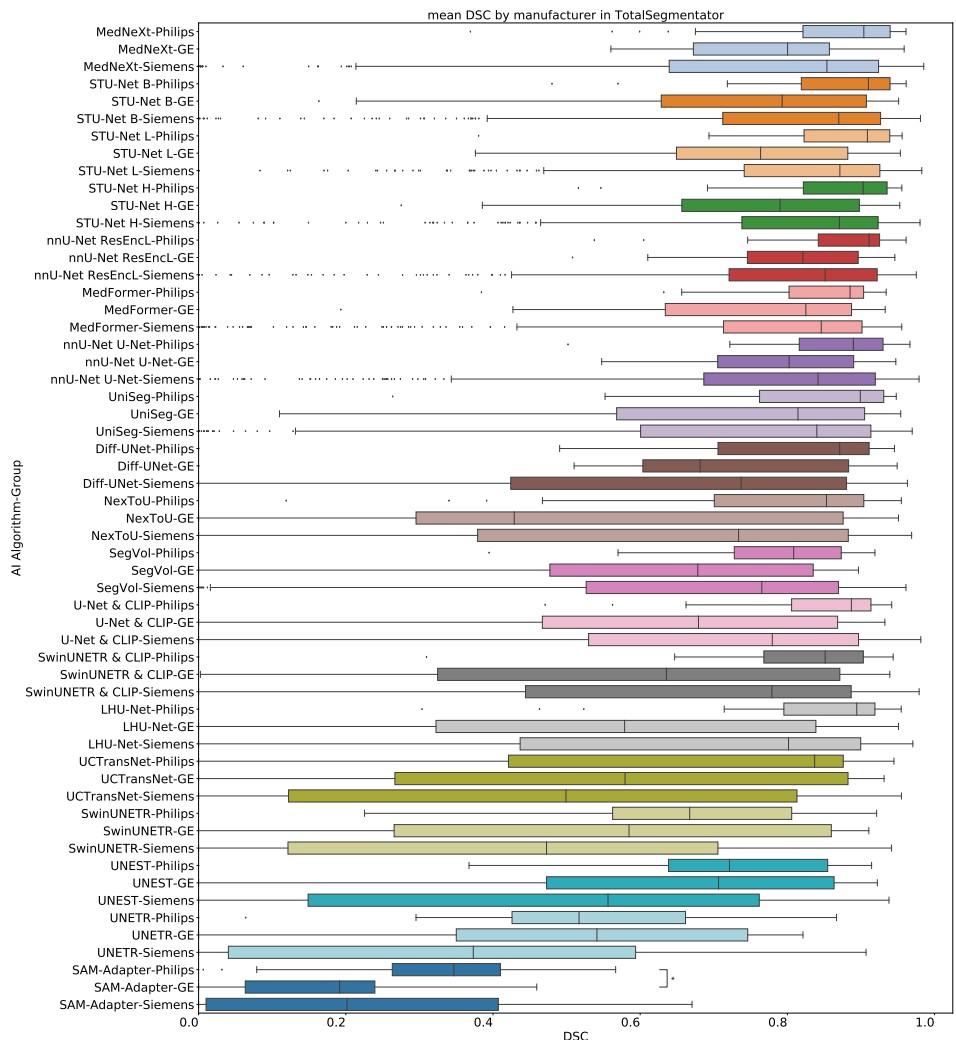

Figure 28: **Boxplot showing average DSC score by manufacturer in the whole TotalSegmentator dataset.** Statistical significance is indicated by stars: * p < 0.05, ** p <0.01, *** p < 0.001, **** p < 0.0001. We perform Kruskal–Wallis tests followed by post-hoc Mann-Whitney U Tests with Bonferroni correction. Here, we did not perform statistical comparisons between diverse AI algorithms.

### D.5.6 Institutes

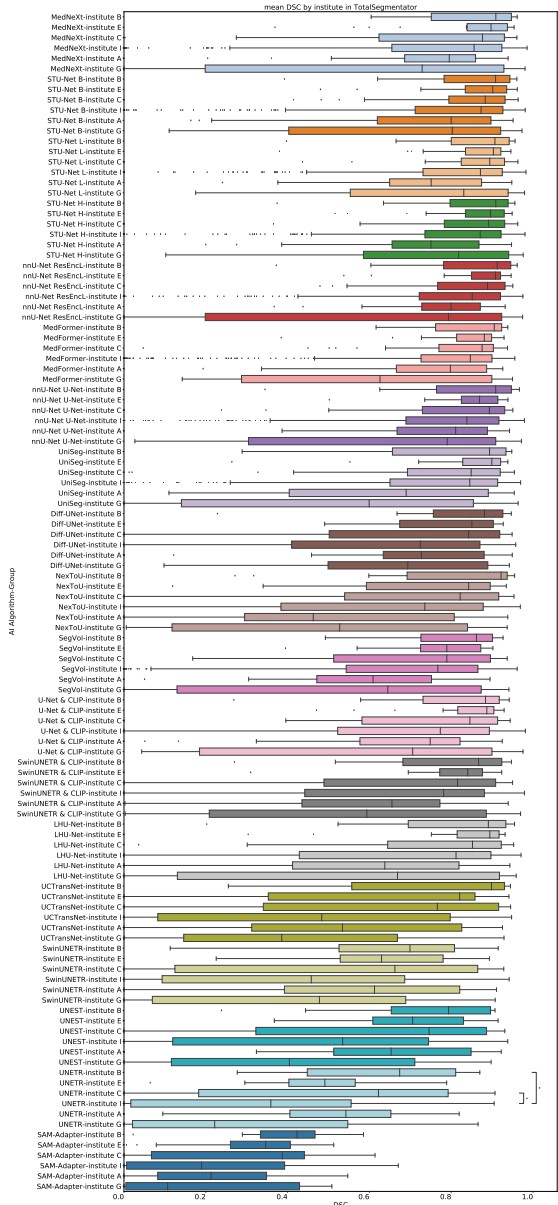

Figure 29: **Boxplot showing average DSC score by institute in the whole TotalSegmentator dataset.** Statistical significance is indicated by stars: * p < 0.05, ** p <0.01, *** p < 0.001, **** p < 0.0001. We perform Kruskal–Wallis tests followed by post-hoc Mann-Whitney U Tests with Bonferroni correction. Here, we did not perform statistical comparisons between diverse AI algorithms. Significant differences across institutes are observed for most AI algorithms, even though all institutes are located on the same country (Switzerland). This finding shows the difficulty of OOD generalization.

### D.5.7 Age: per-class analysis in JHH

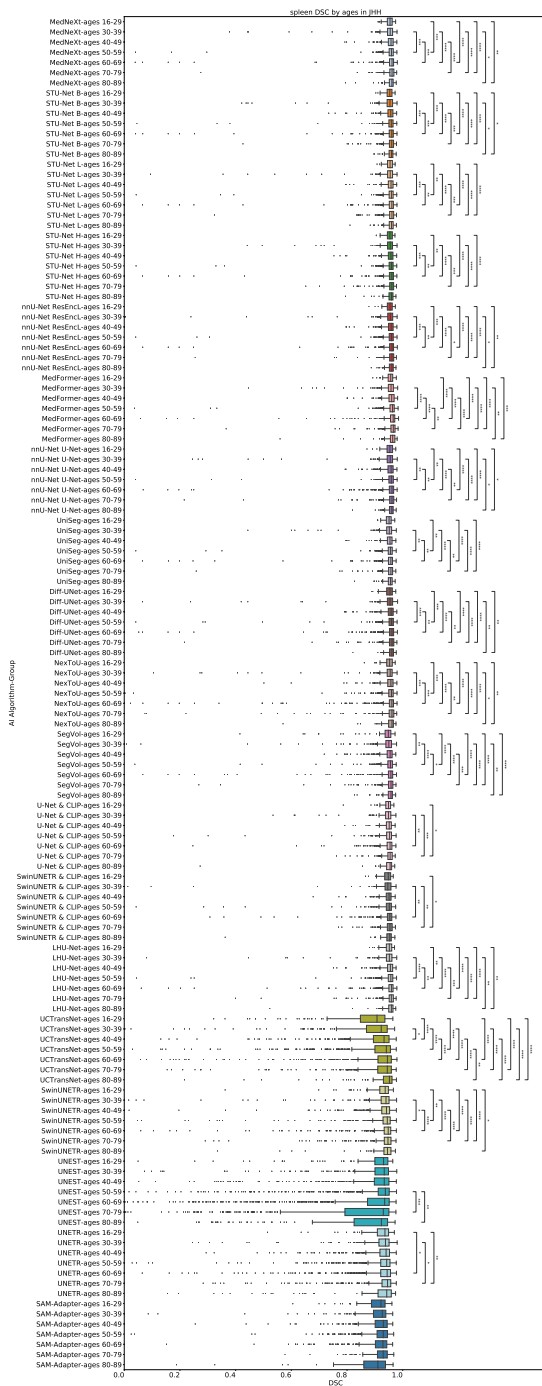

Figure 30: **Boxplot showing spleen DSC score by age in JHH.** Statistical significance is indicated by stars: * p < 0.05, ** p <0.01, *** p < 0.001, **** p < 0.0001. We perform Kruskal–Wallis tests followed by post-hoc Mann-Whitney U Tests with Bonferroni correction. Here, we did not perform statistical comparisons between diverse AI algorithms.

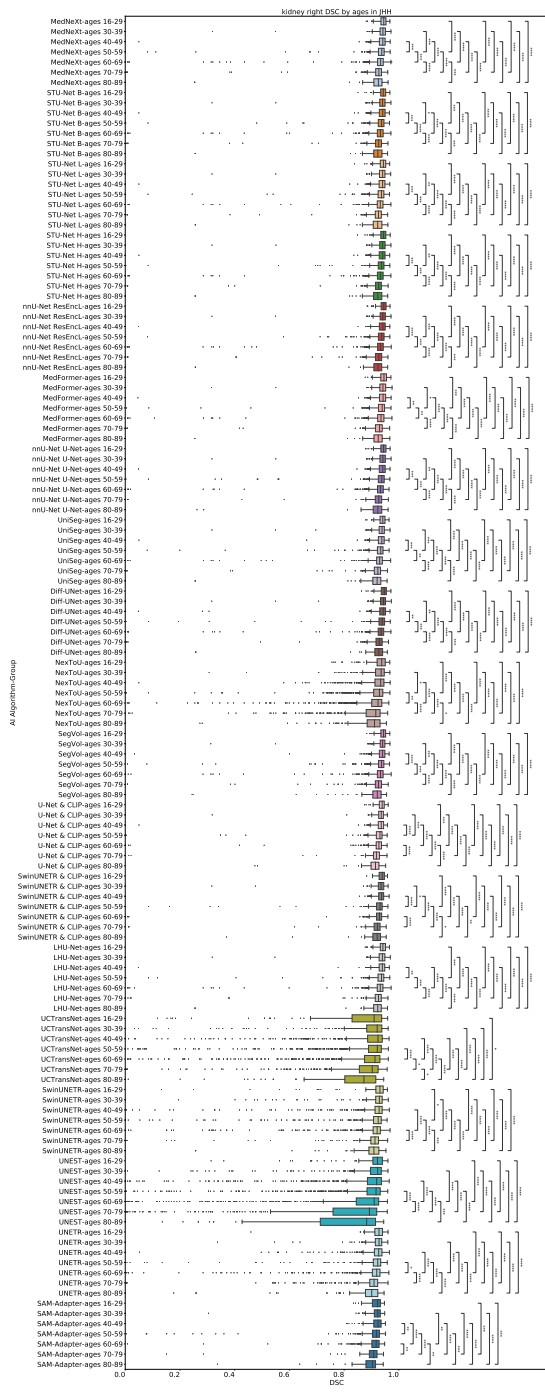

Figure 31: **Boxplot showing right kidney DSC score by age in JHH.** Statistical significance is indicated by stars: * p < 0.05, ** p <0.01, *** p < 0.001, **** p < 0.0001. We perform Kruskal–Wallis tests followed by post-hoc Mann-Whitney U Tests with Bonferroni correction. Here, we did not perform statistical comparisons between diverse AI algorithms.

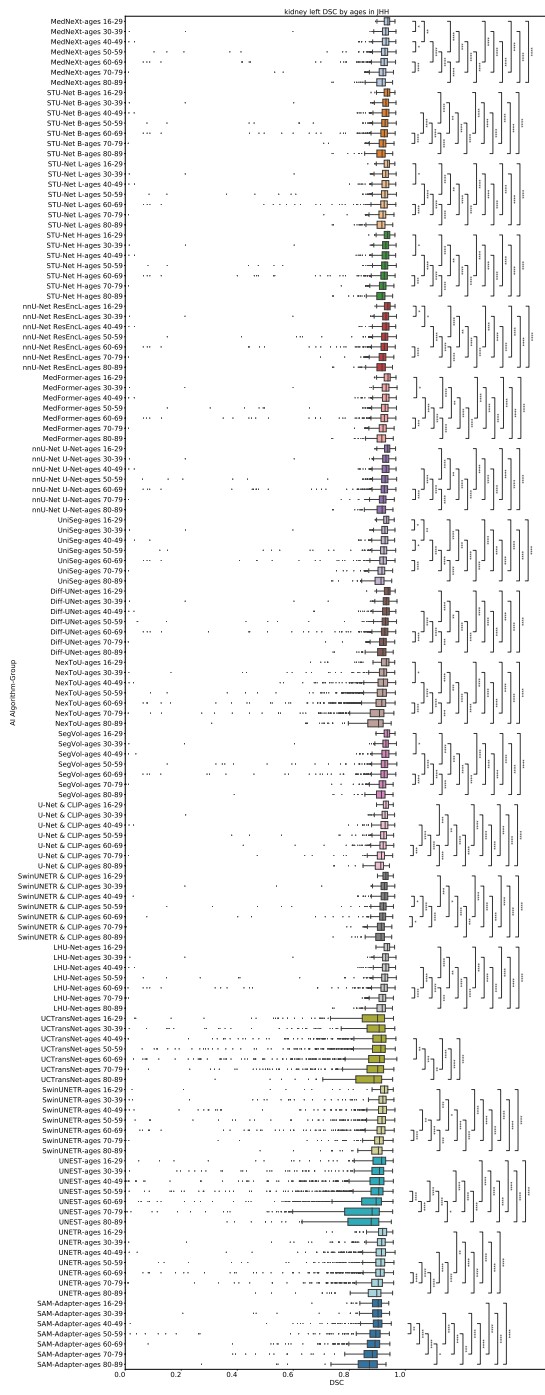

Figure 32: **Boxplot showing left kidney DSC score by age in JHH.** Statistical significance is indicated by stars: * p < 0.05, ** p <0.01, *** p < 0.001, **** p < 0.0001. We perform Kruskal–Wallis tests followed by post-hoc Mann-Whitney U Tests with Bonferroni correction. Here, we did not perform statistical comparisons between diverse AI algorithms.

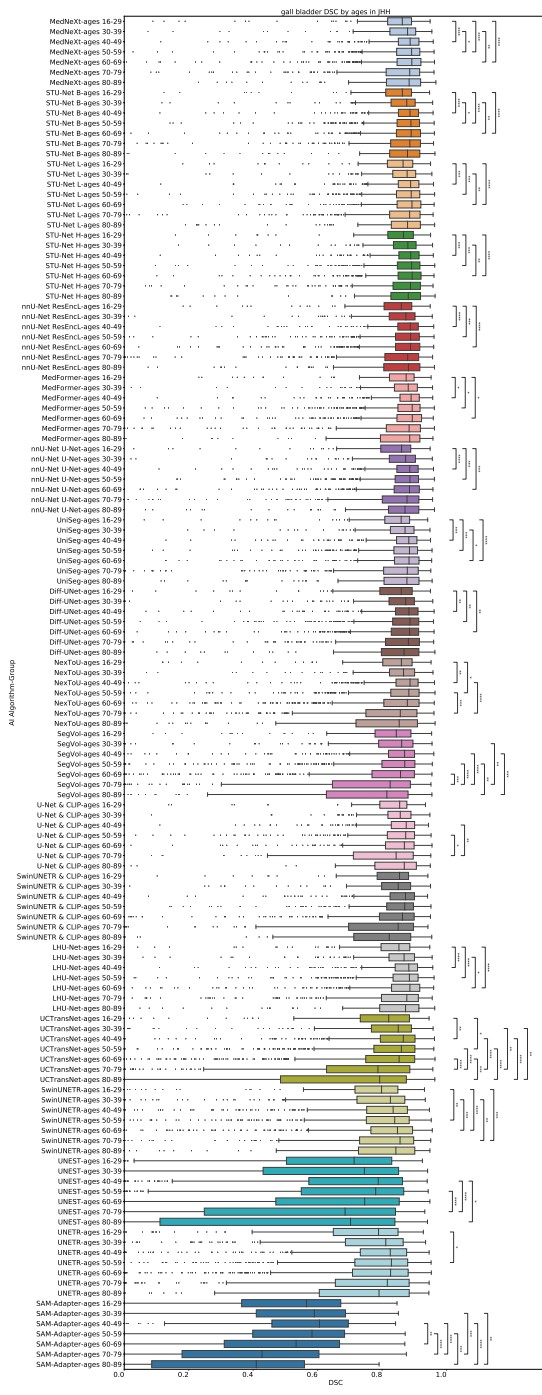

Figure 33: **Boxplot showing gall bladder DSC score by age in JHH.** Statistical significance is indicated by stars: * p < 0.05, ** p <0.01, *** p < 0.001, **** p < 0.0001. We perform Kruskal–Wallis tests followed by post-hoc Mann-Whitney U Tests with Bonferroni correction. Here, we did not perform statistical comparisons between diverse AI algorithms.

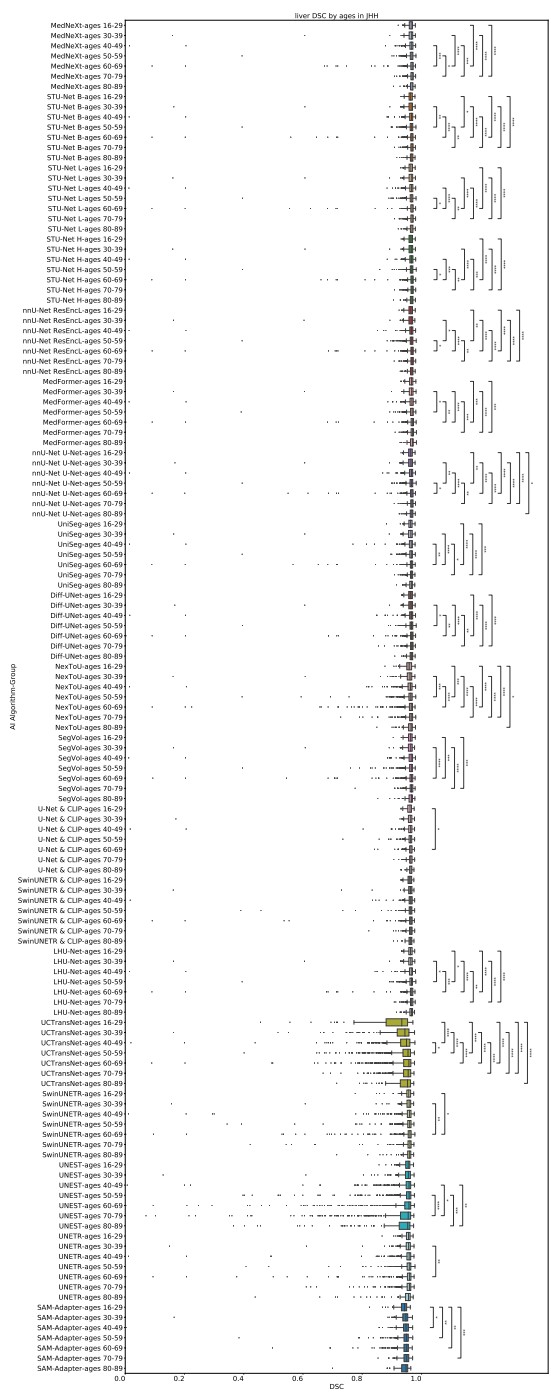

Figure 34: **Boxplot showing liver DSC score by age in JHH.** Statistical significance is indicated by stars: * p < 0.05, ** p <0.01, *** p < 0.001, **** p < 0.0001. We perform Kruskal–Wallis tests followed by post-hoc Mann-Whitney U Tests with Bonferroni correction. Here, we did not perform statistical comparisons between diverse AI algorithms.

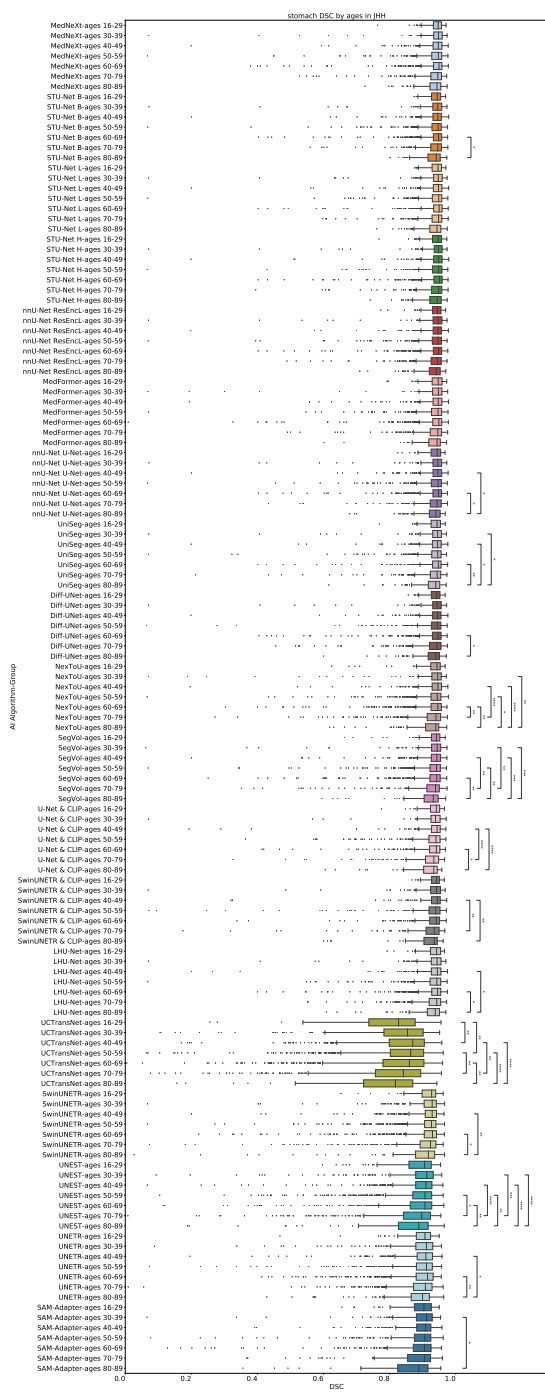

Figure 35: **Boxplot showing stomach DSC score by age in JHH.** Statistical significance is indicated by stars: * p < 0.05, ** p <0.01, *** p < 0.001, **** p < 0.0001. We perform Kruskal–Wallis tests followed by post-hoc Mann-Whitney U Tests with Bonferroni correction. Here, we did not perform statistical comparisons between diverse AI algorithms.

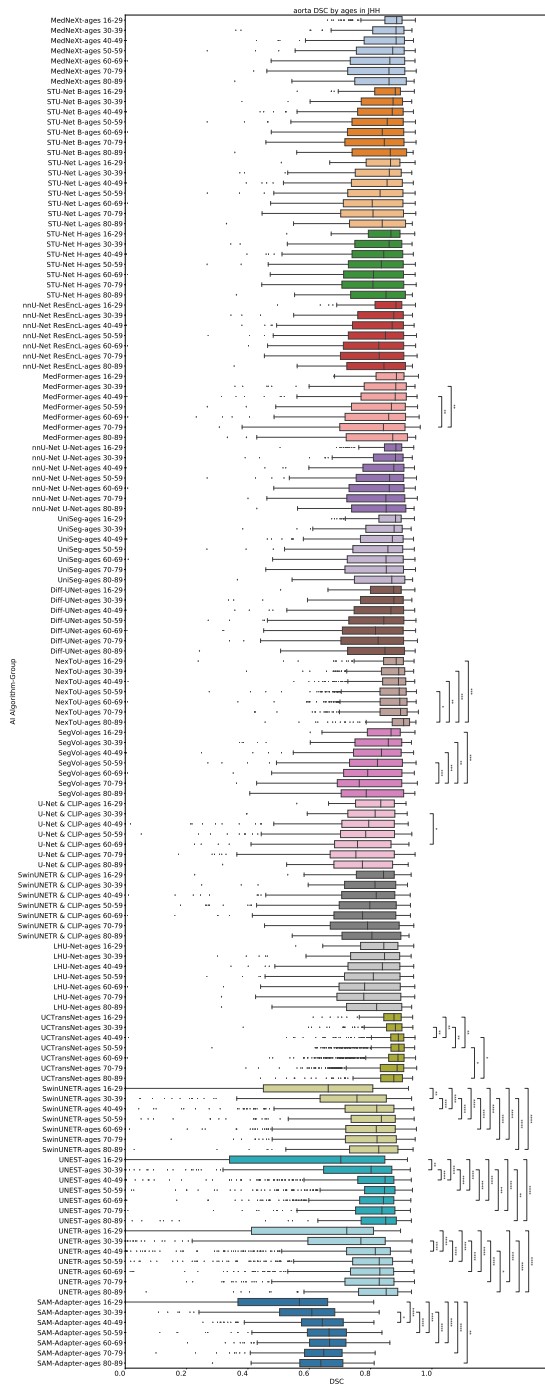

Figure 36: **Boxplot showing aorta DSC score by age in JHH.** Statistical significance is indicated by stars: * p < 0.05, ** p <0.01, *** p < 0.001, **** p < 0.0001. We observed that mean AI performance drops with advanced age, but some AI algorithm's show improving DSC score for aorta after 70. Possibly, an explanation is that the ascending aorta and aortic arch can increase in diameter with age (due to hypertension), and the walls of the vessel will gradually show obvious calcification, possibly making the boundaries clearer. We perform Kruskal–Wallis tests followed by post-hoc Mann-Whitney U Tests with Bonferroni correction. Here, we did not perform statistical comparisons between diverse AI algorithms.

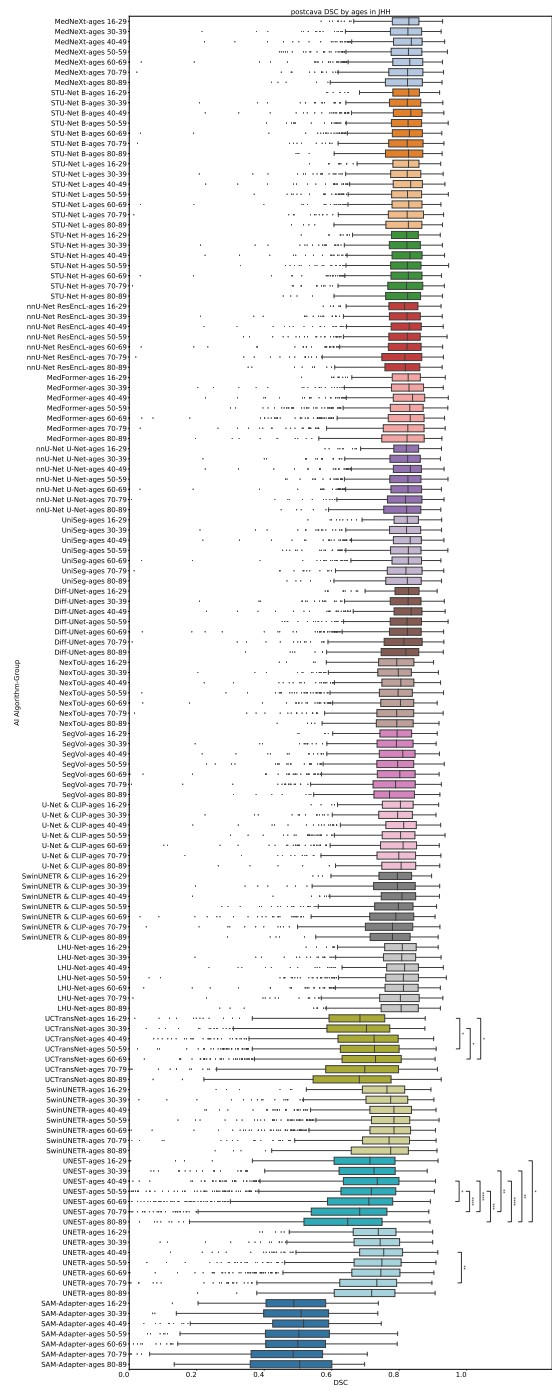

Figure 37: **Boxplot showing postcava DSC score by age in JHH.** Statistical significance is indicated by stars: * p < 0.05, ** p <0.01, *** p < 0.001, **** p < 0.0001. We perform Kruskal–Wallis tests followed by post-hoc Mann-Whitney U Tests with Bonferroni correction. Here, we did not perform statistical comparisons between diverse AI algorithms.

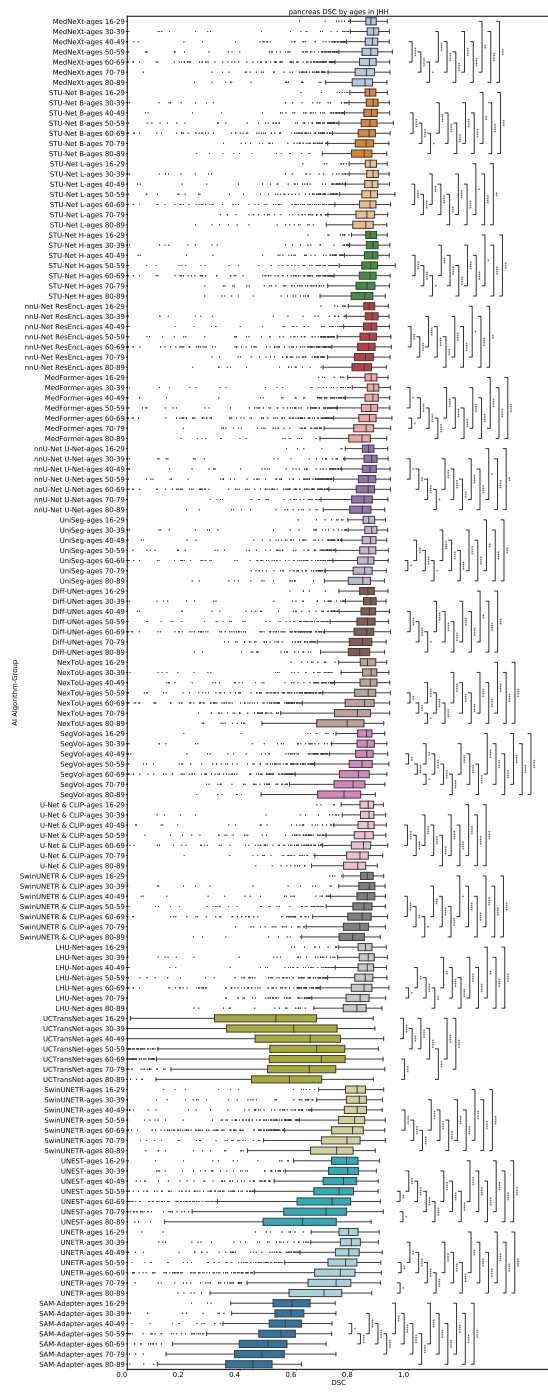

Figure 38: **Boxplot showing pancreas DSC score by age in JHH.** Statistical significance is indicated by stars: * p < 0.05, ** p <0.01, *** p < 0.001, **** p < 0.0001. We perform Kruskal–Wallis tests followed by post-hoc Mann-Whitney U Tests with Bonferroni correction. Here, we did not perform statistical comparisons between diverse AI algorithms.

## D.5.8 Diagnosis: per-class analysis

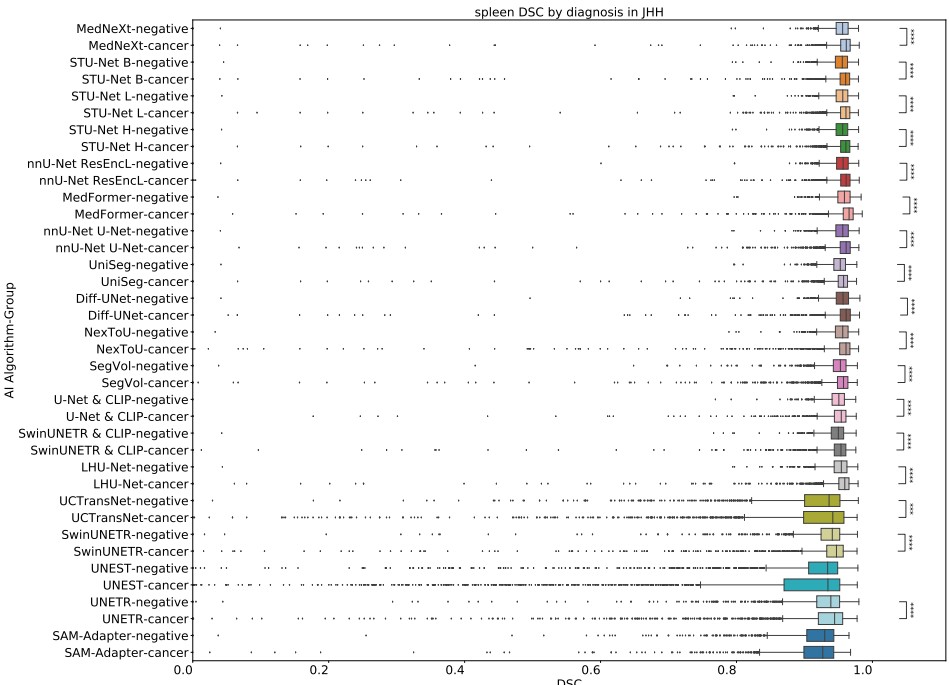

Figure 39: **Boxplot showing spleen DSC score by diagnosis in JHH.** Statistical significance is indicated by stars: * p < 0.05, ** p <0.01, *** p < 0.001, **** p < 0.0001. We perform Kruskal–Wallis tests followed by post-hoc Mann-Whitney U Tests with Bonferroni correction. Here, we did not perform statistical comparisons between diverse AI algorithms.

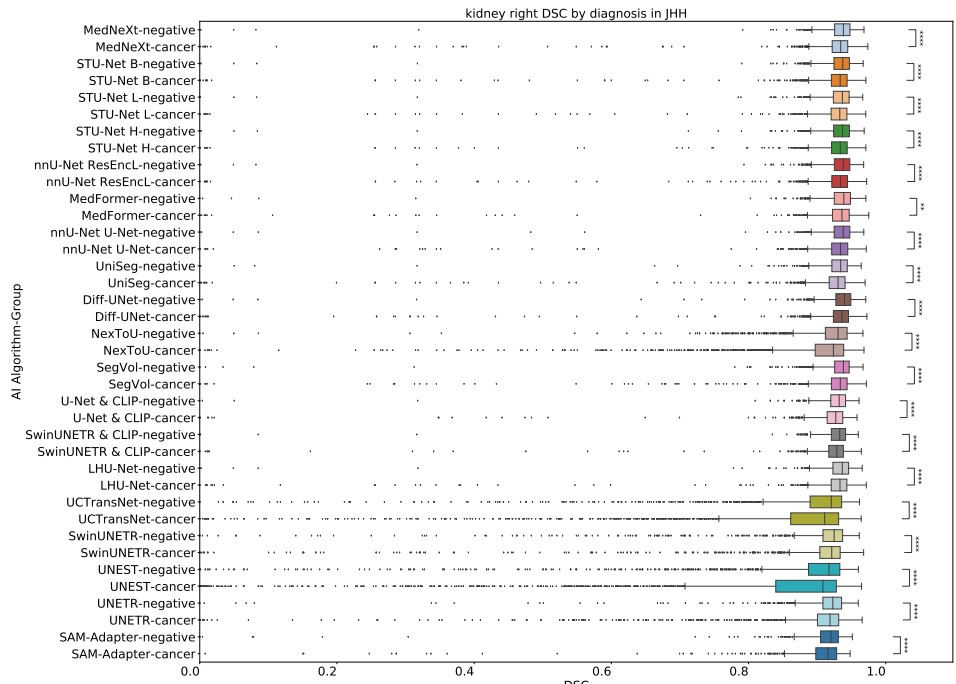

Figure 40: **Boxplot showing right kidney DSC score by diagnosis in JHH.** Statistical significance is indicated by stars: * p < 0.05, ** p <0.01, *** p < 0.001, **** p < 0.0001. We perform Kruskal–Wallis tests followed by post-hoc Mann-Whitney U Tests with Bonferroni correction. Here, we did not perform statistical comparisons between diverse AI algorithms.

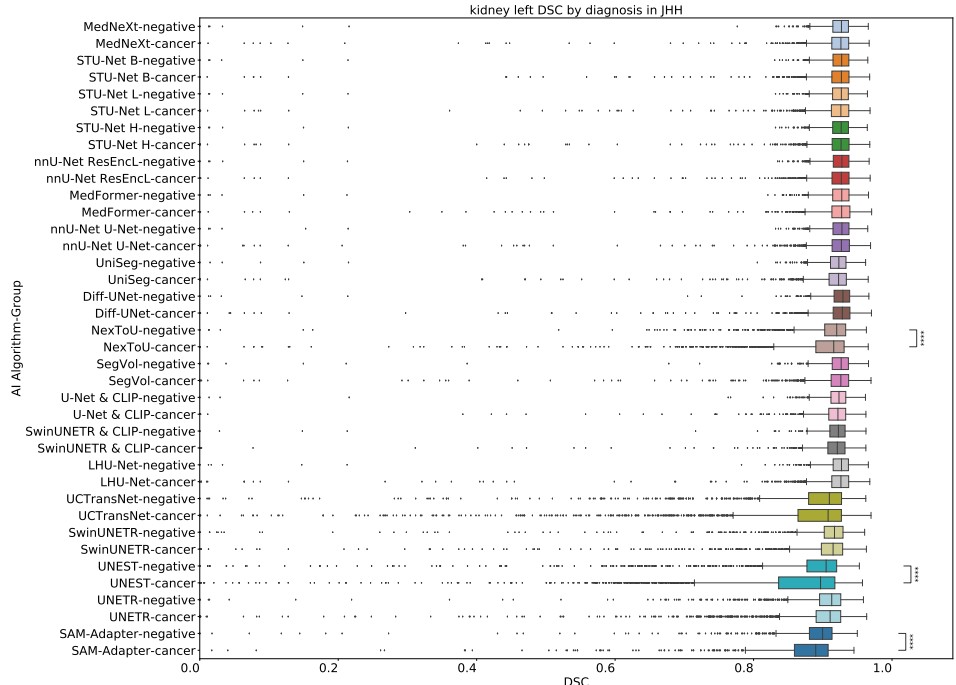

Figure 41: **Boxplot showing left kidney DSC score by diagnosis in JHH.** Statistical significance is indicated by stars: * p < 0.05, ** p <0.01, *** p < 0.001, **** p < 0.0001. We perform Kruskal–Wallis tests followed by post-hoc Mann-Whitney U Tests with Bonferroni correction. Here, we did not perform statistical comparisons between diverse AI algorithms.

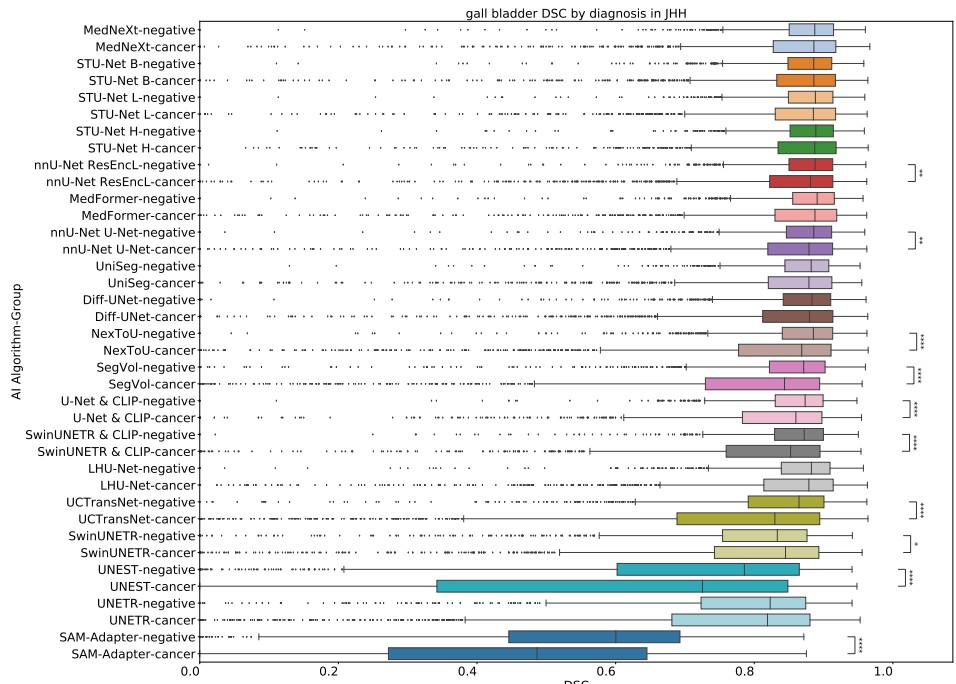

Figure 42: **Boxplot showing gallbladder DSC score by diagnosis in JHH.** Statistical significance is indicated by stars: * p < 0.05, ** p <0.01, *** p < 0.001, **** p < 0.0001. We perform Kruskal–Wallis tests followed by post-hoc Mann-Whitney U Tests with Bonferroni correction. Here, we did not perform statistical comparisons between diverse AI algorithms.

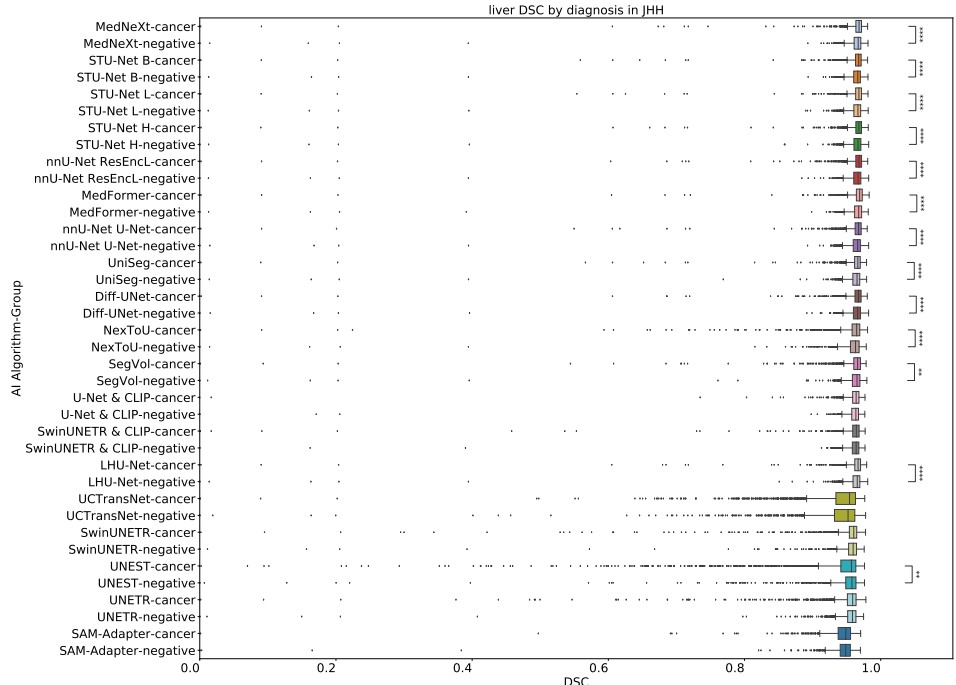

Figure 43: **Boxplot showing liver DSC score by diagnosis in JHH.** Statistical significance is indicated by stars: * p < 0.05, ** p <0.01, *** p < 0.001, **** p < 0.0001. We perform Kruskal–Wallis tests followed by post-hoc Mann-Whitney U Tests with Bonferroni correction. Here, we did not perform statistical comparisons between diverse AI algorithms.

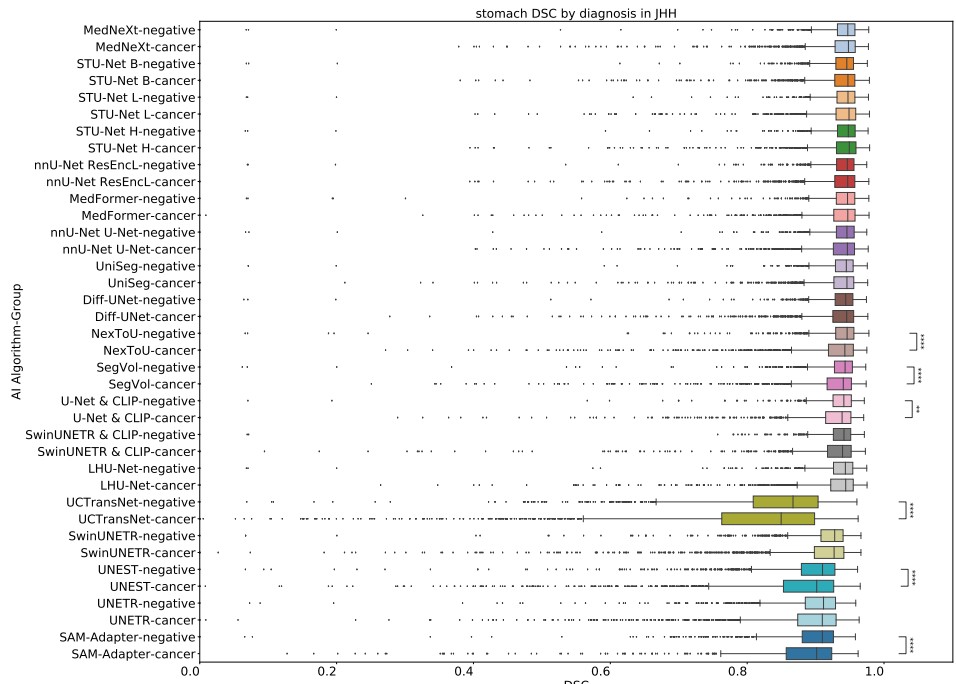

Figure 44: **Boxplot showing stomach DSC score by diagnosis in JHH.** Statistical significance is indicated by stars: * p < 0.05, ** p <0.01, *** p < 0.001, **** p < 0.0001. We perform Kruskal–Wallis tests followed by post-hoc Mann-Whitney U Tests with Bonferroni correction. Here, we did not perform statistical comparisons between diverse AI algorithms.

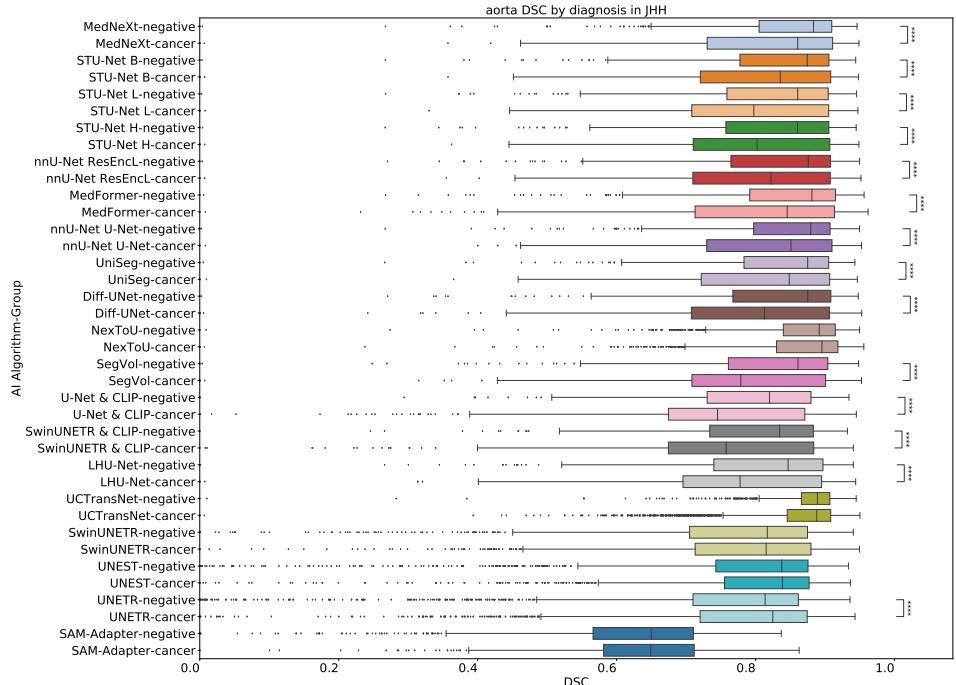

Figure 45: **Boxplot showing aorta DSC score by diagnosis in JHH.** Statistical significance is indicated by stars: * p < 0.05, ** p <0.01, *** p < 0.001, **** p < 0.0001. We perform Kruskal–Wallis tests followed by post-hoc Mann-Whitney U Tests with Bonferroni correction. Here, we did not perform statistical comparisons between diverse AI algorithms.

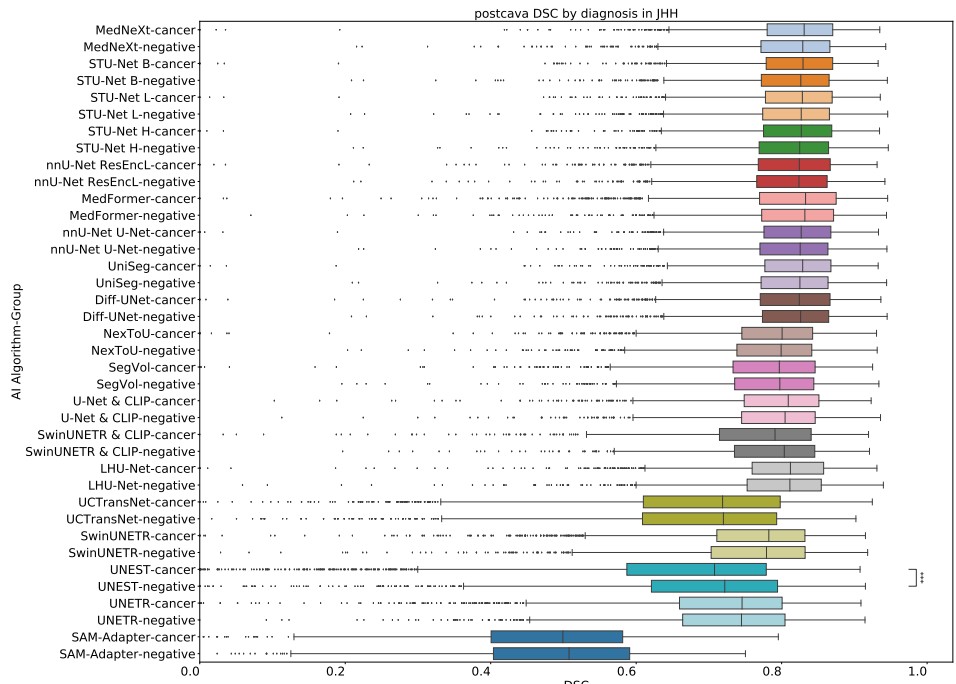

Figure 46: **Boxplot showing postcava DSC score by diagnosis in JHH.** Statistical significance is indicated by stars: * p < 0.05, ** p <0.01, *** p < 0.001, **** p < 0.0001. We perform Kruskal–Wallis tests followed by post-hoc Mann-Whitney U Tests with Bonferroni correction. Here, we did not perform statistical comparisons between diverse AI algorithms.

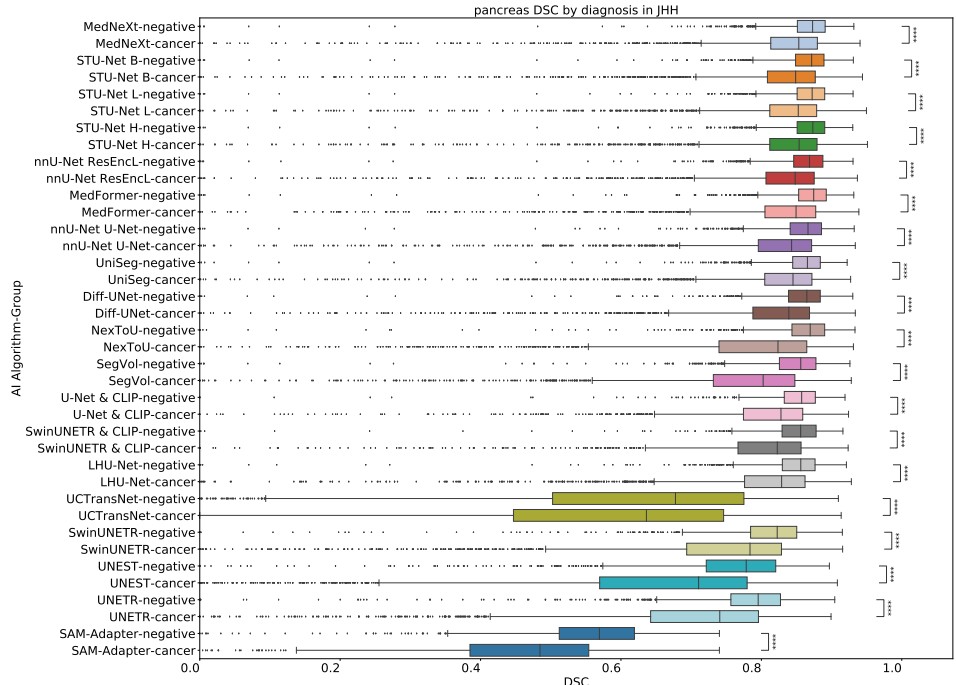

Figure 47: **Boxplot showing pancreas DSC score by diagnosis in JHH.** Statistical significance is indicated by stars: * p < 0.05, ** p <0.01, *** p < 0.001, **** p < 0.0001. We perform Kruskal–Wallis tests followed by post-hoc Mann-Whitney U Tests with Bonferroni correction. Here, we did not perform statistical comparisons between diverse AI algorithms.

### D.5.9 Sex: per-class analysis

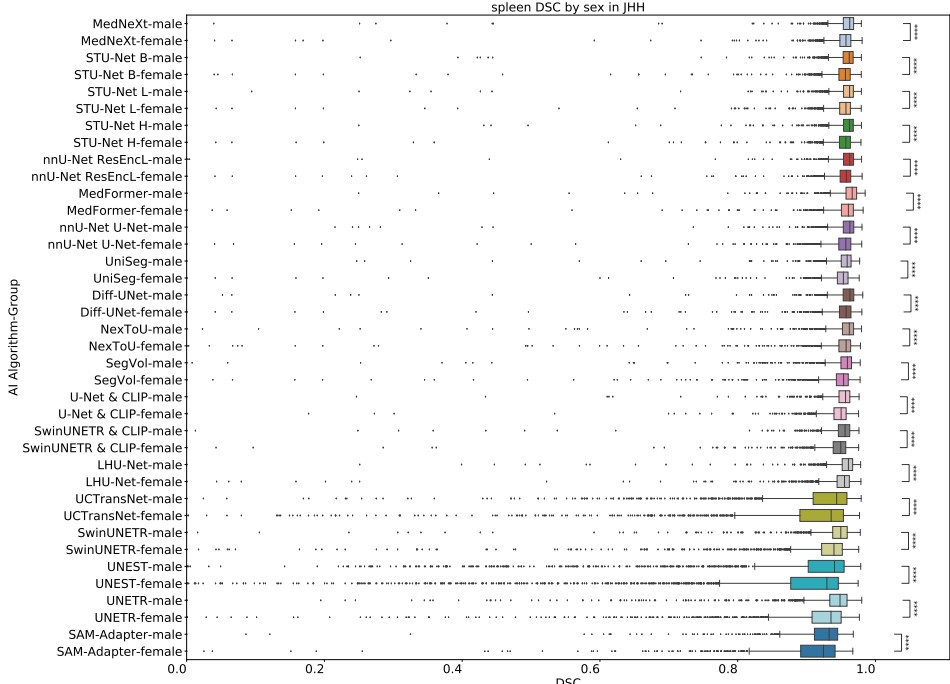

Figure 48: **Boxplot showing spleen DSC score by sex in JHH.** Statistical significance is indicated by stars: * p < 0.05, ** p <0.01, *** p < 0.001, **** p < 0.0001. We perform Kruskal–Wallis tests followed by post-hoc Mann-Whitney U Tests with Bonferroni correction. Here, we did not perform statistical comparisons between diverse AI algorithms.

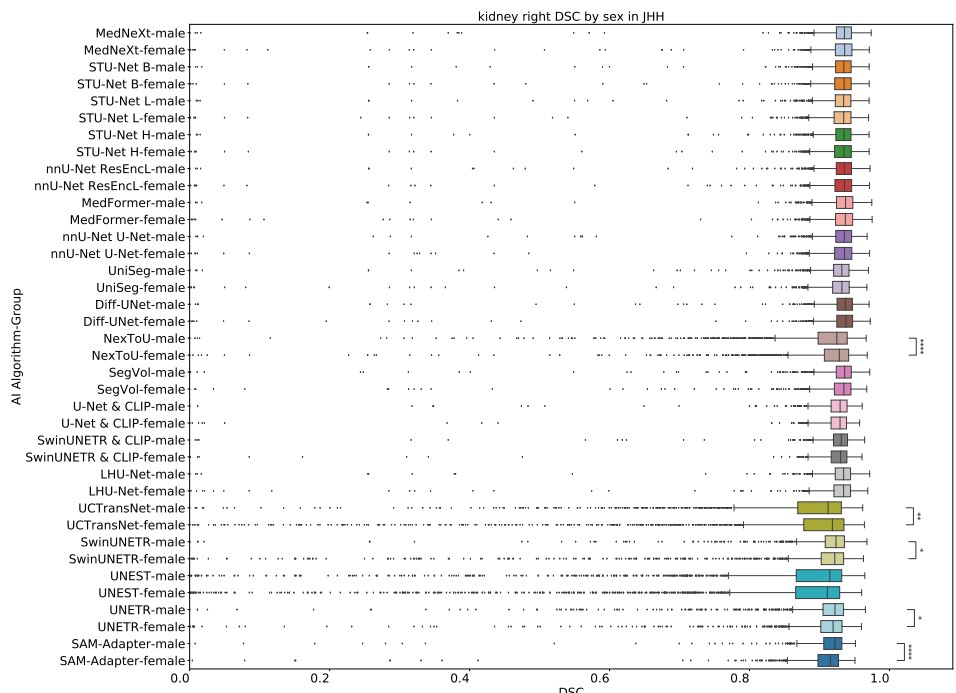

Figure 49: **Boxplot showing right kidney DSC score by sex in JHH.** Statistical significance is indicated by stars: * p < 0.05, ** p <0.01, *** p < 0.001, **** p < 0.0001. We perform Kruskal–Wallis tests followed by post-hoc Mann-Whitney U Tests with Bonferroni correction. Here, we did not perform statistical comparisons between diverse AI algorithms.

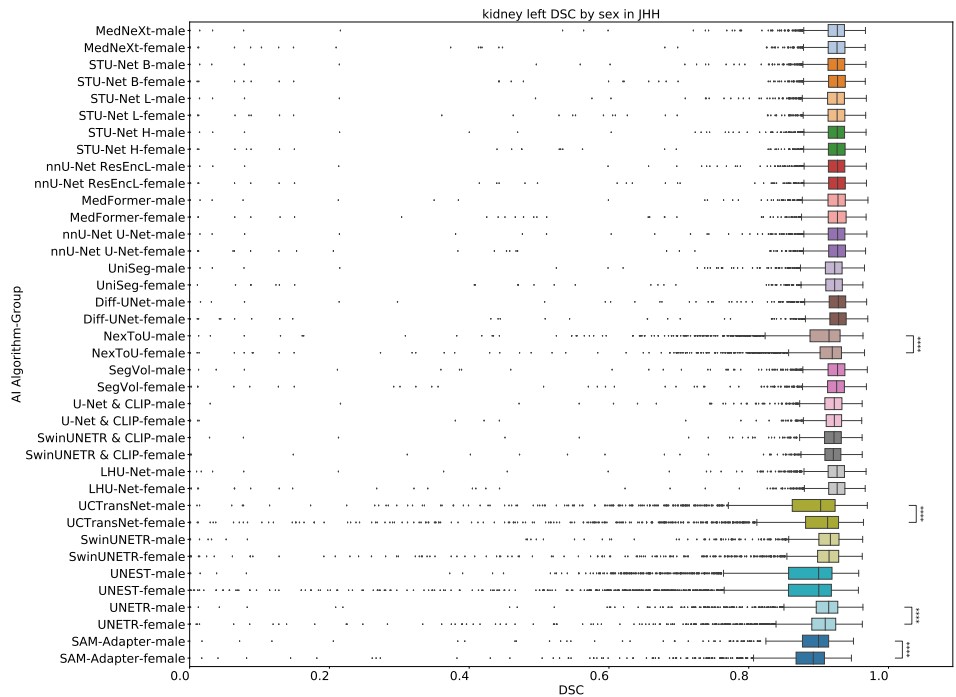

Figure 50: **Boxplot showing left kidney DSC score by sex in JHH.** Statistical significance is indicated by stars: * p < 0.05, ** p <0.01, *** p < 0.001, **** p < 0.0001. We perform Kruskal–Wallis tests followed by post-hoc Mann-Whitney U Tests with Bonferroni correction. Here, we did not perform statistical comparisons between diverse AI algorithms.

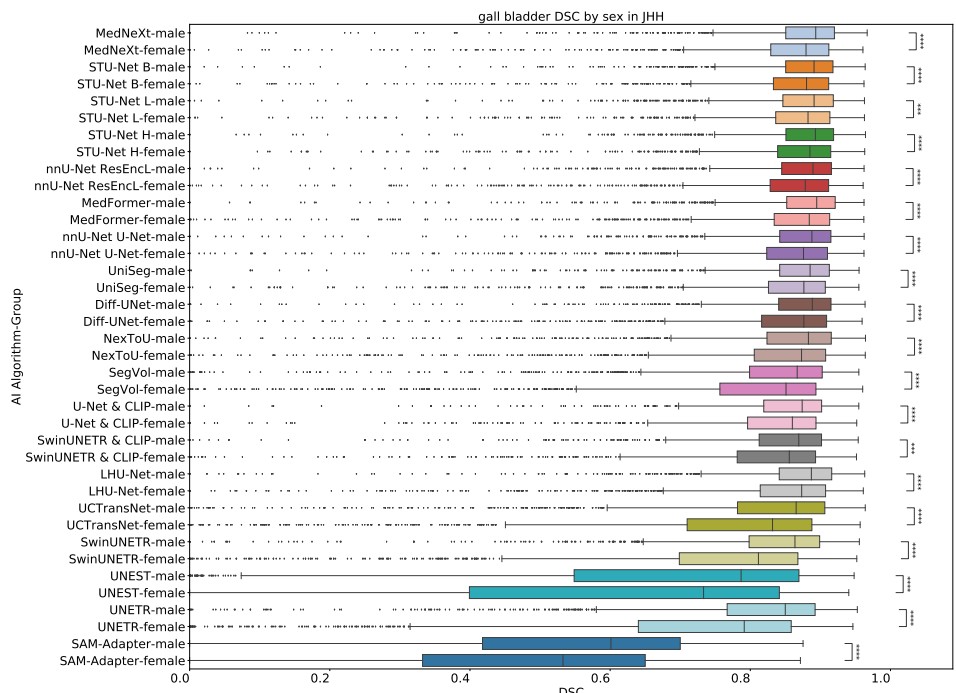

Figure 51: **Boxplot showing gallbladder DSC score by sex in JHH.** Statistical significance is indicated by stars: * p < 0.05, ** p <0.01, *** p < 0.001, **** p < 0.0001. We perform Kruskal–Wallis tests followed by post-hoc Mann-Whitney U Tests with Bonferroni correction. Here, we did not perform statistical comparisons between diverse AI algorithms.

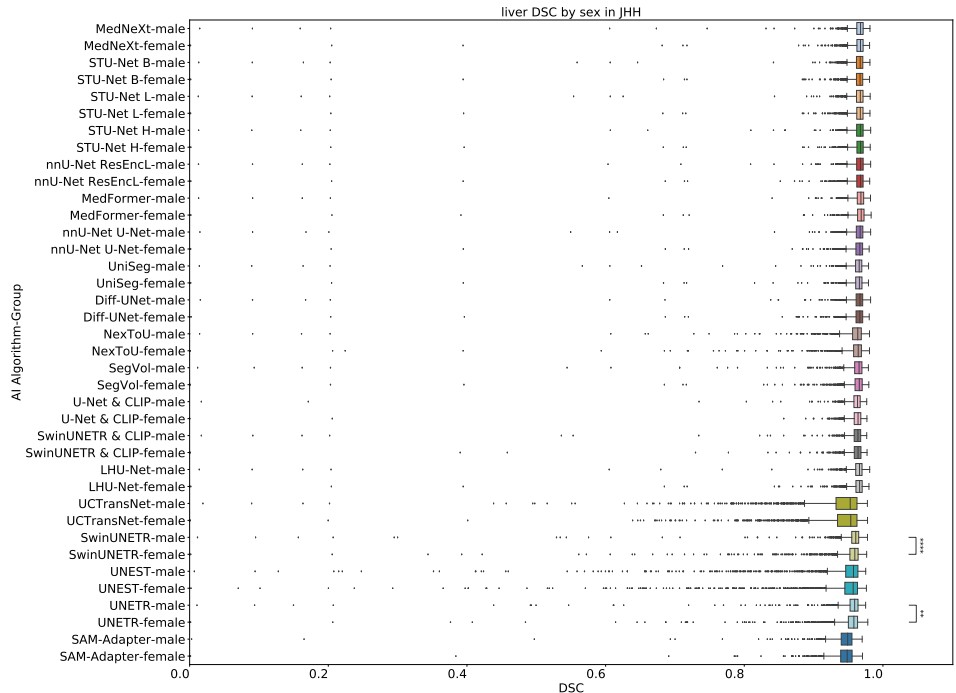

Figure 52: **Boxplot showing liver DSC score by sex in JHH.** Statistical significance is indicated by stars: * p < 0.05, ** p <0.01, *** p < 0.001, **** p < 0.0001. We perform Kruskal–Wallis tests followed by post-hoc Mann-Whitney U Tests with Bonferroni correction. Here, we did not perform statistical comparisons between diverse AI algorithms.

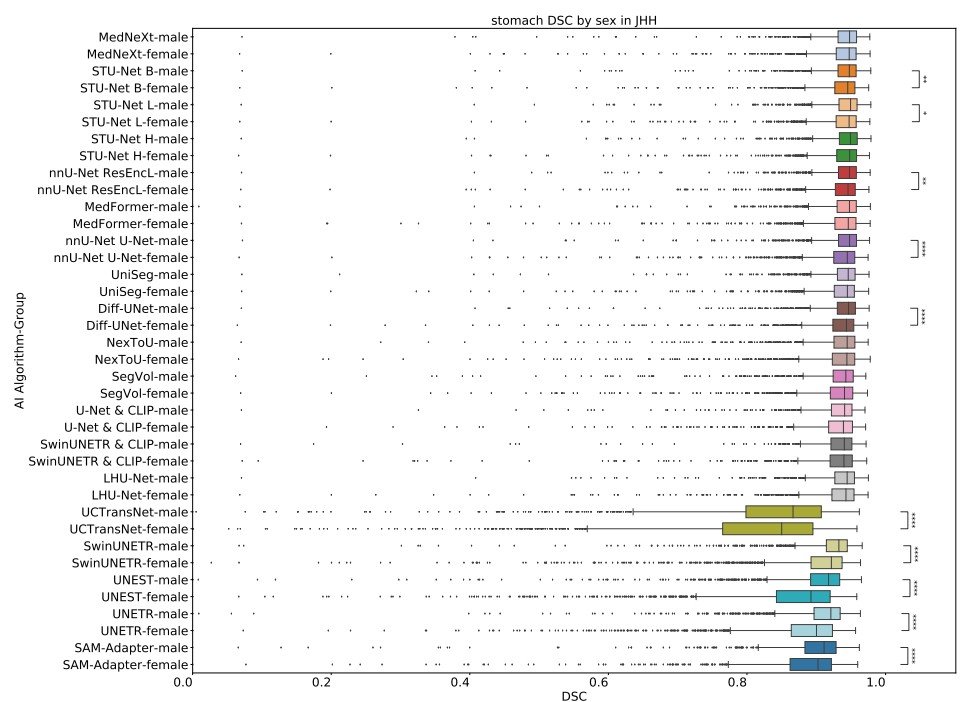

Figure 53: **Boxplot showing stomach DSC score by sex in JHH.** Statistical significance is indicated by stars: * p < 0.05, ** p <0.01, *** p < 0.001, **** p < 0.0001. We perform Kruskal–Wallis tests followed by post-hoc Mann-Whitney U Tests with Bonferroni correction. Here, we did not perform statistical comparisons between diverse AI algorithms.

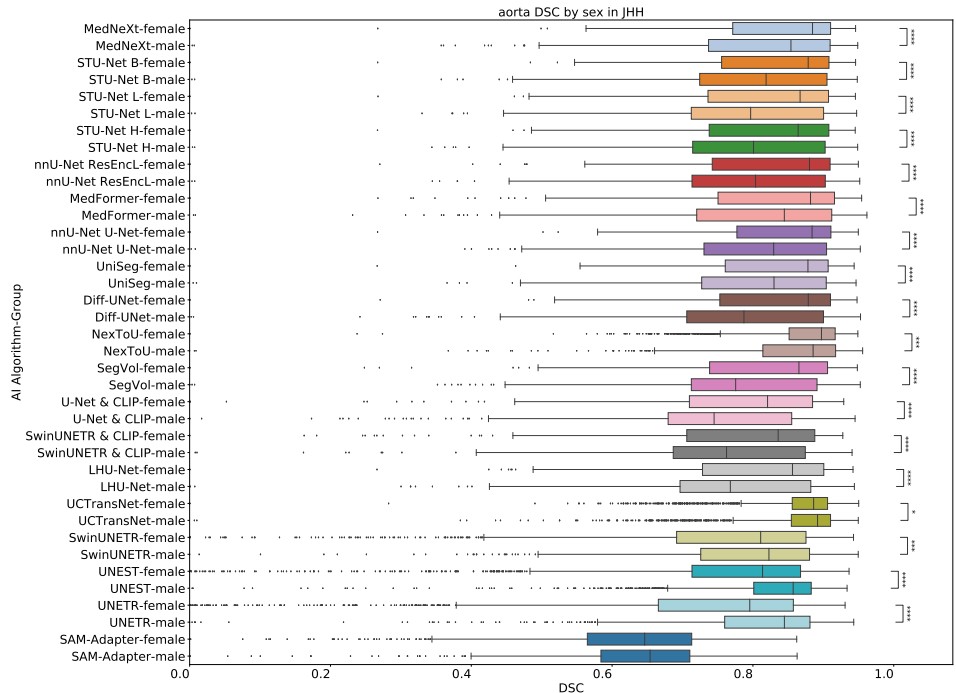

Figure 54: **Boxplot showing aorta DSC score by sex in JHH.** Statistical significance is indicated by stars: * p < 0.05, ** p <0.01, *** p < 0.001, **** p < 0.0001. We perform Kruskal–Wallis tests followed by post-hoc Mann-Whitney U Tests with Bonferroni correction. Here, we did not perform statistical comparisons between diverse AI algorithms.

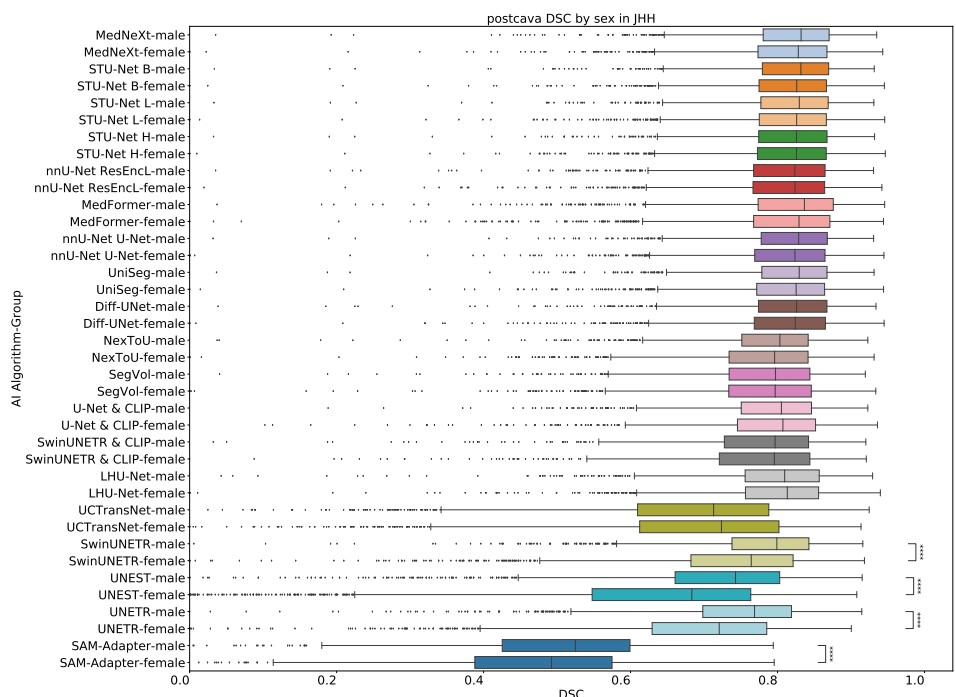

Figure 55: **Boxplot showing postcava DSC score by sex in JHH.** Statistical significance is indicated by stars: * p < 0.05, ** p <0.01, *** p < 0.001, **** p < 0.0001. We perform Kruskal–Wallis tests followed by post-hoc Mann-Whitney U Tests with Bonferroni correction. Here, we did not perform statistical comparisons between diverse AI algorithms.

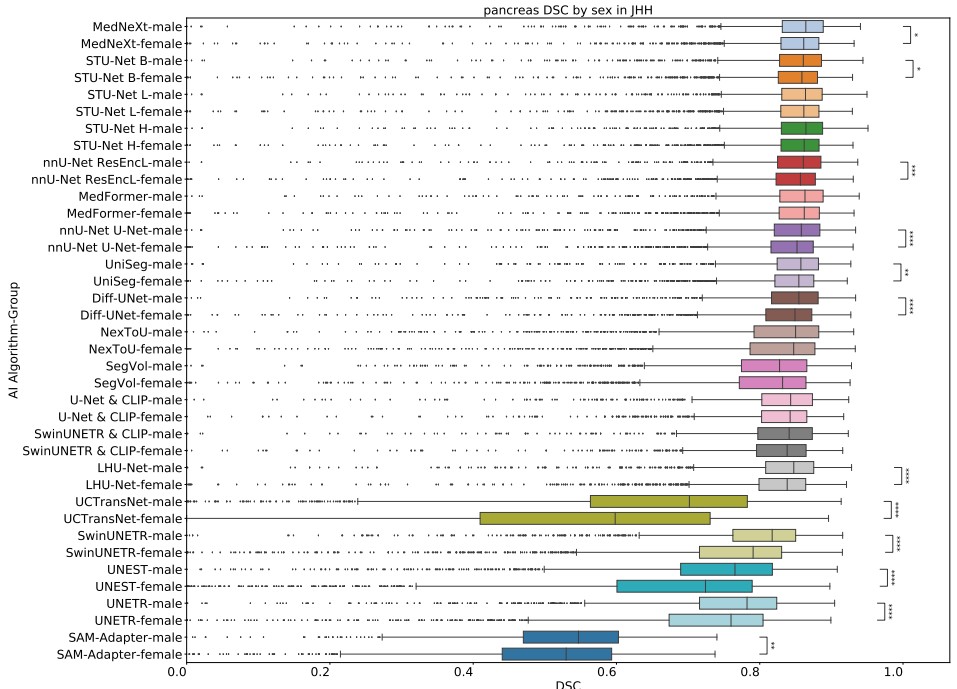

Figure 56: **Boxplot showing pancreas DSC score by sex in JHH.** Statistical significance is indicated by stars: * p < 0.05, ** p <0.01, *** p < 0.001, **** p < 0.0001. We perform Kruskal–Wallis tests followed by post-hoc Mann-Whitney U Tests with Bonferroni correction. Here, we did not perform statistical comparisons between diverse AI algorithms.

## D.5.10 Race: per-class analysis

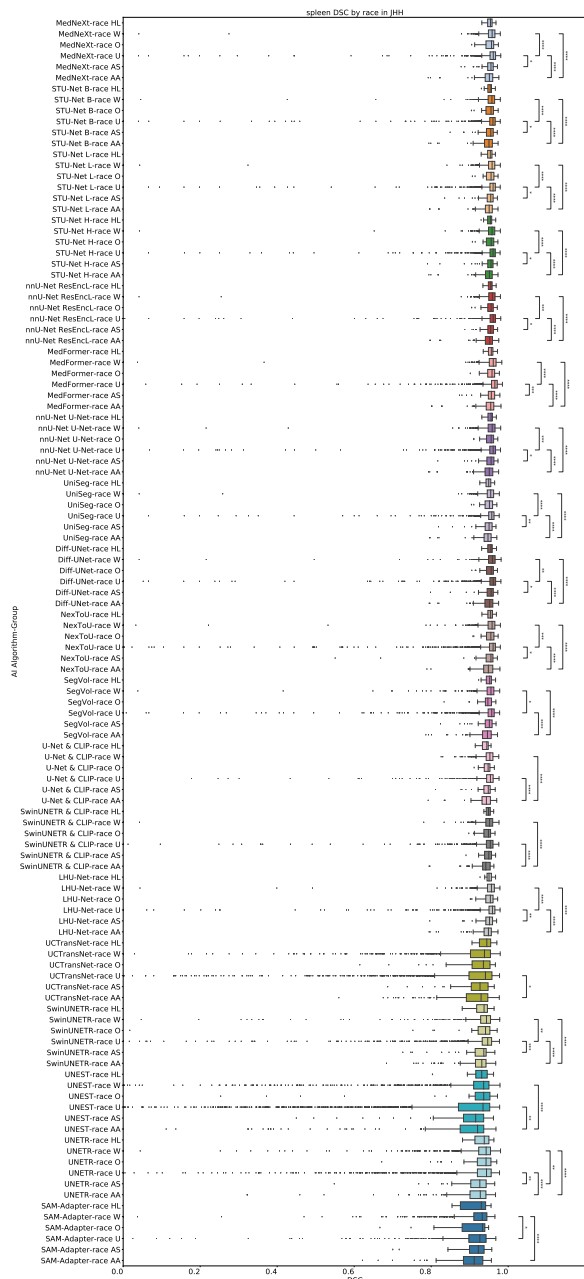

Figure 57: **Boxplot showing spleen DSC score by race in JHH.** Statistical significance is indicated by stars: * p < 0.05, ** p <0.01, *** p < 0.001, **** p < 0.0001. We perform Kruskal–Wallis tests followed by post-hoc Mann-Whitney U Tests with Bonferroni correction. Here, we did not perform statistical comparisons between diverse AI algorithms.

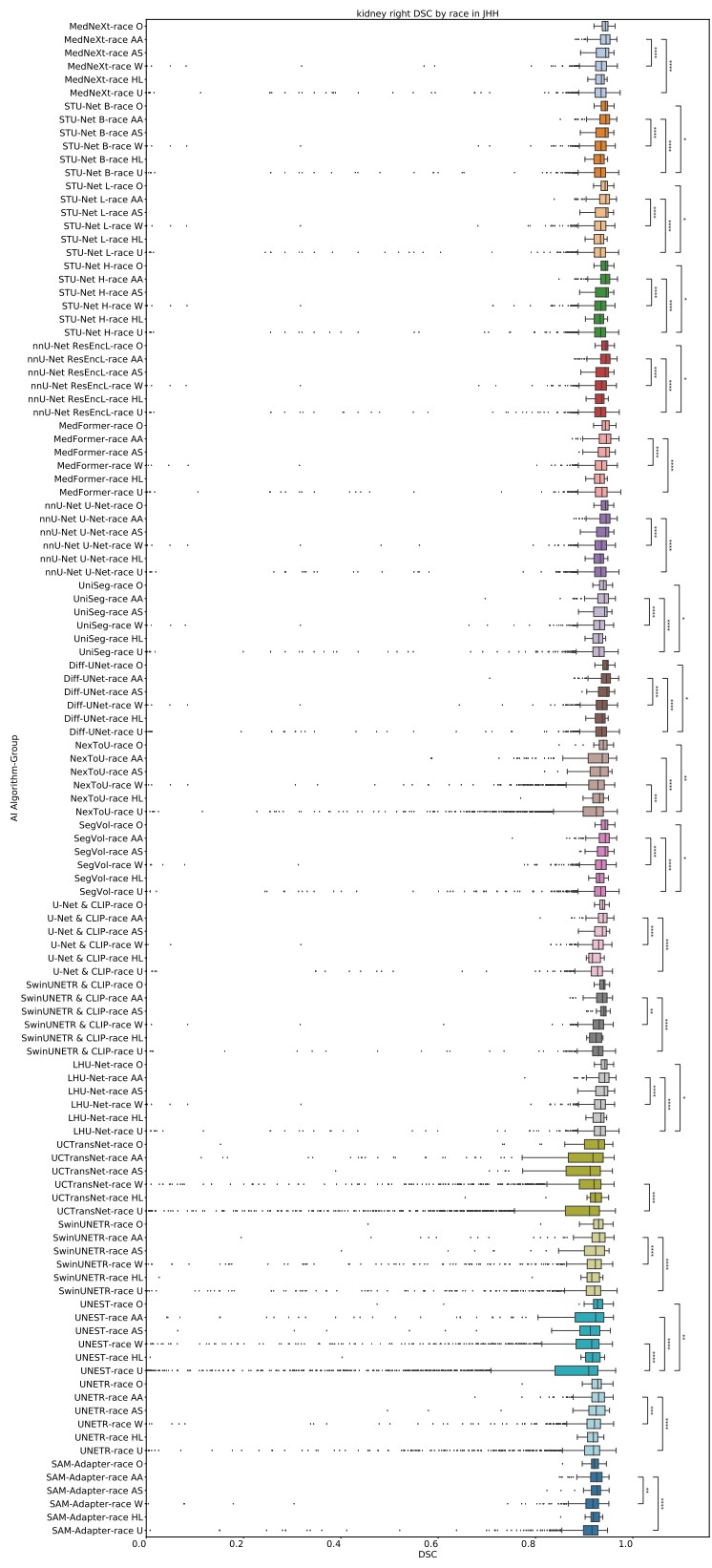

Figure 58: **Boxplot showing right kidney DSC score by race in JHH.** Statistical significance is indicated by stars: * p < 0.05, ** p <0.01, *** p < 0.001, **** p < 0.0001. We perform Kruskal–Wallis tests followed by post-hoc Mann-Whitney U Tests with Bonferroni correction. Here, we did not perform statistical comparisons between diverse AI algorithms.

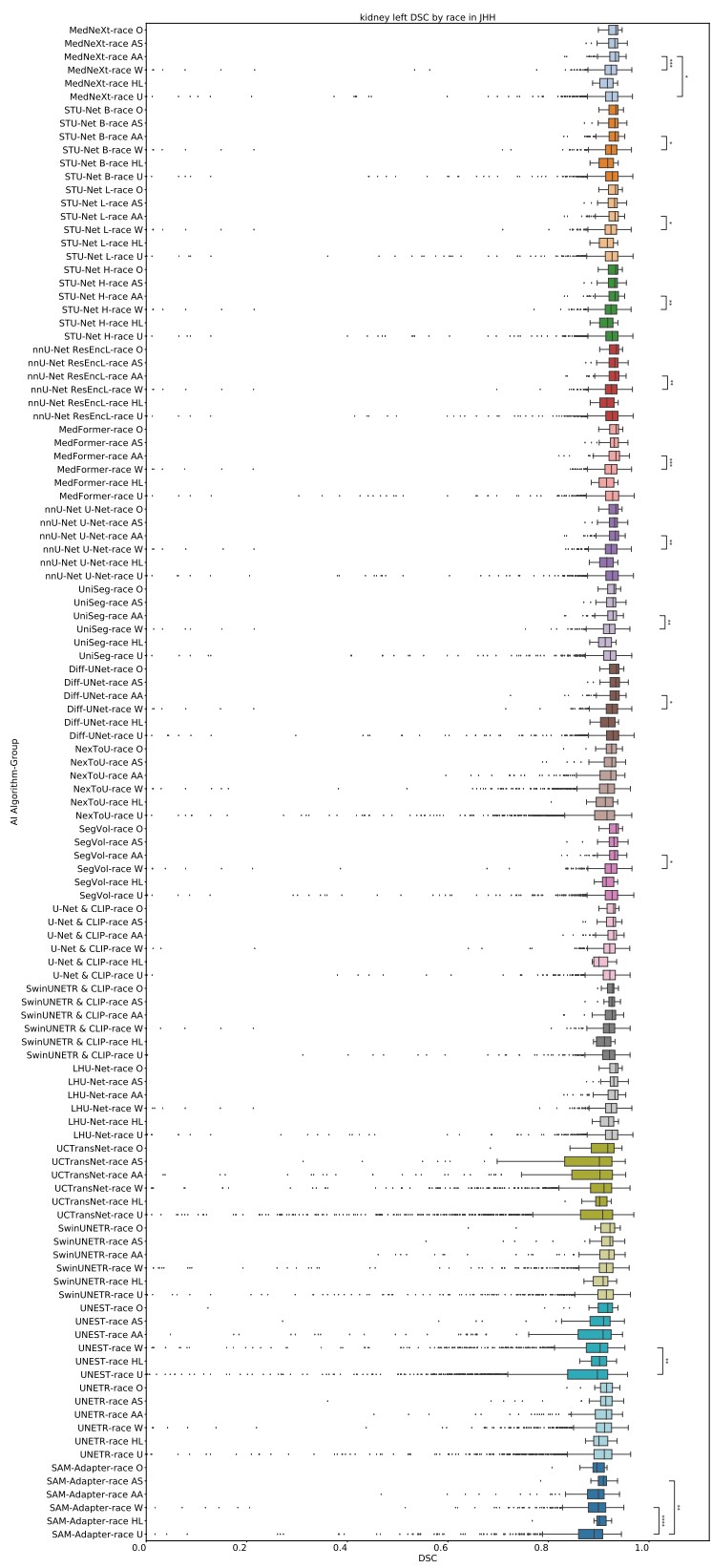

Figure 59: **Boxplot showing left kidney DSC score by race in JHH.** Statistical significance is indicated by stars: * p < 0.05, ** p <0.01, *** p < 0.001, **** p < 0.0001. We perform Kruskal–Wallis tests followed by post-hoc Mann-Whitney U Tests with Bonferroni correction. Here, we did not perform statistical comparisons between diverse AI algorithms.

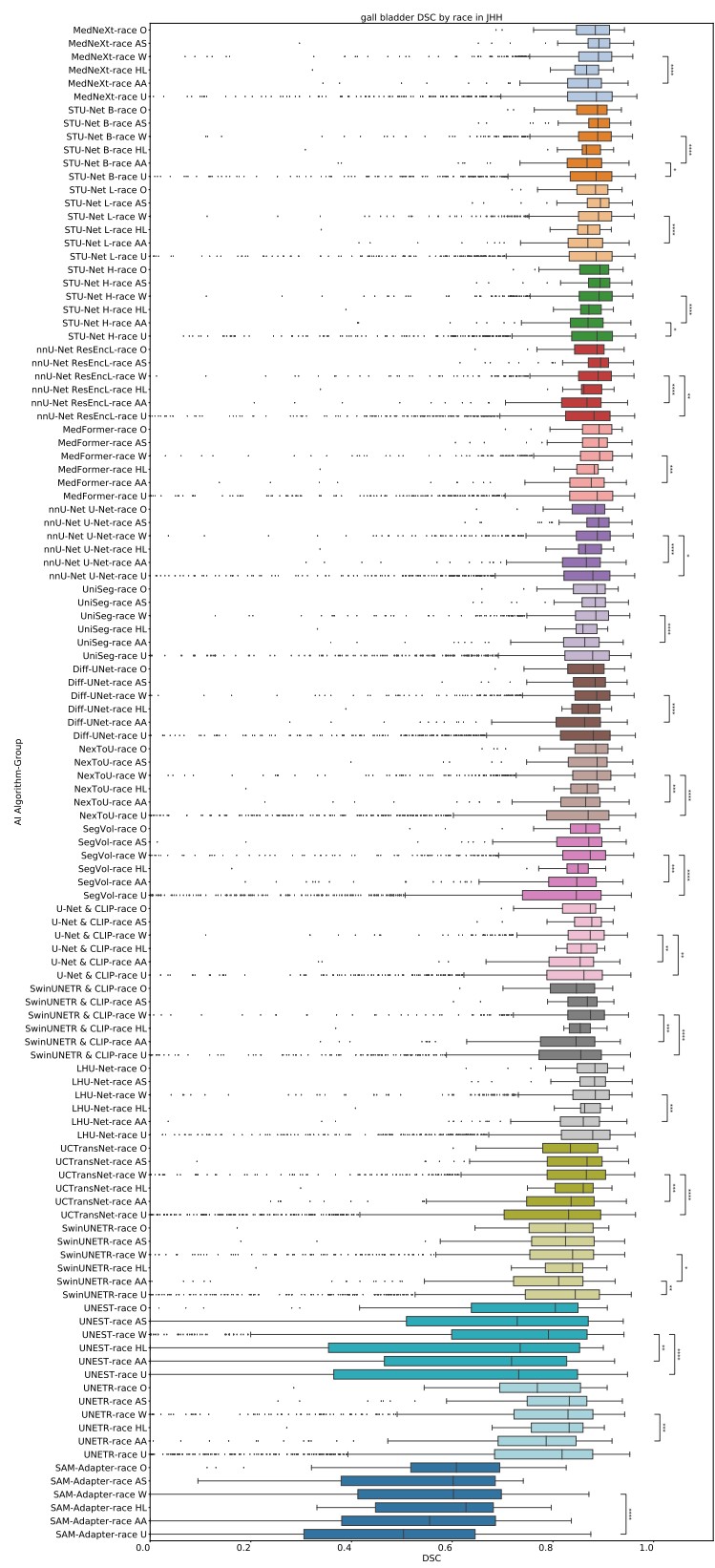

Figure 60: **Boxplot showing gallbladder DSC score by race in JHH.** Statistical significance is indicated by stars: * p < 0.05, ** p < 0.01, *** p < 0.001, **** p < 0.0001. We perform Kruskal–Wallis tests followed by post-hoc Mann-Whitney U Tests with Bonferroni correction. Here, we did not perform statistical comparisons between diverse AI algorithms.

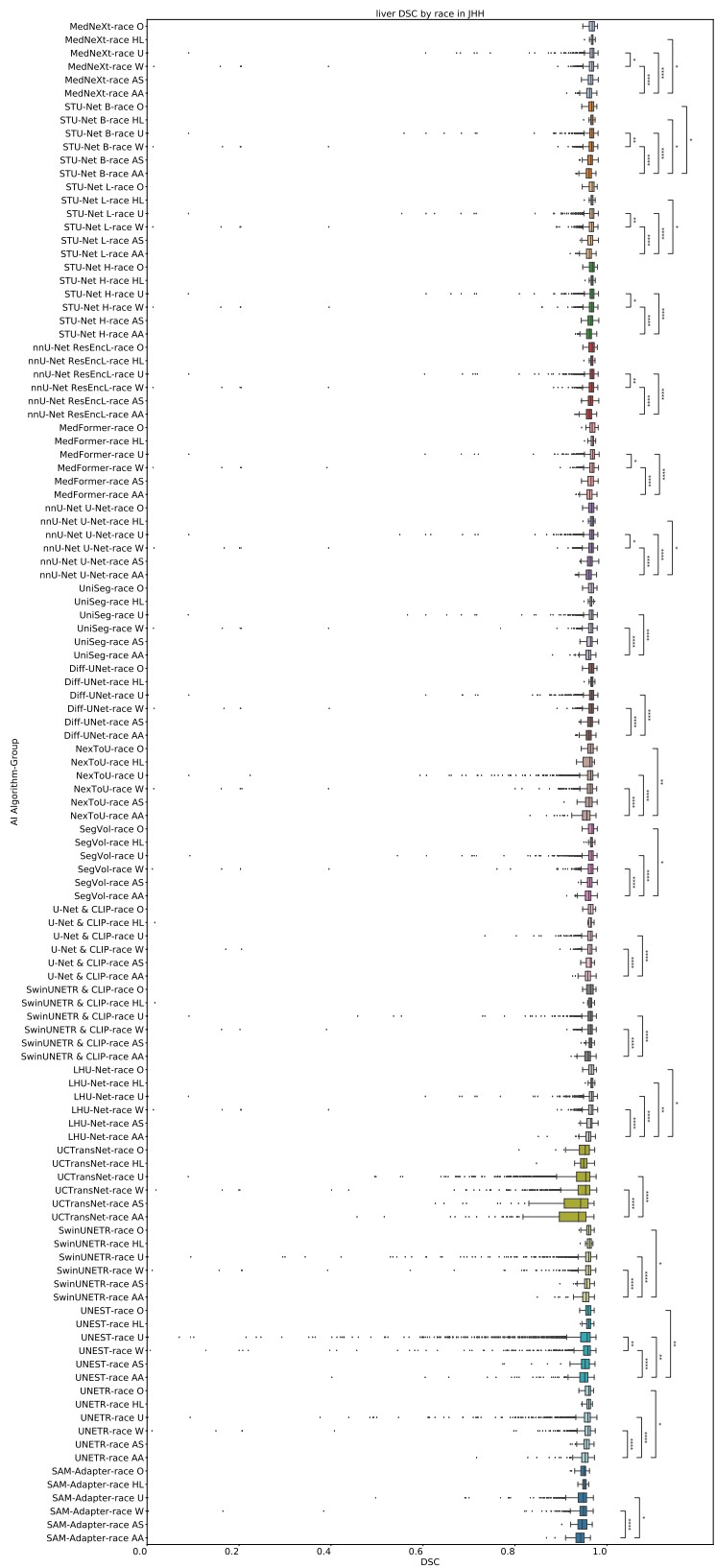

Figure 61: **Boxplot showing liver DSC score by race in JHH.** Statistical significance is indicated by stars: * p < 0.05, ** p <0.01, *** p < 0.001, **** p < 0.0001. We perform Kruskal–Wallis tests followed by post-hoc Mann-Whitney U Tests with Bonferroni correction. Here, we did not perform statistical comparisons between diverse AI algorithms.

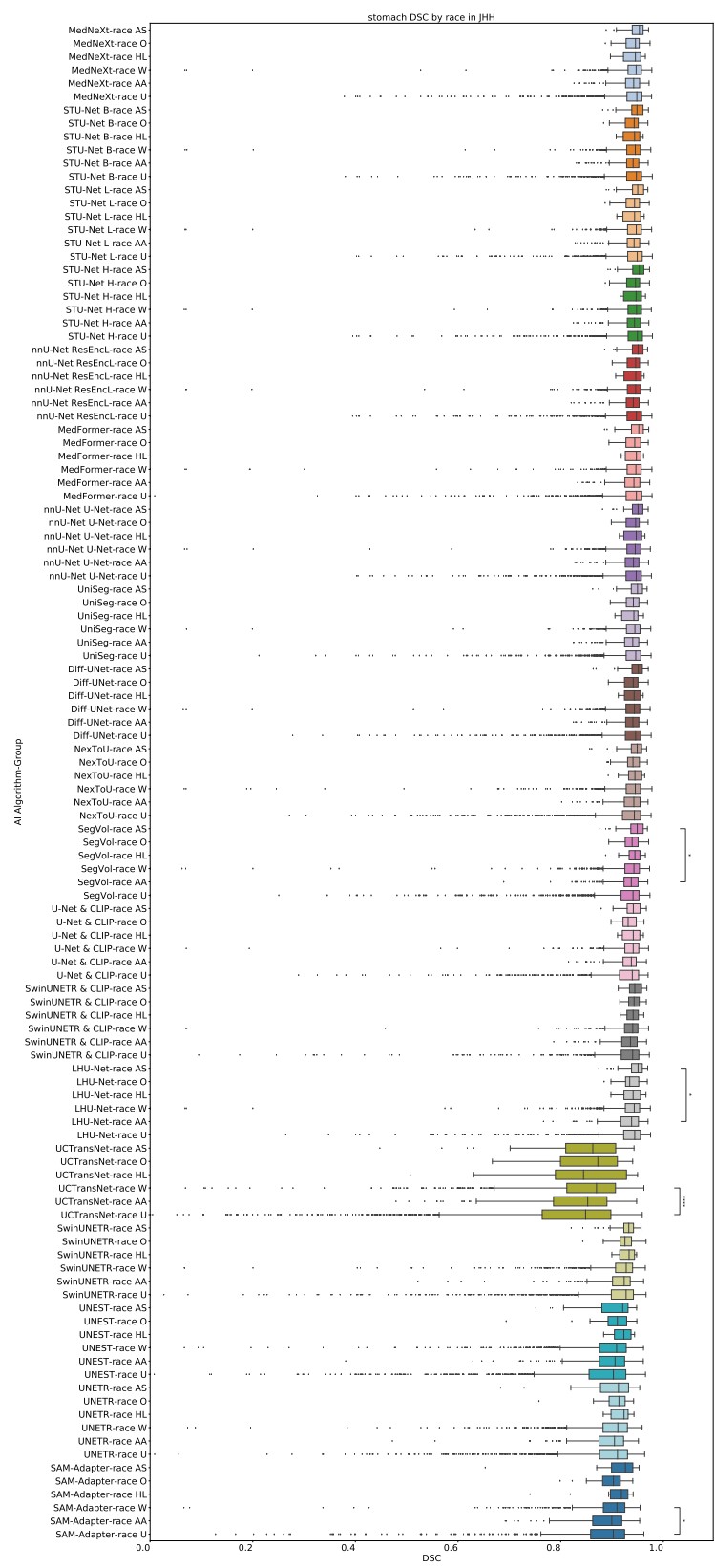

Figure 62: **Boxplot showing stomach DSC score by race in JHH.** Statistical significance is indicated by stars: * p < 0.05, ** p <0.01, *** p < 0.001, **** p < 0.0001. We perform Kruskal–Wallis tests followed by post-hoc Mann-Whitney U Tests with Bonferroni correction. Here, we did not perform statistical comparisons between diverse AI algorithms.

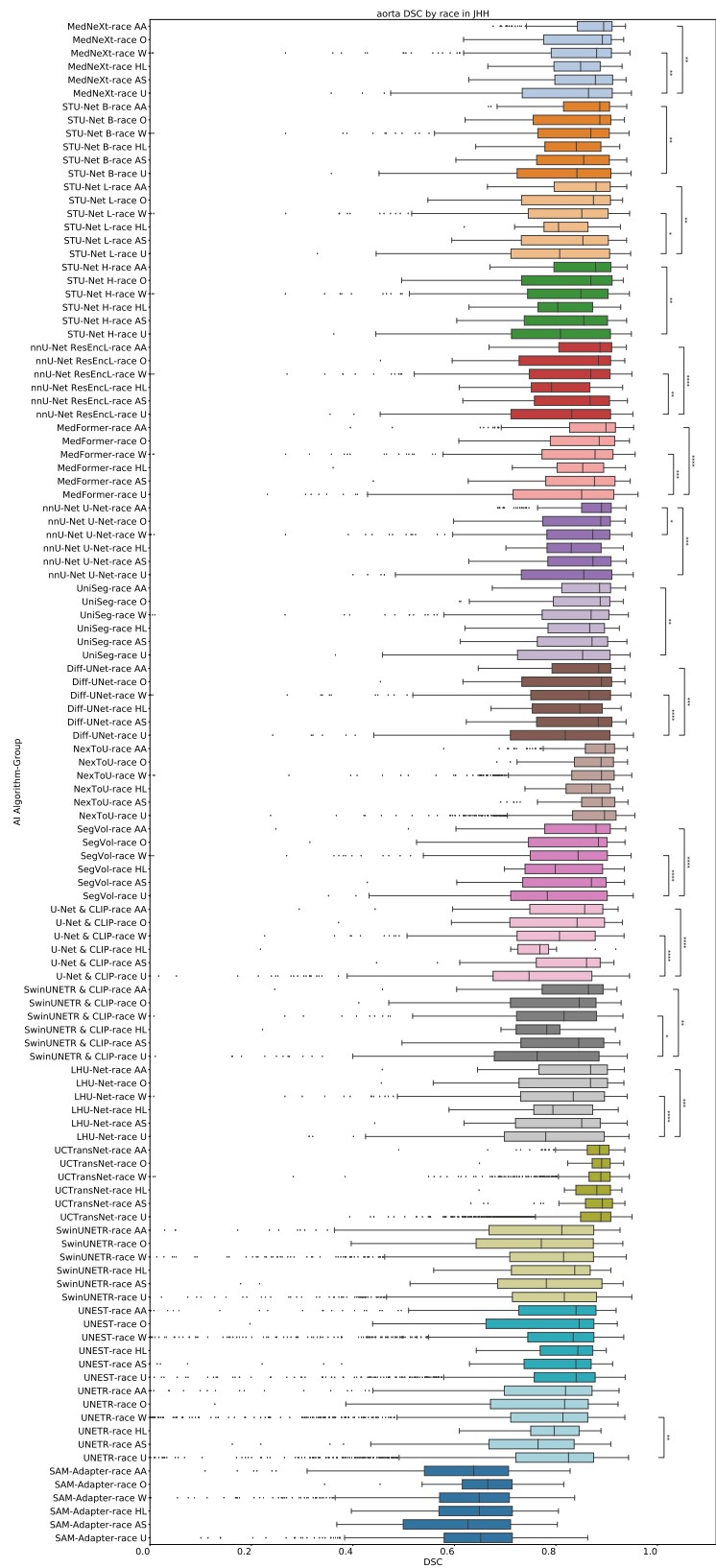

Figure 63: **Boxplot showing aorta DSC score by race in JHH.** Statistical significance is indicated by stars: * p < 0.05, ** p <0.01, *** p < 0.001, **** p < 0.0001. We perform Kruskal–Wallis tests followed by post-hoc Mann-Whitney U Tests with Bonferroni correction. Here, we did not perform statistical comparisons between diverse AI algorithms.

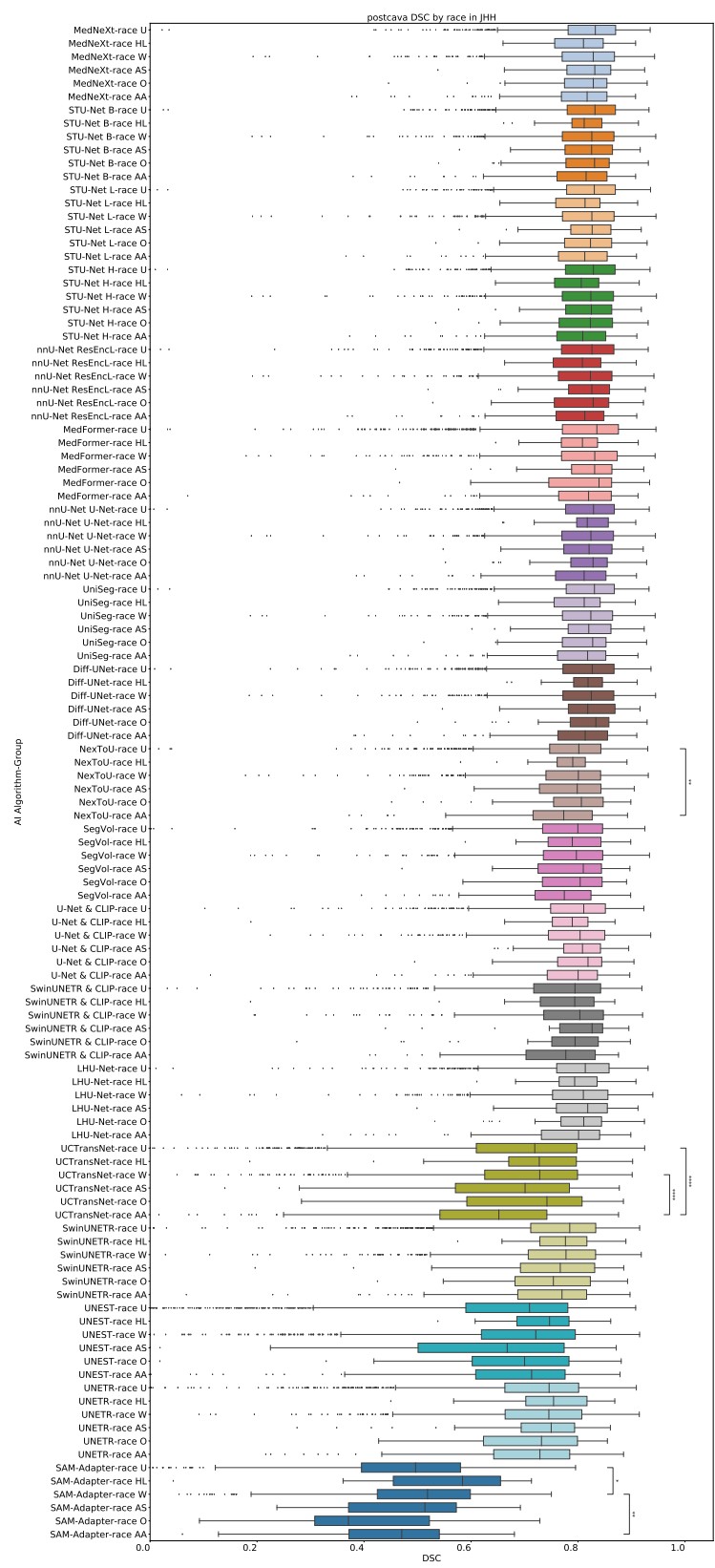

Figure 64: **Boxplot showing postcava DSC score by race in JHH.** Statistical significance is indicated by stars: * p < 0.05, ** p <0.01, *** p < 0.001, **** p < 0.0001. We perform Kruskal–Wallis tests followed by post-hoc Mann-Whitney U Tests with Bonferroni correction. Here, we did not perform statistical comparisons between diverse AI algorithms.

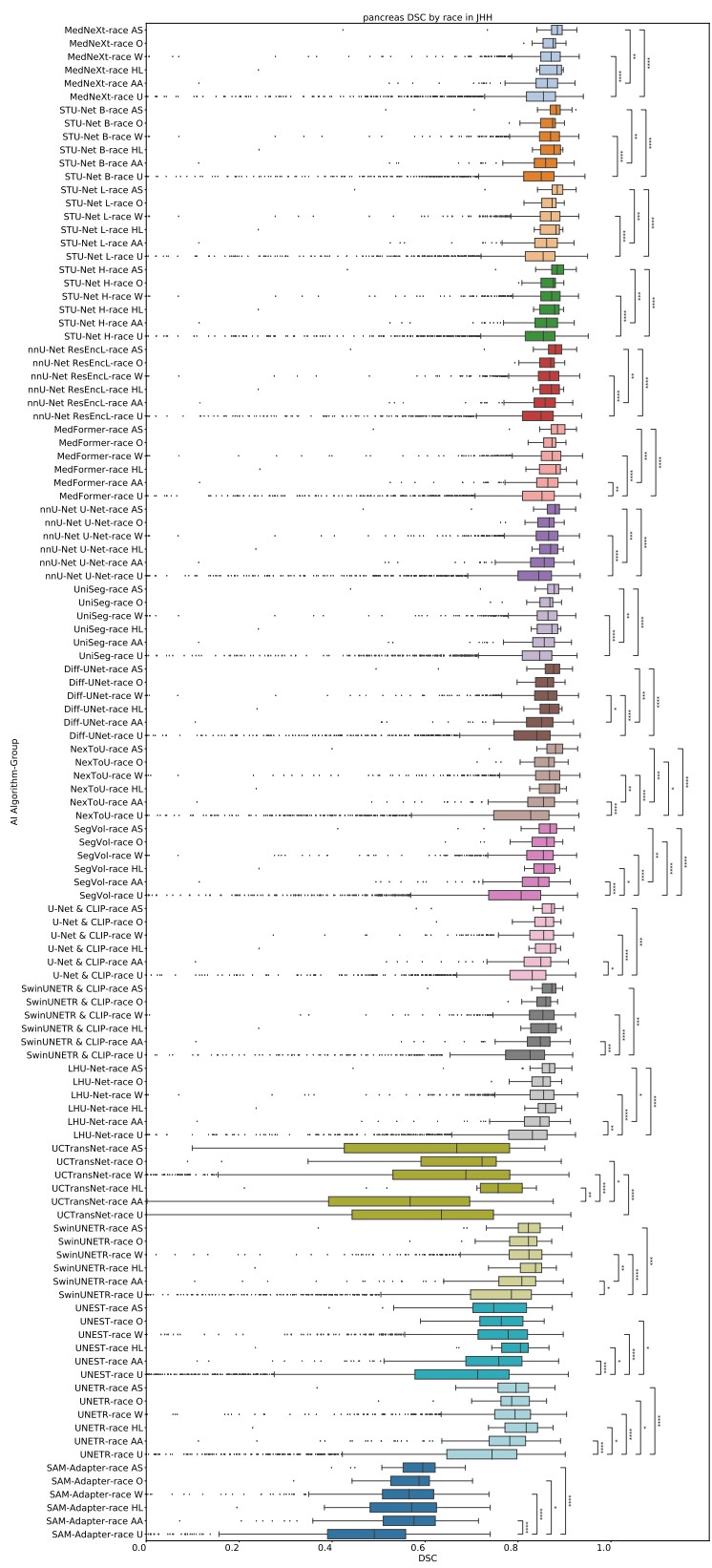

Figure 65: **Boxplot showing pancreas DSC score by race in JHH.** Statistical significance is indicated by stars: * p < 0.05, ** p <0.01, *** p < 0.001, **** p < 0.0001. We perform Kruskal–Wallis tests followed by post-hoc Mann-Whitney U Tests with Bonferroni correction. Here, we did not perform statistical comparisons between diverse AI algorithms.

# E   On Label Noise

AbdomenAtlas 1.0 is an amalgamation of 16 public datasets (Appendix A.1), which, when combined together, resulted in a partially labeled dataset. Radiologists, assisted by AI, provided all the missing labels for 9 anatomical structures, making the dataset fully-labeled [60]. When creating AbdomenAtlas 1.0 we did not revise the labels that were already provided in the public datasets. However, upon future visual inspection, we found that these public datasets may share inconsistent annotation standards, also reported in Liu et al. [47]. For example, the aorta annotation standard is inconsistent in AbdomenCT-12organ and other datasets: part of the upper aorta region is missing in AbdomenCT-12organ, while the aorta annotation is complete in BTCV and AMOS. Moreover, since the public datasets that constitute AbdomenAtlas 1.0 contained both automatic and manual labels, they can also portray human and AI errors.

To address this, we developed an automatic label quality checking tool, based on anatomical priors (e.g., expected shape of organs), to detect and correct noisy labels. This tool indicated that aorta concentrated most of the label noise in AbdomenAtlas 1.0. It has 32.4% of noisy labels, which are mostly the aforementioned incomplete annotations. The second structure with the highest amount of detected errors was the kidneys, but its percentage of noisy labels was much lower: 2.6%. Our tool detected less than 1% of error in other classes. Therefore, the detected errors are mostly concentrated on one of the 9 annotated structures. Moreover, since AbdomenAtlas 1.0 carried the errors and annotation standard inconsistencies found in public datasets, the noise in AbdomenAtlas 1.0 labels represents common annotation errors and inconsistencies. Conversely, studies on AI robustness to label noise commonly rely on artificially generated noise [76]. Thus, we viewed the realistic and quantifiable noise in AbdomenAtlas 1.0 as an opportunity to perform a realistic study on AI robustness to label noise. To further increase the study's realism, we simulate the standard scenario where researchers are unaware of the noise: we did not inform the AI creators about the annotation errors in AbdomenAtlas 1.0 prior to model training. This approach avoided uneven label corrections by only some teams and ensured that the AI algorithms in this benchmark accurately represent the realistic scenario of AI trained on public data with common label noise, without creators actively trying to counteract the noise.

To assess AI robustness to label noise, the algorithms must be tested on datasets whose labels are less noisy than those in the training data. The JHH test set ($N$=5,160) was entirely annotated by radiologists, manually and following a well-defined annotation standard, over 5 years [59]. Thus, it serves as a gold standard for low label noise. Touchstone leverages this large-scale, high-quality test dataset to verify whether AI trained noisy labels, representative of current public datasets, performs well when evaluated with high-quality manual labels. Since TotalSegmentator is not composed of multiple datasets, their annotation standards are consistent, and we detected low levels (<1%) of label noise on them. Thus, they are also adequate for evaluating AI's robustness. Additionally, to better quantify the impact of label noise on AI accuracy, we re-trained ResEncL on AbdomenAtlas 1.0C. This dataset, which we publicly released, is a revised version of AbdomenAtlas 1.0, where labels were improved by radiologists assisted by AI and by our error detection tool. The aorta was the only class where the nnU-Net had large and significant performance increments (e.g., 10.35% DSC improvement in TotalSegmentator). For other structures, improvements are mostly not significant and low, demonstrating that the AI algorithm is robust to moderate levels of label noise (e.g., less than 3% of noisy labels according to our detection tool), but not to excessive noise. The continuous improvement of label noise detection and annotation quality, unifying annotation standards and correcting public datasets' flawed labels, is a continuous commitment of Touchstone.

# F Full Affiliation List

[1] Department of Computer Science, Johns Hopkins University
[2] Department of Pharmacy and Biotechnology, University of Bologna
[3] Center for Biomolecular Nanotechnologies, Istituto Italiano di Tecnologia
[4] NVIDIA
[5] Division of Medical Image Computing, German Cancer Research Center (DKFZ)
[6] Helmholtz Imaging, German Cancer Research Center (DKFZ)
[7] ESAT-PSI, KU Leuven
[8] Faculty of Mathematics and Computer Science, Heidelberg University
[9] HIDSS4Health - Helmholtz Information and Data Science School for Health
[10] Shanghai Jiao Tong University
[11] Shanghai Artificial Intelligence Laboratory
[12] Pattern Analysis and Learning Group, Department of Radiation Oncology, Heidelberg University Hospital
[13] Interactive Machine Learning Group (IML), DKFZ
[14] School of Computer Science and Engineering, Northwestern Polytechnical University
[15] Australian Institute for Machine Learning, The University of Adelaide
[16] College of Computer Science and Technology, Zhejiang University
[17] Hong Kong University of Science and Technology (Guangzhou)
[18] Hong Kong University of Science and Technology
[19] Faculty of Informatics and Data Science, University of Regensburg
[20] Faculty of Electrical Engineering and Information Technology, RWTH Aachen University
[21] Fraunhofer Institute for Digital Medicine MEVIS
[22] Electronic & Information Engineering School, Harbin Institute of Technology (Shenzhen)
[23] Beijing Academy of Artificial Intelligence (BAAI)
[24] The Chinese University of Hong Kong
[25] Peking University
[26] Department of Electrical and Computer Engineering, Duke University
[27] Stony Brook University
[28] Department of Computer Science and Engineering, Department of Chemical and Biological Engineering and Division of Life Science, Hong Kong University of Science and Technology
[29] Data Science and Computation Facility, Fondazione Istituto Italiano di Tecnologia
[30] Ecole Polytechnique Fédérale de Lausanne

# G    Potential Negative Societal Impacts

Potential negative societal impacts of benchmarking AI algorithms for medical image segmentation include reinforcing biases, compromising data privacy, and leading to misuse of AI systems. Standard benchmarks may suffer from in-distribution biases, small test sets, oversimplified metrics, and short-term outcome pressures, which can result in AI models that perform well on benchmarks but fail in real-world applications. These issues can undermine the reliability, fairness, and generalizability of AI systems in medical contexts, potentially causing harm and reducing trust in AI-driven healthcare solutions.