# OpenReview forum: "Touchstone Benchmark: Are We on the Right Way for Evaluating AI Algorithms for Medical Segmentation?"
_NeurIPS.cc/2024/Datasets_and_Benchmarks_Track — NeurIPS 2024 Track Datasets and Benchmarks Poster_

### Official Review · Reviewer_68Sv · 2024-06-21
**Good paper, but some claims overstated**

**Rating:** 5
**Confidence:** 4
**Clarity:** The paper is well-written and coherent.

**Review:**

Overall, this is a strong paper that makes important contributions to medical AI evaluation. The dataset and benchmark will likely be impactful and widely used. The analysis provides valuable insights, though could go deeper in some areas. The main limitation is the focus on a single task (anatomical segmentation), but the authors indicate plans to expand this in future work. While these contributions are valuable, some of the claims in the paper seem overstated and need to be qualified:

1) The authors assert that their benchmark is "fair" and "widely-adopted," but it's unclear how widely adopted it is at this stage, given it's a new benchmark.
2) The claim of "unprecedented scale" is strong, especially considering recent large-scale medical imaging datasets in other domains.
3) While the OOD evaluation is a strength, the paper doesn't fully explore the implications or limitations of this approach.
The authors criticize existing benchmarks for oversimplified metrics, but still rely heavily on standard metrics like Dice score.
4) The analysis of different model architectures and their performance is somewhat surface-level, missing an opportunity for deeper insights.

Despite these criticisms, the paper presents a significant effort in improving medical image segmentation benchmarks. The large-scale dataset and comprehensive evaluation provide valuable resources and insights for the field. However, the authors could have been more measured in some of their claims and provided deeper analysis in several areas.

**Strengths:**

1) The scale and diversity of the dataset is impressive and addresses a key limitation of many existing medical AI benchmarks. The large out-of-distribution test set allows for more statistically robust evaluation.
2) The comprehensive evaluation of many top segmentation algorithms, implemented by their original authors, provides valuable insights into the current state-of-the-art.
3) The analysis of performance across different subgroups (age, sex, scanner type, etc.) is important for understanding potential biases and limitations.
4) The long-term commitment to maintaining and expanding the benchmark is commendable and important for the field.

**Additional Feedback:**

None.

**Correctness:**

Some claims are overstated and it is recommended it other sections that these statements are qualified.

**Documentation:**

No concerns.

**Ethics:**

I presume that all of the data is fully de-identified such that it is HIPAA-compliant or follows NEMA's recommendations, since this is being released as a publicly available dataset, but would like the authors to confirm this.

**Limitations:**

Limitations already highlighted in the opportunities for improvement section.

**Opportunities For Improvement:**

To further strengthen the paper and address potential concerns, the authors could consider addressing the following questions in their rebuttal:

1) Metric diversity. While you criticize existing benchmarks for oversimplified metrics, your analysis still relies heavily on Dice scores. Could you elaborate on why you chose not to include more advanced or task-specific metrics? How might the inclusion of such metrics enhance the benchmark's value?
2) Architectural insights. The comparison between different model architectures (e.g., CNNs vs. Transformers) is relatively surface-level. Can you provide deeper insights into why certain architectures perform better on specific tasks or anatomical structures?
3) Clinical relevance. The benchmark focuses solely on anatomical segmentation. How do you plan to incorporate more clinically relevant tasks, such as pathology detection or severity grading, in future iterations?
4) Error analysis. The paper would benefit from a more thorough error analysis. Can you provide insights into common failure modes across algorithms? Are there specific types of images or anatomical variations that consistently challenge all models?
5) Longitudinal evaluation. You mention a commitment to long-term benchmarking. Can you elaborate on your plans for maintaining and updating the benchmark over time? How will you ensure its continued relevance as the field evolves?
6)

**Relation To Prior Work:**

The paper clearly outlines and cites how it builds upon and relates to prior work in the field of medical segmentation.

**Summary And Contributions:**

The authors introduce Touchstone which is a large-scale benchmark for evaluating AI algorithms in medical image segmentation, specifically focusing on abdominal CT volumes. The authors claim to address several limitations of existing benchmarks, including small test sets, in-distribution evaluation, and oversimplified metrics. The four main contributions are:

1) A large and diverse dataset. 5,195 CT volumes from 76 hospitals for training, and 6,933 CT volumes from 8 additional hospitals for testing. This scale is indeed impressive and addresses the need for more comprehensive evaluation data.
2) Out-of-distribution (OOD) evaluation. The test set comes from different hospitals than the training set, potentially providing a more realistic assessment of algorithm generalization.
3) Comprehensive evaluation of 16 state-of-the-art segmentation algorithms, implemented by their original authors, on this dataset.
4) Analysis of algorithm performance across different demographic groups and imaging characteristics.

---

> ### Author Rebuttal · Authors · 2024-08-18
>
> # Responses to Reviewer 68Sv (Part I)
>
> We are grateful for your recognition of the scientific significance of our paper, *“scale and diversity of the dataset is impressive”*, *“more realistic assessment of algorithm generalization”*, *“comprehensive evaluation”*, *“valuable insights”*, *“long-term commitment … is commendable and important”* and *“likely be impactful and widely used”*. We also appreciate your constructive suggestions and our response to your specific questions follows.
>
> ---
>
> > **Q1.** *The authors assert that their benchmark is "fair" and "widely-adopted," but it's unclear how widely adopted it is at this stage, given it's a new benchmark.*
>
> - We stated in `Sec. 1` that Touchstone Benchmark is *“an effort **towards** a fair, large-scale, and widely-adopted medical AI benchmark.”* Being fair and widely-adopted is our goal.
>
> - Thanks for your comment. We have now made it clearer to avoid potential confusion. The revised sentence is *“To address this AI mismeasurement issue, we present the Touchstone benchmark, which aims for a fair, large-scale, and widely-adopted medical AI benchmark.”*
>
> ---
>
> > **Q2.** *The claim of "unprecedented scale" is strong, especially considering recent large-scale medical imaging datasets in other domains.*
>
> - We agree that the word "unprecedented” scale could be overly strong. Now, it has been revised as “This benchmark is based on annotated CT datasets of **considerable** scale, …”
>
> - There are increasing medical imaging datasets being publicly accessible, many of which are of considerable scale. For comparison, we have summarized some of the most representative datasets in different medical domains in the table below, and included it as a new subsection in our supplementary. We found that many large-scale datasets in other domains contain only **image-wise annotations** while our Touchstone provides **pixel-wise annotations** for testing. The workload of creating pixel-wise annotations is much higher than image-wise annotations. For example, our JHH dataset took **five years** for a team of radiologists and medical trainees to annotate, details were documented in [[Park et al., 2020](https://www.sciencedirect.com/science/article/pii/S2211568419301391)].
>
> | dataset domain | dataset name | # of 2D images | # of annotated classes | annotation level |
> |:---------------|:---------------------|:--------------------:|:--------------------:|:--------------------|
> | Chest | COVID-19 CHEST X-RAY DATABASE | 3,886 | 3 | image-wise |
> | Chest | Chest X-ray PD Dataset | 4,575 | 3 | image-wise |
> | Chest | MedFM ChestDR 2023 | 4,848 | 19 | image-wise |
> | Microscopy | LC25000 | 25,000 | 5 | image-wise |
> | Microscopy | Bone Marrow Cytomorphology | 171,375 | 21 | image-wise |
> | Microscopy | PatchCamelyon | 327,680 | 2 | image-wise |
> | Dermatology | ISIC 2017 | 2,750 | 1 | pixel-wise |
> | Dermatology | ISIC 2020 | 33,126 | 2 | image-wise |
> | Ophthalmology | Retinal OCT-C8 | 24,000 | 8 | image-wise |
> | Ophthalmology | Eyepacs | 35,126 | 5 | image-wise |
> | Ophthalmology | AIROGS | 101,442 | 2 | image-wise |
> | Abdomen | **Touchstone (ours)** | **4,435,918** | **9** | **pixel-wise** |
>
> ---
>
> > **Q3.** *While the OOD evaluation is a strength, the paper doesn't fully explore the implications or limitations of this approach.*
>
> - We have discussed the limitations of the OOD evaluation but they are presented in several pieces of the main paper and supplementary. Following your suggestion, we have now explicitly itemized two limitations of OOD evaluation in Sec. 4 as follows:
>
>   - **Limitation 1**: *“The fact that JHH is a private dataset has both advantages and disadvantages…”* (see `Sec. 4`). The test set for the OOD benchmark, including both images and annotations, must be kept private (see our general responses). Publicly exposed test data can be integrated into public datasets, resulting in multiple models that overfit it and undermine the integrity of the OOD evaluation [[Geirhos et al., 2020](https://www.nature.com/articles/s42256-020-00257-z), [Recht et al., 2019](https://proceedings.mlr.press/v97/recht19a/recht19a.pdf)].
>
>   - **Limitation 2**: *“AI performance can vary significantly across different datasets, with per-class differences of 10–20% common, and up to 80% observed; thus, out-of-distribution evaluation across multiple hospitals is crucial for ensuring AI’s reliability and clinical adoption.* (see `Sec. 4`). A limitation of OOD evaluation is that testing on one or few OOD datasets cannot ensure good generalization to all possible real-world clinical settings. The more OOD test sets, the better [[Saenz et al., The Lancet Digital Health 2024](https://www.thelancet.com/journals/landig/article/PIIS2589-7500(23)00222-4/fulltext)].  Here, we evaluated on three OOD datasets, and we are establishing collaborations to increase the number of OOD test sets in the future editions of Touchstone.

---

> ### Author Rebuttal · Authors · 2024-08-18
>
> # Responses to Reviewer 68Sv (Part II)
>
>
> > **Q4.** *Metric diversity. While you criticize existing benchmarks for oversimplified metrics, your analysis still relies heavily on Dice scores. Could you elaborate on why you chose not to include more advanced or task-specific metrics? How might the inclusion of such metrics enhance the benchmark's value?*
>
> - Thanks for your comment. There is miscommunication on the meaning of oversimplified metrics and we have now made it clearer in the paper. We are **not criticizing** the use of Dice score, but the use of “average” Dice over classes, ages, gender, populations, scanners, pathology, etc. We referred to this average performance as “simplified metric” in `Sec. 1`, which cannot faithfully reveal the performance of AI algorithms.
>
> - Our paper has presented diverse and in-depth analyses including the performance of each AI algorithm across diverse demographic groups such as race, age, sex, pathology, and scanner (`Figures 25-72`), NSD scores (`Tables 9, 11, 13, 15` and `Figures 8-11, 16, 18, 20, 22, 24`), worst-case performance (`Figures 8-11`), Demographic Parity Difference—a metric that captures bias across diverse demographic groups (`Figures 8-11`), inference time/memory (`Table 5`), training computational cost (`Table 6`), ranking stability (`Figure 12`), pairwise statistical tests (`Figures 13-16`), etc. The analyses’ conclusions are summarized in the main paper, but, due to space limitations in the main text, our 80-page supplementary presented a better picture of the depth of our analyses. All the results are now made available at our [GitHub](https://github.com/MrGiovanni/Touchstone).
>
> ---
>
> > **Q5.** *Architectural insights. The comparison between different model architectures (e.g., CNNs vs. Transformers) is relatively surface-level. Can you provide deeper insights into why certain architectures perform better on specific tasks or anatomical structures?*
>
> - In `Sec. D.3`, we have provided architectural insights into both the top-ranking and bottom-ranking algorithms. But we find it difficult to extract trustworthy architectural insights directly from our current benchmark results.
>   - For example, `Tables 2-3` show that top performing models in our benchmark are usually CNNs within the nnU-Net framework. However, it's unclear if this is due to an intrinsic advantage of CNNs over Transformers or just an indication of nnU-Net’s superior pipeline configuration. Given that Transformers are newer, future frameworks, designed for them, could potentially enhance their performance. I.e., mature frameworks that extract the best from both CNNs and transformers should allow fairer architectural comparisons in the future. **Beyond medical imaging, the architectural debate between CNNs and Transformers in computer vision has been ongoing and remains unresolved [[Bai et al., NeurIPS 2021](https://proceedings.neurips.cc/paper/2021/file/e19347e1c3ca0c0b97de5fb3b690855a-Paper.pdf); [Wang et al., 2022](https://arxiv.org/pdf/2206.03452)].**
>
> - Our benchmark provides **“predictions-only”** results, which can be heavily influenced by many factors such as preprocessing, data augmentation, post-processing, and training hyper-parameters [[Isensee et al., Nat. Methods 2021](https://www.nature.com/articles/s41592-020-01008-z)]. To draw convincing architectural insights, extensive ablation studies under controlled settings are required. However, conducting ablation studies for all **16** AI algorithms would be extremely costly for us.
>
> - We anticipate further insights and details from the AI inventors' upcoming technical reports, including extensive ablation studies. We are also happy to assist the inventors in their ablation studies by providing feedback on the OOD evaluation results of their algorithm variants.

---

> ### Author Rebuttal · Authors · 2024-08-18
>
> # Responses to Reviewer 68Sv (Part III)
>
> > **Q6.** *Clinical relevance. The benchmark focuses solely on anatomical segmentation. How do you plan to incorporate more clinically relevant tasks, such as pathology detection or severity grading, in future iterations?*
>
> - Thank you for your suggestion. Anatomical segmentation is **directly relevant** to many clinical tasks, such as:
>    - Improving robotic surgery guidance [[Park et al., 2024](https://www.ncbi.nlm.nih.gov/pmc/articles/PMC10998267/)]
>   - Measuring liver volume in cirrhosis patients [[Roth et al., 2015](https://link.springer.com/chapter/10.1007/978-3-319-24553-9_68)]
>   - Measuring pancreatic volume in diabetes patients [[Roth et al., 2015](https://link.springer.com/chapter/10.1007/978-3-319-24553-9_68)]
>   - Measuring kidney, liver and spleen volume in patients with autosomal dominant polycystic kidney disease (ADPKD). This is the **most common** inherited kidney disease, which causes cysts to develop in the kidneys and liver, causing their enlargement, often accompanied by an enlarged spleen. Organ volume measurement is used to monitor the progression of ADPKD and can help predict the onset of different stages of the disease. [[He et al., 2024](https://www.sciencedirect.com/science/article/abs/pii/S1076633223004762)]
>
> - Anatomical segmentation can also **indirectly benefit** many other clinical tasks. Intuitively, precisely localizing the anatomical structures is the foundation of the tasks such as pathology detection and severity grading.
>   - Pre-training on large organ segmentation datasets improves AI performance for tumor segmentation fine-tuning [[Li et al., 2024](https://www.cs.jhu.edu/~alanlab/Pubs23/li2023suprem.pdf)].
>   - Anatomical segmentation is useful for severity grading, since understanding the interaction between tumors and adjacent anatomical structures is [key for cancer staging and treatment planning](https://www.cancer.gov/types/pancreatic/patient/pancreatic-treatment-pdq).
>
> - We are planning to incorporate cancer imaging tasks in the next editions of the Touchstone benchmark, such as **tumor detection/diagnosis** in CT scans. Based on our initial estimation, our dataset (i.e., AbdomenAtlas and JHH) consist of 60% of normal (healthy) CT volumes and 40% of abnormal (with tumors) CT volumes. These can potentially be used to benchmark AI algorithms in tumor-related tasks. However, we must emphasize that annotating tumors is much more challenging than annotating organs, since tumors have blurry boundaries, subtle intensity, and potentially small size.
>
> ---
>
> > **Q7.** *Error analysis. The paper would benefit from a more thorough error analysis. Can you provide insights into common failure modes across algorithms? Are there specific types of images or anatomical variations that consistently challenge all models?*
>
> - We provided visual examples and demographic analysis for **four** specific types of common failure modes in `Figures 2, 5-7` . Due to the page limit, `Figures 5-7` are in the supplementary. These four types are summarized as follows:
>
>   1. **Pathological Conditions.** Algorithms tend to perform worse on patients with pathological conditions compared to normal patients. For example, in the JHH dataset, patients diagnosed with cancer have significantly lower DSC scores than those without, as shown in `Figure 2`.
>
>   2. **Incorrect Annotations.** Low DSC scores in algorithms can result from incorrect annotations. For example, in the first two rows of `Figure 7`, annotators in the DAP Atlas dataset mistakenly labeled a kidney cyst as the gallbladder. Consequently, AI algorithms appeared to fail on these CT volumes. However, after radiologists visually inspected the AI predictions, it was confirmed that the annotations were incorrect, and the AI predictions were accurate.
>
>   3. **Tubular Structures.** The boundaries of small tubular structures continue challenging all algorithms. For example, the boundaries of aorta and IVC are difficult for all algorithms to segment precisely, as shown in the last two rows of `Figure 5`.
>
>   4. **Low-Quality CT Volumes.** Algorithms also struggle with low-quality CT volumes. For example, as shown in the first three rows and the last row of `Figure 6`, algorithms fail to locate the aorta when CT volumes have low resolution and severe artifacts.
>
> - Thanks for your suggestions, we have now included these insights of common failure modes in the supplementary.

---

> ### Author Rebuttal · Authors · 2024-08-18
>
> # Responses to Reviewer 68Sv (Part IV)
>
> > **Q8.** *Longitudinal evaluation. You mention a commitment to long-term benchmarking. Can you elaborate on your plans for maintaining and updating the benchmark over time? How will you ensure its continued relevance as the field evolves?*
>
> - Thank you for your question. We discussed our plan for long-term benchmarking in `Sec. 1` Contribution #5. Following your suggestion, we have now elaborated this in our supplementary with more details. Throughout the next **five years**, we plan to continuously enhance our benchmark, leveraging our **strong** and **long-term** collaborations with leading clinical, technology, and research institutions globally, such as [Johns Hopkins Hospital](https://www.hopkinsmedicine.org/the-johns-hopkins-hospital), [City of Hope](https://www.cityofhope.org/), [University Hospital Essen](https://www.uk-essen.de/), [NVIDIA](https://www.nvidia.com/en-us/), [Microsoft](https://www.microsoft.com/en-us/), [DKFZ](https://www.dkfz.de/en/index.html), and [UCSF](https://www.ucsf.edu/). Our efforts focus on four key initiatives:
>
>   1. **Scaling Medical Images**: Through our collaborations, we are acquiring new CT volumes to  increase the size of our dataset for both training and testing AI algorithms (see `Sec. 1` Contribution #5). We have already collected **29,218 CT volumes** (11,385,719 slices; sized 4TB). In addition, we plan to expand Touchstone to the MRI modality.
>
>   2. **Annotating More Classes**: We have built a team of additional **15** expert radiologists and **60** medical trainees, who can annotate more anatomical structures in our datasets, in particular, abnormalities like **tumors** (see the last second paragraph in `Sec. 4`).
>
>   3. **Automatic OOD Evaluation**: Our partner institutions will implement an automatic online platform to perform third-party OOD evaluation and accommodate numerous algorithm submissions (see the last second paragraph in `Sec. 4`).
>
>   4. **Advancing Technical Frontiers**: We plan to expand beyond standard segmentation, and leverage massive datasets to empower and benchmark new AI technologies, such as self-supervised learning, SAM 2.0 and vision-language algorithms.
>
> ---
>
> > **Q9.** *I presume that all of the data is fully de-identified such that it is HIPAA-compliant or follows NEMA's recommendations, since this is being released as a publicly available dataset, but would like the authors to confirm this.*
>
> - Yes, you are correct. We confirm that all of the data is fully de-identified.

---

> ### Author Response · Authors · 2024-08-28
> **To Reviewer 68Sv: Anything Else We Can Address for You?**
>
> Dear Reviewer 68Sv,
>
> Thank you once again for your time and thoughtful comments, which were important to enhance our manuscript.
>
> As the discussion period approaches its end, we wanted to check if you have any questions or concerns regarding our manuscript or our answer to your comments?
>
> We would be more than happy to provide any further clarifications, discussions, or additional experiments to ensure all your concerns are fully addressed.
>
> With best regards,\
> Authors

---

### Official Review · Reviewer_5KZ1 · 2024-07-22
**Meta-dataset of abdominal segmentation**

**Rating:** 5
**Confidence:** 5

**Review:**

The paper is well-written and easy to follow. However, some more clarity about the annotation protocol of the JHH subset would be greatly appreciated.

The strongest point of the work is the inclusion of the JHH dataset, which is the largest dataset used for testing on this topic. The high value of this contribution, however, is mitigated by the fact that it remains private. Hence, some of the features of a "ideal benchmark" (as per the authors claim) cannot be really exploited by other researchers in the community.

There is a trend in proposing datasets that are a collection of previously published datasets. This is the case of this work, where on of the subsets, Abdominal, is already a previous work from the research group, collecting previously published abdominal CT images. Without a doubt, there is some value in this type of work, but a question arises: until when should re-collection of data is worth of publishing?

Please see other sections for details on strengths, limitations, correctness, clarity, etc.

**Strengths:**

Significance
- Very large testing set. As claimed in the paper, the largest testing set available.

Relevance
- Relevant dataset for those working on abdominal segmentation from CT images

Quality of the research
- An interesting aspect is that authors from the methods from the benchmark were involved in the work. Nonetheless, this is not original, per se. Medical imaging challenges (https://grand-challenge.org/) are a long-term well-established practice.

**Additional Feedback:**

For a paper on medical image segmentation, it is disappointing not to see a single visual example of the images/annotations/results.

**Clarity:**

Some aspects to improve clarity:
- The abstract and the title should reflect what the paper is about. It takes a while to figure out that this work is about abdominal segmentation from CT images, not medical image segmentation, in general, as both abstract and title tend to suggest.
- More details could be provided on the annotation protocol for the new images. It is just briefly mentioned that a team of radiologists was involved.

**Correctness:**

- The analysis on sex, gender, race and metadata is done on a subset of the complete dataset. While it is true that JHH is the largest subset of data, it contradicts the claim that the strength of the conclusions is supported by the diversity of the data distribution. Of all the datasets, JHH is the least diverse as it comes from a single center.

**Documentation:**

The JHH dataset is a new dataset for which more details regarding dataset collection would be greatly appreciated. Precisely:
- The team of radiologists is the one of 15 radiologists?
- How were the images assigned to this team?
- Precisely, how many years did it take to complete the task? Did the team change over this time?
- Are the 3 experts also annotating? It seems they do, but they cannot validate those they annotate. However, this is a guess. It is not explicit from the text.
- What organs were annotated? Are these the same as for the other datasets?
- Was there a specific protocol followed for the annotations? e.g. what was the window-level setup? was it established or could each doctor adjust the image according to their preferences?
- How is the AI assistance configured?

**Limitations:**

The fact that JHH is private is mentioned as a potential limitation.

The metadata across datasets is different and has an impact on some of the claims in this work. That could be discussed.

**Opportunities For Improvement:**

Significance
- The title is an overstatement. This work focuses on abdominal segmentation from CT, not medical image segmentation in general. This should appear clearly in both the abstract and title. As it is, it is misleading.
- The presented data collects 3 previous datasets, already published, and includes a fourth one that it is private. This is a big limitation as the new added value is rather limited.

Relevance
- As per the previous point, the scope is limited to those working on abdominal segmentation. While this is not a problem per se, it is not as broad as general medical image segmentation

**Relation To Prior Work:**

- The prior work provides an overview of other CT datasets
- The paper omits to mention that the idea of involving algorithm developers as part of the benchmark is a well-established practice in the medical imaging community. This is done typically through challenges (https://grand-challenge.org/). This is not the first effort that commits to "Evaluating new algorithms with long-term commitment". There are plenty of works on that. The MICCAI society, for instance, has a special interest group surveying this type of efforts (https://miccai.org/index.php/special-interest-groups/challenges/)

**Summary And Contributions:**

This paper introduces a benchmark for abdominal segmentation from CT images. It updates a previously published dataset (i.e. reference [44] which already collected multiple abdominal CT datasets) with some relatively recent abdominal datasets (TotalSegmentator and DAP), as well as, a very large proprietary dataset (JHH). All except the latter are of public access.

The proposed benchmark focuses on the assessment of out-of-distribution detection. The 3 added datasets (TotalSegmentator, DAP and JHH) are used specifically for this. Within the benchmark, some of the most popular segmentation frameworks (e.g. nnUnet, MONAI) and other recent ones (Vision-Language, SAM, diffusion-based) are considered for evaluation. An interesting feature of the benchmark is that the authors of the frameworks were involved in the evaluation.

---

> ### Author Response · Authors · 2024-08-18
> **To Reviewer 5KZ1: Request for Clarifications**
>
> Dear Reviewer 5KZ1,
>
> We thank you for your time reviewing our manuscript. We are working on addressing all your questions, but we are a bit confused about one of your comments:
>
> > *"The metadata across datasets is different and has an impact on some of the claims in this work. That could be discussed."*
>
> 1. Could you please clarify which of our claims is affected by the metadata differences?
>
> 2. How does the metadata differences impact these claims?
>
> Your answer is very important for us to better address your concern. Thank you very much.
>
> Sincerely,
>
> Authors of Paper 217

---

> > ### Comment · Reviewer_5KZ1 · 2024-08-19
> >
> > Dear authors,
> > Sure! Let me rephrase as indeed I was not very clear.
> >
> > I refer to the results you present in Figure 2, which lead to several claims around gender, scanner effects, etc. The available metadata (age range, race, gender, diagnosis, etc) is not consistent across datasets. This may be a limitation as it is not possible to observe a pattern across datasets and drive conclusions, such as those discussed in section 3.2.
> >
> > Moreover, in the case of age, which is common to all, the behavior is not consistent. The drop in performance is mild at an advanced age for TotalSegmentator, whereas in the other two the drop is more pronounced.
> >
> > Could you please comment on this?

---

> > > ### Author Response · Authors · 2024-08-20
> > > **The Need for Extensive Metadata Analyses**
> > >
> > > Dear Reviewer 5KZ1,
> > >
> > > Thanks for your clarification.
> > >
> > > > *The available metadata...is not consistent across datasets. This may be a limitation as it is not possible to observe a pattern across datasets and drive conclusions.*
> > >
> > > - Yes, it is a pity that metadata are not easily accessible in most public datasets. For TotalSegmentator and DAP Atlas, we have tried to request more metadata, but they were simply missed in these datasets (e.g., race).
> > >
> > > - The lack of metadata in public datasets has impacted our analysis as pointed out by you. We agree that a reliable pattern must be observed across datasets. So we have now toned down some of our claims in Sec. 3.2, especially those lacking sufficient evidence from all three datasets (e.g., race). We add this declare in the front of Sec. 3.2.
> > >
> > >   - *We studied correlation between AI performance and the five types of metadata in Fig. 2. Metadata of age, sex, and diagnosis are analyzed on all three datasets while race and manufacturer are only analyzed on one dataset due to the factors out of our control.*
> > >
> > > ---
> > >
> > > > *In the case of age, which is common to all, the behavior is not consistent.*
> > >
> > > - **Multi-confounder effect** [[Skelly et al., 2012](https://www.thieme-connect.de/products/ejournals/abstract/10.1055/s-0031-1298595)]: For example, patients of age >80 in the TotalSegmentator, DAP Atlas, and JHH datasets may differ considerably in their diagnoses and other confounders. These underlying differences can lead to inconsistent observations across datasets. To address this issue, analyses controlled for multiple combinations of confounders are needed. Therefore, we made the following statement and action.
> > >     - *”We have analyzed AI performance by metadata such as sex, age, and race but realized that a more rigorous analysis could be based on combined criteria (e.g., white females aged 30–40)”* (`Sec. 4`).
> > >     - We **released** per-sample DSC and NSD scores for all AI algorithms. The data and code on our [GitHub page](https://github.com/MrGiovanni/Touchstone) allows anyone to easily perform **combined** criteria analyzes with statistical tests.
> > >
> > > ---
> > >
> > > > *The drop in performance is mild at an advanced age for TotalSegmentator, whereas in the other two the drop is more pronounced.*
> > >
> > > - TotalSegmentator results show much larger variance than other two datasets (see `Fig. 2`). This makes the drop in performance seem mild but actually, it is not. Specifically, from age 50 to 89, TotalSegmentator has a median DSC dropped 0.8$\rightarrow$0.78 (**-0.02**); DAP Atlas dropped 0.9$\rightarrow$0.86 (**-0.04**); JHH dropped 0.86$\rightarrow$0.84 (**-0.02**).
> > >
> > > ---
> > >
> > > Our study underscores the need for detailed metadata for algorithmic benchmark, which is currently a big limitation in the medical domain. Evidenced by our `Table 1`, only KiTS & FLARE provided metadata analysis on sex, age, and/or race. Our Touchstone not only provides more extensive metadata analyses, including diagnosis, but also offers an order of magnitude more test data ($N$=6,933) for benchmarking.
> > >
> > > For next editions of Touchstone, we are creating more annotated, OOD test sets with detailed metadata, thanks to our current collaborations with other medical research institutions, such as [City of Hope](https://www.cityofhope.org/), [University Hospital Essen](https://www.uk-essen.de/), [DKFZ](https://www.dkfz.de/en/index.html), and [UCSF](https://www.ucsf.edu/).
> > >
> > > We thank you again for your insightful comment, which led to fruitful discussions that improved our manuscript and raised awareness for the necessity of more well-documented test sets.
> > >
> > > Sincerely,
> > >
> > > Authors of Paper 217

---

> > > > ### Comment · Reviewer_5KZ1 · 2024-08-21
> > > >
> > > > Many thanks for the detailed answer and for modifying the text accordingly.

---

> > > > > ### Author Response · Authors · 2024-08-28
> > > > > **To Reviewer 5KZ1: Any Areas Where We Could Provide Further Clarification?**
> > > > >
> > > > > Dear Reviewer 5KZ1,
> > > > >
> > > > > Thank you again for your time and insightful comments. Addressing them has significantly improved the quality of our paper. We were particularly pleased to include an in-depth analysis of the multi-confounder effect, as summarized in our last comment to you.
> > > > >
> > > > > As the discussion period draws to a close, may we ask you if there are any aspects of our work or our rebuttal that remain unclear or concerning to you?
> > > > >
> > > > > We would be more than happy to offer any further clarifications, engage in additional discussions, or conduct further experiments to ensure all your concerns are fully addressed.
> > > > >
> > > > > Best regards,\
> > > > > Authors of Paper 217

---

> ### Author Rebuttal · Authors · 2024-08-18
>
> # Responses to Reviewer 5KZ1 (Part I)
>
> We would like to thank you for your diligent efforts and constructive comments on our paper, which have helped us think more deeply and craft a significantly better benchmark paper. We are grateful for recognizing the strengths of our paper, *“well-written and easy to follow”*, *“ largest dataset used for testing on this topic”*, and *“an interesting aspect is that authors from the methods … were involved in the work”*. Below please find the responses to some specific comments.
>
> ---
>
> > **Q1.** *Some more clarity about the annotation protocol of the JHH subset would be greatly appreciated.*
>
> - Thanks for your comment. Our previous papers [[Park et al., 2020](https://www.sciencedirect.com/science/article/pii/S2211568419301391); [Xia et al., 2022](https://www.medrxiv.org/content/10.1101/2022.09.24.22280071v1)] (cited in our `Sec. 2.1`) have documented the annotation protocol of the JHH subset in depth. For completeness, we have now also explained the annotation protocol in our Supplementary, and we will summarize it in the answer to your **Q9**.
>
> ---
>
> > **Q2.** *The strongest point of the work is the inclusion of the JHH dataset, which is the largest dataset used for testing on this topic. The high value of this contribution, however, is mitigated by the fact that it remains private. Hence, some of the features of an "ideal benchmark" (as per the authors claim) cannot be really exploited by other researchers in the community.*
>
> - Having JHH ($N$=5,172) [available for third-party evaluation](https://huggingface.co/datasets/AbdomenAtlas/AbdomenAtlas1.0Mini) is a big plus for OOD benchmarks. We are making every effort to have this dataset publicly accessible, or be available by request, but this can cause many **problems**—including completely destroying the OOD benchmark—that we should be aware of.
>
>   - For example, Medical Segmentation Decathlon (MSD) [[Antonelli et al., Nat. Commun. 2022](https://www.nature.com/articles/s41467-022-30695-9)] was a benchmark with publicly accessible test data and its test labels were private. Similarly, BTCV [[Landman et al., 2015](https://www.synapse.org/Synapse:syn3193805/wiki/89480)] released both testing data and labels. However, due to the growing need for more data/labels in the medical domain, even MSD/BTCV test sets have been annotated and integrated into recent public datasets, like FLARE2023 [[Ma et al., 2023](https://flare.grand-challenge.org/)] and AbdomenAtlas [[Qu et al., NeurIPS 2023](https://www.cs.jhu.edu/~alanlab/Pubs23/qu2023abdomenatlas.pdf)]. Therefore, any AI models trained or pre-trained on these public datasets are problematic in the MSD/BTCV leaderboard. With widespread access to test data, it becomes challenging to fairly compare models, as some may be overly optimized for the benchmark rather than for real-world performance.
>
> - As a result, researchers must continue to seek or develop new datasets—preferably with data and labels that have never been disclosed. This is critical in many fields as well. Yann Lecun—*“[beware of testing on the training set](https://twitter.com/ylecun/status/1723752958037315874)”*—in response to the incredible results achieved by GPT. **Our JHH dataset ($N$=5,172) is a valuable resource that other researchers can exploit to reduce data leakage risks and improve the reliability of OOD benchmark results.**
>
> - Our Touchstone Benchmark is still in the initial stage, so we are very careful with the decision of releasing JHH data/labels. It must be managed carefully to ensure its benefits outweigh the risks.
>
> ---
>
> > **Q3.** *There is a trend in proposing datasets that are a collection of previously published datasets. This is the case of this work, where on of the subsets, Abdominal, is already a previous work from the research group, collecting previously published abdominal CT images. Without a doubt, there is some value in this type of work, but a question arises: until when should re-collection of data is worth of publishing?*
>
> - To be clear, our paper is dedicated to the benchmark contribution for the NeurIPS D&B Track. **We did not propose new datasets:** The AbdomenAtlas, TotalSegmentator, DAP Atlas, and JHH datasets have already been published, and we cited them properly and clarified their data licenses in `Sec. 2.1`. After multiple checks in our paper, we assure that no single claim was made about new dataset collection/annotation.
>
> ---
>
> > **Q4.** *The title is an overstatement. This work focuses on abdominal segmentation from CT, not medical image segmentation in general. This should appear clearly in both the abstract and title. As it is, it is misleading.*
>
> - Thanks for your valuable feedback regarding our title and abstract. We agree that the previous title can be confusing. Now, the revised title is **“Touchstone Benchmark: Are We on the Right Way for Evaluating AI Algorithms for Abdominal CT Segmentation?”** and in the abstract, we have now made it clearer that the current benchmark focuses on semantic segmentation in abdominal CT scans. We are actively exploring further refinements for greater clarity.
>
> ---
>
> > **Q5.** *The presented data collects 3 previous datasets, already published, and includes a fourth one that is private. This is a big limitation as the new added value is rather limited.*
>
> - We stress that this is **not** a paper about dataset contribution. Instead, we highlighted five key contributions related to the creation of a new large-scale benchmark in `Sec. 1`, and the benchmark contributions were quantified and compared in `Table 1`.
>   - According to its official guidelines, NeurIPS D&B Track calls for “[benchmarks on new or **existing** datasets, as well as benchmarking tools](https://neurips.cc/Conferences/2024/CallForDatasetsBenchmarks)”.
>
> - Previously exclusive to select groups at JHU, this work made the JHH dataset now **valuable** to the broader research community for third-party OOD evaluation (`Sec. 4`).

---

> ### Author Rebuttal · Authors · 2024-08-18
>
> # Responses to Reviewer 5KZ1 (Part II)
>
> > **Q6.** *As per the previous point, the scope is limited to those working on abdominal segmentation. While this is not a problem per se, it is not as broad as general medical image segmentation*
>
> - We have revised our title and abstract (please refer to **Q4**) clarifying that our benchmark considers abdominal segmentation. We agree that this information should be clear even before the introduction.
>
> ---
>
> > **Q7.** *The metadata across datasets is different and has an impact on some of the claims in this work. That could be discussed.*
>
> - Could you please clarify which of our claims is affected by the metadata differences? How does the metadata differences impact these claims? (see ***our official comment***).
>
> ---
>
> > **Q8.** *The analysis on sex, gender, race and metadata is done on a subset of the complete dataset. While it is true that JHH is the largest subset of data, it contradicts the claim that the strength of the conclusions is supported by the diversity of the data distribution. Of all the datasets, JHH is the least diverse as it comes from a single center.*
>
> - Thank you for your comment. The diversity of data distribution includes more than just the number of centers; it also includes age, sex, manufacturer, diagnosis, and many other factors (see `Sec. 3.2` and `Figures 25-72`). Importantly, JHH is the **only** dataset that provides race information, allowing us to compare the results (`Fig. 2`); the race information is unknown in TotalSegmentator and DAP Atlas. Therefore, the inclusion of JHH is value-added because it **enabled** the analysis on race.
>
>   - *“We did not find significant performance differences across diverse races in JHH, our only test set with race metadata”* (our new revision).
>
> - We have conducted the analysis on sex, gender, and cancer diagnosis on **three datasets**, i.e., JHH, TotalSegmentator, and DAP Atlas. Through this multi-center analysis, we found common patterns across datasets which based our conclusions.
>
>   - *“AI performance reduces for advanced age. Median DSC starts dropping around the fifties, and differences to younger patients become statistically significant ($p<0.05$) after 70, in both JHH and DAP Atlas. The descending trend after 50 is visible also in TotalSegmentator, but, with the higher performance variability in the dataset, differences are not significant. However, the creators of TotalSegmentator observed that aging caused attenuation in CT scans [56]...”* (copied from  `Sec. 3.2`).
>
> - With the success of the first edition of Touchstone Benchmark, we are actively pursuing multi-center, OOD datasets, as recommended by the reviewer, to further enhance the benchmark. This is difficult for many well-known reasons—patient privacy, ethical compliance, data annotation, intellectual property, etc. *Rome wasn't built in a day.* **A multi-center, OOD dataset can never be made without accumulating the contribution of every single-center dataset.** We hope this benchmark initiative at JHH, [a highly regarded institution](https://hub.jhu.edu/2024/07/16/johns-hopkins-hospital-us-news-best-hospitals-2024-25/), could also inspire more institutes to contribute their private datasets for third-party OOD evaluation.

---

> ### Author Rebuttal · Authors · 2024-08-18
>
> # Responses to Reviewer 5KZ1 (Part III)
>
> > **Q9.** *Some more clarity about the annotation protocol of the JHH subset would be greatly appreciated.* and *More details could be provided on the annotation protocol for the new images. It is just briefly mentioned that a team of radiologists was involved.*
>
> - Thank you for your suggestions, we have explained the annotation protocol below. However, to be clear, we reiterate that collecting and annotating the JHH dataset is **not** a contribution of our paper. We have cited the papers (in `Sec. 2.1`) that described the creation of JHH in depth, including the annotation protocol [[Park et al., 2020](https://www.sciencedirect.com/science/article/pii/S2211568419301391), [Xia et al., 2022](https://www.medrxiv.org/content/10.1101/2022.09.24.22280071v2.full.pdf)]. We copied the protocol here for your reference:
>
>   -  *“Abdominal organs were manually segmented in both arterial and venous phase CT images ... The entire arterial phase for all organs was generally annotated first and then venous phase images of the same donor were annotated sequentially. Early in the study, we tested two different ways of segmentation; phase-alternate process … and sequential process (annotating all organs in one phase first and moved to the other phase). We decided to use the sequential process ... Each one phase took approximately 4 hours to manually segment by a reader. Each reader annotated the 22 structures … Each reader used the best software options such as display plane and window level, 2D or three-dimensional (3D) segmentation and sensitivity setting to annotate each structure based on its shape, attenuation and contrast to adjacent organs in each dataset. The contours were then verified by one of three board-certified radiologists with between five and thirty years of body CT imaging experience. The reader and the radiologist had face-to-face sessions to go over each case. If there were any disagreements or errors found by the radiologist, they were corrected at the time of the sessions.”*
>
> - More details, such as protocols for boundary determination of contiguous structures, are available in [[Park et al., 2020](https://www.sciencedirect.com/science/article/pii/S2211568419301391)]. Following your suggestion, we have now summarized the annotation protocol and its details in the revised **supplementary**.
>
> ---
>
> > **Q10.** *The paper omits to mention that the idea of involving algorithm developers as part of the benchmark is a well-established practice in the medical imaging community. This is done typically through challenges (https://grand-challenge.org/). This is not the first effort that commits to "Evaluating new algorithms with long-term commitment". There are plenty of works on that. The MICCAI society, for instance, has a special interest group surveying this type of efforts (https://miccai.org/index.php/special-interest-groups/challenges/)*
>
> - This is miscommunication—we did **not** claim or mention anywhere in the paper that we are the first to have algorithm inventors involved or the first to make long-term benchmark commitment. Instead, our paper recognized that these features are critical to be done for any benchmark (`Sec. 1`),  and our `Table 1` **endorsed almost all preexisting challenges** on abdominal CT segmentation and justified how we advanced these features compared with them.
>
> - In related benchmarks & our innovation (`Sec. 1`) we mention that, in challenges, algorithms are trained by the the people most interested in its success — i.e., the algorithm’s inventors:
>   - *”fairer benchmarks, where each AI algorithm is configured by the people most interested in its success, are … mostly found in challenges.”*
>
> - To make this clearer, following your suggestion, we have now revised the first part of related benchmarks & our innovation in Sec. 1 to:
>   - *“In a general sense, we define a benchmark as an algorithmic comparison. Accordingly, the most common type of benchmark are the standard comparisons found in thousands of papers where authors present new methods and train baseline algorithms for comparison. As previously explained, this type of benchmark incurs the risk of unfairness, due to possible asymmetric efforts made in optimizing the proposed and alternative algorithms. However, **challenges** are a different type of benchmark, where developers train their own algorithms and submit them for third-party evaluation, mitigating the risk of unfair comparisons. For this reason, Table 1 contrasts our Touchstone benchmark to a non-exhaustive list of the most influential abdominal CT segmentation challenges.”*
>
> - Moreover, to avoid any potential overclaim, we have now acknowledged previous **long-term** CT segmentation challenges in related benchmarks & our innovation (Sec. 1):
>   - *“This work is the starting point of a long-term benchmark, which we commit to maintain and improve over the years. Considering the importance of long-term commitment, we must acclaim **KiTS**, an abdominal segmentation challenge that had 3 editions since 2019 [[Heller et al., 2019](https://kits19.grand-challenge.org/data/);[Heller et al., 2021](https://kits-challenge.org/kits21/);[Heller et al., 2023](https://kits-challenge.org/kits23/)]; and **FLARE**, a challenge being consistently held yearly since 2021 [[Ma et al., 2021](https://flare.grand-challenge.org/);[Ma et al., 2022](https://flare22.grand-challenge.org/);[Ma et al., 2023](https://codalab.lisn.upsaclay.fr/competitions/12239)].”*
>
> ---
>
> > **Q11.** *For a paper on medical image segmentation, it is disappointing not to see a single visual example of the images/annotations/results.*
>
> - Thank you for your comment, but this is not the case. We provided **extensive** visual examples in our supplementary.  Specifically, `Figures 3-4` in `Sec. A.2` and `Figures 5-7` in `Sec. D.1` show visual examples of the CTs/annotations/results.

---

> ### Author Rebuttal · Authors · 2024-08-18
>
> # Responses to Reviewer 5KZ1 (Part IV)
>
>
> > **Q12.** *The JHH dataset is a new dataset for which more details regarding dataset collection would be greatly appreciated. Precisely:*
>
> - Certainly, we are happy to provide more details regarding the collection and annotation of the JHH dataset. We have addressed your questions point-by-point below and included this information in an additional section in our revised **supplementary**.
>
> >**Q12-1.** *The team of radiologists is the one of 15 radiologists?*
> - The annotation team contains 15 radiologists, as reported in [[Park et al., 2020](https://www.sciencedirect.com/science/article/pii/S2211568419301391); [Xia et al., 2022](https://www.medrxiv.org/content/10.1101/2022.09.24.22280071v2.full.pdf)].
>
> >**Q12-2.** *How were the images assigned to this team?*
> - Images were **randomly** assigned to each annotator.
>
> >**Q12-3.** *Precisely, how many years did it take to complete the task? Did the team change over this time?*
> - The annotation of JHH took **5 years** to complete. The team did not change over five years.
>
> >**Q12-4.** *Are the 3 experts also annotating? It seems they do, but they cannot validate those they annotate. However, this is a guess. It is not explicit from the text.*
> - No, the three **expert** radiologists were **not** annotating JHH. They just validated each annotated CT scan.
>   - *”The contours were then **verified** by one of three board-certified radiologists with between five and thirty years of body CT imaging experience”* [[Park et al., 2020](https://www.sciencedirect.com/science/article/pii/S2211568419301391)].
>   - *”... verified by one of three additional experienced radiologists, **none of whom** performed the annotations.”* [[Xia et al., 2022](https://www.medrxiv.org/content/10.1101/2022.09.24.22280071v2.full.pdf)].
>
> >**Q12-5.** *What organs were annotated? Are these the same as for the other datasets?*
> - There are **22** anatomical structures annotated in JHH. They are:
>   -  *“The organs annotated included the liver, pancreas, gallbladder, kidneys, stomach, duodenum, spleen, adrenal glands, and bones… Blood vessels were categorized as aorta, celiac artery, and superior mesenteric artery, inferior vena cava, portal vein, superior mesenteric vein, splenic vein and renal veins. The pancreatic duct and common bile duct were annotated when visible.”* [[Park et al., 2020](https://www.sciencedirect.com/science/article/pii/S2211568419301391)].
>
> - Our current benchmark considers nine anatomical structures. These structures are **same** in JHH and other datasets, such as our training set (AbdomenAtlas) and other two test sets (TotalSegmentator and DAP Atlas).
>
> >**Q12-6.** *Was there a specific protocol followed for the annotations? e.g. what was the window-level setup? was it established or could each doctor adjust the image according to their preferences?*
> - Please refer to **Q9** for the protocol followed for the annotations. More details can be found at [[Park et al., 2020](https://www.sciencedirect.com/science/article/pii/S2211568419301391)] .
> - Annotators were free to **adjust** the window-level setup and the image according to their preferences.
>   - *“Each reader used the **best software options** such as display plane and window level, 2D or three-dimensional (3D) segmentation and sensitivity setting to annotate each structure based on its shape, attenuation and contrast to adjacent organs in each dataset.”*
>
> >**Q12-7.** *How is the AI assistance configured?*
> - JHH was **manually** annotated with **no AI** assistance, needing 5 years for complete annotation.
>   - *“It required an average of **3 hours** to manually annotate the images of **one** healthy individual”* [[Xia et al., 2022](https://www.medrxiv.org/content/10.1101/2022.09.24.22280071v2.full.pdf)].

---

### Official Review · Reviewer_GiJy · 2024-08-03
**Comprehensive Evaluation and extremely useful Dataset Contribution**

**Rating:** 7
**Confidence:** 4
**Correctness:** Yes

**Review:**

The paper does a rigorous job of evaluating major segmentation algorithms including creation of new large evaluation dataset and is easy to follow and well written. Some details could have been added, for example, how were the duplicate CT scans detected?
Since publicly available datasets have also been utilized for training and evaluation which themselves have been derived, clarity/explanation is required on how train/test leakage was avoided. For example, DAPAtlas dataset used for evaluation and AbdomenAtlas used for training both seem to contain images from AMOS Datasets.

**Strengths:**

Large multi-center varied Dataset construction is a massive undertaking and the efforts put in creating the training and evaluation dataset at such scale is commendable.  Organizing the challenge and bringing in targeted invitations to creators of recent SOTA segmentation models to perform a comprehensive benchmarking and evaluation resulting in generalizable insights is a good contribution to the community.

**Additional Feedback:**

What do you mean by “inference speed must be faster than 1e6 mm3 per second”, how do you enforce this?
Wouldn't it have been easier to enforce the compute requirements during inference via "segmentation for each volume should take no more than x seconds/minutes"?

Supplementary Material: Citations seem off. CT-ORG [46] -> should have been CT-ORG [45]; AbdomenCT-1K [42] -> [41]

**Clarity:**

Additionally, It is unclear if the segmentation for DAPAtlas is entirely generated by AI or were manually corrected as well.

**Documentation:**

Yes

**Limitations:**

Limitations in Dataset construction issues such as label noise is clearly stated. If the segmentation for DAPAtlas was entirely generated by AI, it might be pertinent to add this information in the limitations section with the added implication that the dataset contains only silver standard groundtruth and that performance of models on this dataset may be less insightful ( and that most models seem to do better on this dataset since the groundtruth may not contain detailed segmentation).

**Opportunities For Improvement:**

DAP Atlas dataset (part of the evaluation dataset in the proposed dataset) itself is constructed from aggregation of other publicly available datasets, and specifically, AMOS which is also used to construct the Training set in the proposed dataset. What was done to avoid (train/test) overlap?

Could you please describe how duplicate CT scans were detected?

**Relation To Prior Work:**

Yes

**Summary And Contributions:**

The paper constructs a (partially derived) large CT Segmentation Dataset for 9 organs. A major emphasis of the proposed dataset and benchmark is on comprehensive evaluation on large multicenter dataset on varied population. This includes construction of JHH dataset (5172 CT and Segmentation pair) exclusively for evaluation purposes and specifically for the proposed benchmark.

Additionally, although public datasets were used, additional labelling was performed for completeness, since the original datasets might not have common labels as they were independent efforts.

Disaggregated evaluation of the major segmentation algorithms is done across age, sex, gender, manufacturer, pathologies etc. on large (unprecedented) evaluation dataset resulting in generalizable insights.

---

> ### Author Rebuttal · Authors · 2024-08-17
>
> Many thanks for your encouraging review. We are especially grateful for your **`Accept`** and for recognizing *“comprehensive benchmarking ”*, *“generalizable insights”*, *“rigorous job of evaluating…”*, and *“easy to follow and well written”*.
>
> ---
>
> > **Q1.**  *DAP Atlas dataset (part of the evaluation dataset in the proposed dataset) itself is constructed from aggregation of other publicly available datasets, and specifically, AMOS which is also used to construct the Training set in the proposed dataset. What was done to avoid (train/test) overlap?*
>
> - There is no overlap between DAP Atlas and AMOS. These two datasets are exclusively collected from two different countries, i.e., **DAP Atlas is from Germany and AMOS is from China**.
>
>   - The DAP Atlas dataset [[Jaus et al., 2023](https://arxiv.org/pdf/2307.13375)] is a part of the AutoPET dataset [[Gatidis et al., 2022](https://autopet.grand-challenge.org/Dataset/)], which is collected from University Hospital Tübingen (Tübingen, Germany) and University Hospital of the Ludwig Maximilian University (Munich, Germany).
>
>   - The AMOS dataset is collected from China [[Ji et al., 2022](https://arxiv.org/pdf/2206.08023)].
>
> ---
>
> > **Q2.** *Could you please describe how duplicate CT scans were detected?*
>
> - Duplicate scans were identified by generating a **3D perceptual hash** [[Xu et al., 2015](https://www.atlantis-press.com/proceedings/meic-15/19738)] for each image in the dataset. By comparing the similarity of these hashes, duplicates were reliably detected, a finding that was further confirmed through manual inspection of CT scans with high perceptual hash similarities.
>
> ---
>
> > **Q3.** *If the segmentation for DAP Atlas was entirely generated by AI, it might be pertinent to add this information in the limitations section with the added implication that the dataset contains only silver standard groundtruth and that performance of models on this dataset may be less insightful (and that most models seem to do better on this dataset since the groundtruth may not contain detailed segmentation).*
>
> - Thank you, your comment is important and correct. We have now added this information in a Limitations section, as follows:
>
>   - *“The labels in the DAP Atlas dataset are entirely generated by AI without human annotation or revision [[Jaus et al., 2023](https://arxiv.org/pdf/2307.13375)], which may result in lower quality compared to TotalSegmentator (largely AI-generated with some human revisions) and JHH (fully human-annotated). Consequently, the results and conclusions derived from DAP Atlas may be less insightful than those from the other two test sets.”*
>
> ---
>
> > **Q4.** *Additionally, It is unclear if the segmentation for DAP Atlas is entirely generated by AI or were manually corrected as well.*
>
> - DAP Atlas labels were entirely generated by AI [[Jaus et al., 2023](https://arxiv.org/pdf/2307.13375)].
> ---
>
> > **Q5.** *What do you mean by “inference speed must be faster than 1e6 mm$^3$ per second”, how do you enforce this? Wouldn't it have been easier to enforce the compute requirements during inference via "segmentation for each volume should take no more than x seconds/minutes"?*
>
> - The CT volume size varies. The smallest and largest volumes in JHH are $512\times512\times62$ and $512\times512\times1899$, respectively. In TotalSegmentator, they are $149\times149\times29$ and $283\times162\times851$; in DAP Atlas, they are $512\times512\times240$ and $512\times512\times2651$. Larger volume naturally takes longer time for inference. So it cannot be simply “segmentation for each volume should take no more than x seconds/minutes.”
>
> - **How do you enforce this?** We multiply the number of voxels and the voxel spacing to get the total physical volume of a CT scan in mm$^3$. Then, we multiply this result by 1e6, to get the maximum time (seconds) an algorithm can take evaluating the CT scan.
>
> - To make it clearer, we have now added a new section in the supplementary to explain this metric. We appreciate your valuable feedback.
>
> ---
>
> > **Q6.** *Supplementary Material: Citations seem off. CT-ORG [46] -> should have been CT-ORG [45]; AbdomenCT-1K [42] -> [41]*
>
> - Thank you for pointing out this issue, we have corrected it.

---

### Official Review · Reviewer_mP5R · 2024-08-06
**Touchstone benchmark is a comprehensive and meaningful work for medical segmentation**

**Rating:** 7
**Confidence:** 3
**Clarity:** Yes, see the Pros above

**Review:**

**Pros:**
1. The motivation and contributions of this work are clearly described in the abstract, introduction, and conclusion sections, for example, I like that the authors used five points in the introduction sections to elaborate their motivations (glad to see that over-simplified metrics and unfair comparisons have been pointed out here)
2. The benchmark is a large-scale collaborative benchmark for medical segmentation, showing efforts to connect the medical and the machine learning areas (invitation to the inventors of AI algorithms) for inter-discipline research and enhance worldwide collaborations. Additionally, the authors said they are committed to maintaining and expanding this benchmark for several years
3. The paper is organized very well, proposing problems/motivations in the introduction section and evidencing with experimental results to support the claims as mentioned earlier
4. The 81-page supplementary material provides a very detailed introduction to the dataset and evaluated AI algorithm, and also includes as many analyses as possible for reproducibility and future work
5. The experimental design is careful and meaningful

**Cons:**

[Major concerns]
1. Introduction: "Second, small-size test sets. Annotating medical data is expensive and time-consuming, but training AI requires substantial annotated data [43, 44]. Therefore, most annotated data is used for training, leaving very little assigned for testing." I am curious if self-supervised learning can help in this problem since self-supervised learning is a way to avoid annotated data for training
2. Table 2: as the authors mentioned in section 2.1 "JHH - N=5,172", what is the "proprietary JHH subset (N=803)" here? why don't the authors include validation results on the whole set?
3. Conclusion and discussion: " We revised AbdomenAtlas 1.0 to create a cleaner version, AbdomenAtlas 1.0C, reducing label errors in the aorta to 5.4% and in the L&R kidneys to 0.6%", since the noise-free ground truth is not available, how do the authors calculate these numbers?

[Minor concerns]
1. Introduction: "The development of AI algorithms has led to enormous progress in medical segmentation if evaluated
using standard benchmarks" -> hard to understand, if evaluated?
2. Introduction: "We argue that standard benchmarks are misleading for comparing AI algorithms and delay progress" -> "Misleading" might be too harsh a word here, we need to agree that research is a long way and we can't avoid problems in between
3. Section 2.2: "We employed the same computer for all submitted algorithms" - is that only for evaluation?

**Strengths:**

See Pros

**Additional Feedback:**

N/A

**Correctness:**

Yes, the datasets, evaluation methods, and experiment design are appropriate and performed correctly, the details can be found in the supplementary material and open-source codes on GitHub

**Documentation:**

Yes, very detailed in supplementary material and GitHub codes

**Ethics:**

No, the authors mentioned that "All personally identifiable information was removed and the use of this dataset has received Institutional Review Board (IRB) approval from Johns Hopkins Medicine under IRB00403268."

**Limitations:**

Yes, in section 4 (main paper) and section H (supplementary material)

**Limitations:** noisy labels, usage of the private dataset (JHH), the demand for a web-based automatic evaluation

**Potential negative societal impact:** reinforcing biases, compromising data privacy, and leading to misuse of AI systems

**Opportunities For Improvement:**

See Cons

**Relation To Prior Work:**

Yes, prior work is discussed carefully in the introduction section and Table 1

**Summary And Contributions:**

**Summary:**
The paper proposed a large-scale collaborative benchmark for medical segmentation to solve the misalignment issue (in-distribution and small-size test sets, oversimplified metrics, unfair comparisons, etc.) in real-world scenarios.

**Contributions:**
1. The dataset is at an unprecedented scale: training volumes from 76 medical institutions around the world, and 6933 testing volumes from 8 additional hospitals
2. The benchmark is evaluating various AI algorithms in an out-of-distribution way, which is more aligned with real-world scenarios
3. The authors invited inventors of various AI algorithms to train their algorithms on the available training set, and evaluated those trained algorithms independently as a third party, to ensure fair comparison
4. The authors also provided extensive results in the main paper and supplementary material, including results based on age, sex, and race, which is meaningful for other research topics, such as model fairness

---

> ### Author Rebuttal · Authors · 2024-08-17
>
> We are very grateful for your **`Accept`** and accolades on our contributions, *“...a large-scale collaborative benchmark...”*, *“the dataset is at an unprecedented scale”*, *“...benchmark...aligned with real-world scenarios...”*, and *“...fair comparison...provided extensive results...”*. We have fixed all the issues that you kindly pointed out.
>
> ---
>
> > **Q1.** *Introduction: "Second, small-size test sets. Annotating medical data is expensive and time-consuming, but training AI requires substantial annotated data [43, 44]. Therefore, most annotated data is used for training, leaving very little assigned for testing." I am curious if self-supervised learning can help in this problem since self-supervised learning is a way to avoid annotated data for training.*
>
> - Self-supervised learning can certainly reduce annotation efforts needed to train AI algorithms. On the other hand, AI evaluation—the focus of our paper—still needs considerable manual-annotated, high-quality data, which unfortunately cannot be avoided by self-supervised learning.
>
> - For example, the construction and annotation of JHH ($N$=5,172) took a team of expert radiologists and medical trainees **five years** to complete [[Park et al., 2020](https://www.sciencedirect.com/science/article/pii/S2211568419301391)], where every CT scan was manually annotated by humans. Notably, the JHH dataset stand-alone, dedicated for testing in our paper, is even **much larger than** a combination of both training and test sets of most preexisting CT benchmarks. Detailed comparison as follows (extracted from `Table 1`).
>
> | benchmark | number of CT for training  | number of CT for testing |
> |:---------------|:---------------------:|:--------------------:|
> | MSD-CT |947 |465  |
> | FLARE’22 |2,050 |800 |
> | FLARE’23 |4,000 |400 |
> | KiTS21  |300 |100 |
> | AMOS22-CT |200 |200 |
> | LiTS  |130 |70 |
> | BTCV  |30 |20 |
> | CHAOS-CT |20 |20 |
> | Touchstone (**ours**) | 5,195 | 6,933 ($N_{JHH}$=5,172) |
>
> ---
>
> > **Q2.** *Table 2: as the authors mentioned in section 2.1 "JHH - $N$=5,172", what is the "proprietary JHH subset ($N$=803)" here? Why don't the authors include validation results on the whole set?*
>
> - JHH ($N$=803) is a randomly selected subset of JHH ($N$=5,172). We used this subset to give prompt feedback to authors after they submitted their checkpoints. In practice, a smaller subset is needed during the benchmark because inference on 5,172 CT volumes requires at least **168 GPU hours per model**; it can cause significant delay especially when reaching the peak period of participants submitting their models.
>
> - In our supplementary, `Tables 8-9` reported the performance on the **whole** JHH ($N$=5,172). We thank you for pointing this out, and we have now presented the $N$=5,172 results to the main paper and reported $N$=803 results to the supplementary.
>
> ---
>
> > **Q3.** *Conclusion and discussion: " We revised AbdomenAtlas 1.0 to create a cleaner version, AbdomenAtlas 1.0C, reducing label errors in the aorta to 5.4% and in the L&R kidneys to 0.6%", since the noise-free ground truth is not available, how do the authors calculate these numbers?*
>
> - These numbers are estimated based on medical priors. When a “ground truth” did not satisfy these priors, we considered it a label error, and had expert radiologists revise it. The medical priors are as follows:
>
>   - **Aorta:** (1) no overlap with the kidneys, (2) no overlap with the pancreas, and (3) no overlap with the spleen.
>
>   - **L&R Kidney:** (1) no overlap between the left and right kidneys and (2) the distance between the left kidney and the liver is greater than the distance between the right kidney and and the liver.
>
> - Thank you for your comment, we have introduced a new section in the supplementary to clarify these medical priors and label error estimation.
>
> ---
>
> > **Q4.** *Introduction: "The development of AI algorithms has led to enormous progress in medical segmentation if evaluated using standard benchmarks" -> hard to understand, if evaluated?*
>
> - AI algorithms can continue to improve the performance over years on a **small, IID test set** (i.e., standard benchmarks), but this does not guarantee the improvement if they are evaluated on **large, OOD test sets**, i.e., *“medical images from multiple hospitals, varied in different scanners, clinical protocols, patient demographics, or disease prevalences”* (`2nd sentence in Sec. 1`).
>
> - Thank you very much for your feedback. We have now revised the sentence to make it clearer.
>
> ---
>
> > **Q5.** *Introduction: "We argue that standard benchmarks are misleading for comparing AI algorithms and delay progress" -> "Misleading" might be too harsh a word here, we need to agree that research is a long way and we can't avoid problems in between.*
>
> - Thank you for your suggestion. We have now revised the sentence as *"We will discuss five common benchmark pitfalls, which may cause confusion in algorithm comparisons and delay progress."*
>
> ---
>
> > **Q6.** *Section 2.2: "We employed the same computer for all submitted algorithms" - is that only for evaluation?*
>
> - Yes, we employed the same computer for evaluating all submitted algorithms. In the training, the participants were allowed to use any computational resources as long as their inference speed meets our standard, i.e., *“the inference speed must be faster than 1e6 mm$^3$ per second”* (see `Sec. 2.2`).

---

### Author Rebuttal · Authors · 2024-08-14

We will address each and every question under each Reviewer’s session, but before that, we must make four declarations to ensure we and all Reviewers are aligned, and promote a productive and in-depth discussion in the next few days.

---

- **Declare #1**: We had an **80-page supplementary** that can already address many of the reviewers’ questions by providing critical information about data/label visualization, evaluation metrics, architectural insights, and error analysis. This supplementary is very important, so we ask reviewers to please read and consider it.

---

- **Declare #2**: We **did NOT claim** anywhere in the paper that we are the first to test on out-of-distribution data, introduce very large test sets, analyze results from multiple perspectives, invite algorithm inventors, and make long-term commitment. Instead, we compared all these five features in our benchmark against preexisting benchmarks (Table 1).

---

- **Declare #3**. All the five contributions claimed in our paper are about **benchmarking instead of dataset creation**. To be clear, collecting and annotating the JHH dataset is NOT the contribution of our paper. The JHH dataset has been published and well-documented in [Park et al., 2020](https://www.sciencedirect.com/science/article/pii/S2211568419301391) (cited in our Sec. 2.1), with several examples provided in its Figures 3-6.

---

- **Declare #4**: Out-of-distribution (OOD) test data (both images and annotations) must be **kept private**. Leaking test data to the public can lead to overfitting and compromise the integrity of OOD evaluation, immediately [[Geirhos et al., 2020](https://www.nature.com/articles/s42256-020-00257-z); [Recht et al., 2019](https://proceedings.mlr.press/v97/recht19a/recht19a.pdf)]. If we release any "OOD" data, we will have to find another test set and preserve it privately to maintain the reliability of OOD evaluation.

---

We appreciate the reviewers' generally positive feedback and their recognition of the value of our Touchstone Benchmark. We are committed to the **long-term** evaluation of algorithms submitted by any researcher (see our discussions and [HuggingFace webpage](https://huggingface.co/datasets/AbdomenAtlas/AbdomenAtlas1.0Mini)). Even after this paper's submission, we continue to receive **increasing requests** from researchers who want to join our Touchstone Benchmark. The number of participants has grown from **14**, as reported in this paper, to **55**. The next edition of this benchmark is now being hosted by MICCAI 2024, featuring **4,067** additional CT volumes and **16** more annotated structures (dataset released at [HuggingFace](https://huggingface.co/datasets/AbdomenAtlas/_AbdomenAtlas1.1Mini)). Therefore, we are excited to get your suggestions for the improvement of our Touchstone Benchmark in the long-term.

---

### Author Response · Authors · 2024-08-29

Dear Reviewers and AC,

We hope that previously posted rebuttal has addressed the concerns from the reviewers.

We are pleased to announce that the live leaderboard of our Touchstone Benchmark has been officially released to the public and is now available on our [webpage](https://github.com/MrGiovanni/Touchstone). The training data & labels, benchmark results, and analysis tools are fully accessible to everyone. We offer JHH (**5,176 manually annotated CT volumes**)—the largest test dataset for abdominal segmentation—for third-party out-of-distribution evaluation.

Our Touchstone Benchmark is evolving. The next edition is an open challenge at MICCAI-2024, offering **4,067** CT volumes and **16** annotated structures in addition to previous one. It has already attracted **55** research teams from **12** countries, and we anticipate even greater interest following our [call-for-benchmark](https://www.cs.jhu.edu/~zongwei/advert/Call4Benchmark.pdf).

Sincerely,

Authors of Paper 217

---

### Decision · Program_Chairs · 2024-09-26

**Decision:**

Accept (Poster)

**Comment:**

This paper presents a valuable benchmark for abdominal CT segmentation, offering a large-scale and diverse dataset of high value to medical AI evaluation. Key concerns were addressed by the authors, such as aligning the title and scope of the work and clarifying the use of standard metrics like Dice scores while also providing additional insights into demographic-based performance. Importantly, the decision to keep the JHH dataset private is justified to maintain the integrity of out-of-distribution testing, which is crucial for the medical community. Given these contributions and the thoughtful responses to feedback, I recommend acceptance.